# Constraining the rise of oxygen with oxygen isotopes

B.A. Killingsworth [1,5*], P. Sansjofre[1,6], P. Philippot [2,3], P. Cartigny[3], C. Thomazo [4] & S.V. Lalonde [1]

After permanent atmospheric oxygenation, anomalous sulfur isotope compositions were lost from sedimentary rocks, demonstrating that atmospheric chemistry ceded its control of Earth's surficial sulfur cycle to weathering. However, mixed signals of anoxia and oxygenation in the sulfur isotope record between 2.5 to 2.3 billion years (Ga) ago require independent clarification, for example via oxygen isotopes in sulfate. Here we show <2.31 Ga sedimentary barium sulfates (barites) from the Turee Creek Basin, W. Australia with positive sulfur isotope anomalies of $\Delta^{33}S$ up to $+1.55‰$ and low $\delta^{18}O$ down to $-19.5‰$. The unequivocal origin of this combination of signals is sulfide oxidation in meteoric water. Geochemical and sedimentary evidence suggests that these S-isotope anomalies were transferred from the paleo-continent under an oxygenated atmosphere. Our findings indicate that incipient oxidative continental weathering, ca. 2.8–2.5 Ga or earlier, may be diagnosed with such a combination of low $\delta^{18}O$ and high $\Delta^{33}S$ in sulfates.

[1] CNRS-UMR6538 Laboratoire Géosciences Océan, European Institute for Marine Studies, Université de Bretagne Occidentale, 29280 Plouzané, France. [2] Géosciences Montpellier, CNRS-UMR 5243, Université de Montpellier, Montpellier Cedex 5, France. [3] Institut de Physique du Globe de Paris, Sorbonne-Paris Cité, UMR 7154, CNRS-Université Paris Diderot, 75005 Paris Cedex 05, France. [4] UMR CNRS/uB 6282 Laboratoire Biogéosciences, Université de Bourgogne Franche-Comté, 6 Bd Gabriel, 21000 Dijon, France. [5] Present address: Institut de Physique du Globe de Paris, Sorbonne-Paris Cité, UMR 7154, CNRS-Université Paris Diderot, 75005 Paris Cedex 05, France. [6] Present address: Muséum d'Histoire Naturelle, Sorbonne Université, UMR CNRS 7590, Institut de Minéralogie, de Physique des Matériaux et de Cosmochimie, 75005 Paris, France. *email: bryan.a.killingsworth@gmail.com

Today, Earth's oxygen-rich (21% O$_2$ by volume) atmosphere drives a marine sulfur cycle dominated by oxidative weathering. Rivers supply the ocean with substantial loads of dissolved sulfate (SO$_4^{2-}$) derived from the oxidation of pyrite and the dissolution of sulfate minerals in roughly equal proportions[1]. Within marine and terrestrial settings, microbial sulfate reduction (MSR) processes exert the most important controls on sulfur isotopic fractionation of SO$_4^{2-}$. During MSR, the partial reduction of SO$_4^{2-}$ preferentially converts $^{32}$S to sulfide that is mostly re-oxidised, whereas a fraction of the sulfide product is retained within iron sulfide minerals[2]. Because of mass-dependent sulfur fractionations from MSR, marine sulfide has a large, but relatively $^{34}$S-depleted, range in $\delta^{34}$S (see Methods for isotope notations) and a $\Delta^{36}$S/$\Delta^{33}$S slope of approximately $-9$[3]. Simultaneously, marine SO$_4^{2-}$ is concentrated at 28 mM, enriched in $^{34}$S ($\delta^{34}$S = 21‰[4];, and has a small $^{33}$S enrichment ($\Delta^{33}$S < 0.1‰).

In contrast, before atmospheric oxygenation sufficiently drove the oxidation of sulfur at the Earth's surface, the surface sulfur cycle was first controlled by atmospheric inputs of sulfur that can be traced within Archaean age, 4.0–2.5 Ga, sedimentary rocks displaying strong sulfur mass independent fractionation (S-MIF) in their $\Delta^{33}$S values between +14‰ and $-4$[5–7] (Fig. 1a). There

are lingering geochemical[8] and quantitative[9] challenges still to be understood about the preservation and generation of Archaean S-MIF signals. However, it is generally accepted that S-MIF results from atmospheric photochemical reactions operating under low O$_2$ of <0.001% of the present atmospheric level (PAL) of oxygen, causing surface sulfur fluxes of insoluble S$_0$ with $\Delta^{33}$S > 0‰ and soluble SO$_4^{2-}$ with $\Delta^{33}$S < 0‰ that were not homogenised during their transfer into sedimentary rocks[10,11].

The disappearance of anomalous $\Delta^{33}$S signals from the sedimentary record is a crucial constraint on the rise of oxygen around the Archaean-Proterozoic boundary, but despite intensive efforts its exact timing remains unclear. This transition is constrained in continuous stratigraphic section only in S. Africa, within three different age-equivalent cores from the Transvaal Basin. There, $\Delta^{33}$S decreases below 0.4‰ between 2316 and 2326 ± 7 million years (Ma) ago, a horizon that has been interpreted as the shift to >0.001% PAL of oxygen and considered by some as the Great Oxidation Event (GOE) itself[6,12,13]. Alternatively, the slow disappearance of $\Delta^{33}$S signals from the rock record after 2.45 Ga may be attributable to the increasingly important oxidative weathering of an older, S-MIF-bearing, continental sulfide reservoir whose anomalous isotope compositions (i.e., $\Delta^{33}$S >

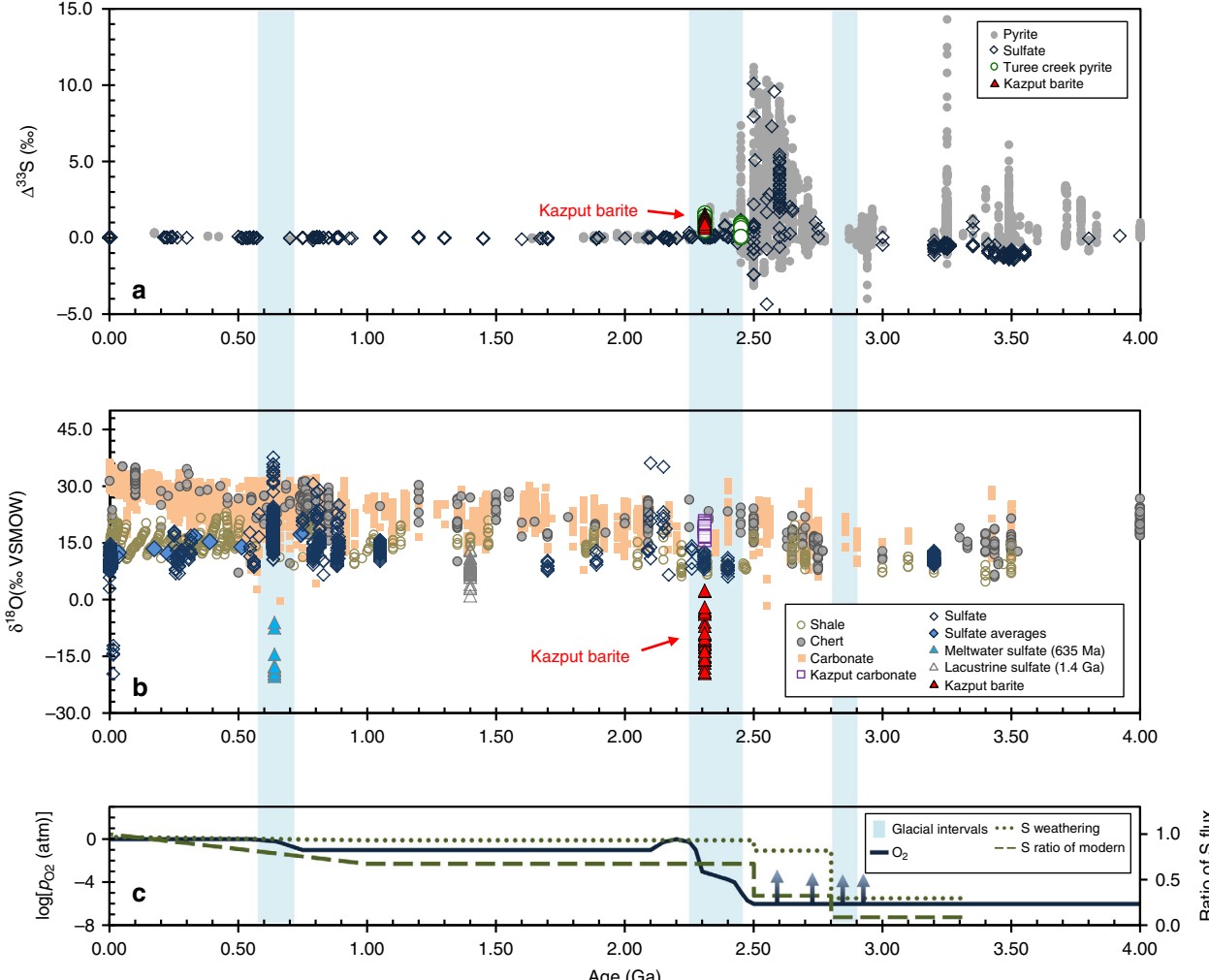

**Fig. 1** Kazput Formation barite sulfur and oxygen isotope data from this study are shown alongside compilations of time series data and temporal estimates of sulfur fluxes and atmospheric O$_2$. In further detail, shown are **a** pyrite and sulfate $\Delta^{33}$S data, **b** $\delta^{18}$O data of sedimentary carbonate, chert, shale, and sulfate and **c** the evolution of O$_2$ showing maximum ranges[67] with ratios of sulfur influxes to the ocean of total versus present day and weathering versus total flux[61]. The shaded regions represent intervals of glaciations of possibly global significance. The Turee Creek pyrite data in **a** are from[16], and Kazput carbonate data in **b** are from[33]. Data sources and details are provided in Supplementary Note 1, while source data are provided in a Source Data file

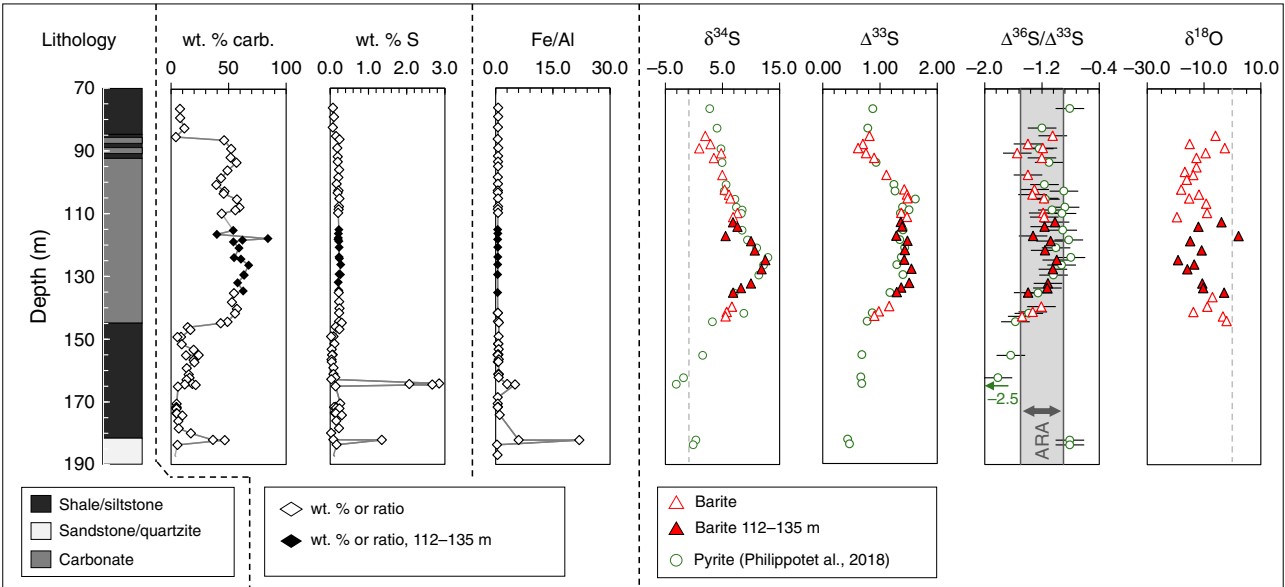

**Fig. 2** The stratigraphic distribution of barite stable isotope data ($\delta^{34}$S, $\Delta^{33}$S, $\Delta^{36}$S and $\delta^{18}$O), carbonate abundance (wt. % carb.), total sulfur (wt. % S), and Fe/Al mass ratios are plotted by their depth in drill core TCDP3 of the lowermost Kazput Formation. Pyrite isotope and weight percent sulfur data are from[16], while Fe/Al mass ratios are from[31]. Isotope data analytical uncertainties ($2\sigma$) are ± 0.1‰, ± 0.01‰, ± 0.2‰ and ± 0.4‰ for $\delta^{34}$S, $\Delta^{33}$S, $\Delta^{36}$S and $\delta^{18}$O, respectively, and are shown as error bars or are smaller than the symbols. The ARA on the $\Delta^{36}$S/$\Delta^{33}$S plot is the Archaean Reference Array, as in Fig. 3. The filled symbols are from the core depth interval 112–135 m with specific sulfur-oxygen isotope correlations that are discussed in the text and shown in Fig. 3. Figure source data are included in the Source Data file

0.4‰) would be transferred to sulfate until this source was either exhausted or negligible as compared to contemporaneous, non-anomalous, sulfur sources[14]. Subsequent evidence has supported[15,16] and challenged[17] this assertion. Because all of these studies interpret the record of S-MIF directly, independent tests are needed to delineate the importance of atmospheric and weathering controls on the early surficial sulfur cycle.

The sparse sulfate record during the Archaean Eon, and into the early Palaeoproterozoic, has prevented a closer examination of the oxidative side of Earth's early sulfur cycle. Evaporites and bedded barites are discontinuous, while more continuous records are hindered by weakly concentrated carbonate-associated sulfates (CAS) that are vulnerable to contamination[18], diagenetic alteration[19] and loss during metamorphic recrystallization[20]. Furthermore, as CAS concentrations should approximate ambient seawater $SO_4^{2-}$ concentrations, low early Earth seawater sulfate can challenge the recovery of measurable CAS. For example, between 2.5 and 2.3 Ga, it can be difficult to obtain sufficient CAS from sediments for sulfur isotope measurements, where up to 1500 grams of carbonate may be necessary[6]. Such low CAS yields, being especially vulnerable to contamination and overprinting, may discourage the additional measurement of oxygen isotopes.

Here, to investigate the influence of early Earth sulfide weathering, we focus on barite ($BaSO_4$), a highly insoluble and diagenesis-resistant mineral that offers a robust record of sulfate S and O isotope compositions[21]. The majority (70–92%[22]) of oxygen atoms incorporated in sulfide-derived $SO_4^{2-}$ are contributed by ambient water at the locus of oxidation (e.g., $\delta^{18}$O values of present-day seawater ≈ 0‰, meteoric waters ≈ 0‰ to −20‰, snow and ice ≈ −20‰ to −60‰[23]), with a lesser contribution from other oxidants such as $O_2$, followed by negligible isotopic exchange[24]. For example, sulfates in sediments are normally enriched in $^{18}$O versus their water oxygen source, where seawater at $\delta^{18}$O ≈ 0‰, or lower[25], results in marine $SO_4^{2-}$ being ≥6‰ through time (Fig. 1b). Meanwhile, MSR leaves a diagnostic positive correlation between the $\delta^{34}$S and $\delta^{18}$O of affected

sulfate[26]. Thus, after its formation, the relatively conservative behavior of $SO_4^{2-}$ may permit discrimination between its different environmental origins and transformations.

The 2.45 Ga[27,28] to 2.2 Ga[29] Turee Creek Group sedimentary succession from W. Australia is ideal for seeking barite records to characterize the response of the surface sulfur cycle around the time of atmospheric oxygenation. Here we report Turee Creek Group barites with positive $\Delta^{33}$S values and distinctly negative $\delta^{18}$O values whose paleoenvironmental and temporal context suggest a first-order weathering control, mediated by biologically enhanced preservation, on their anomalous sulfur isotope signals ca. 2.3 Ga.

## Results and discussion

**S- and O- isotopic signatures from Turee Creek barites**. All of our barites register signals of S-MIF, featuring $\Delta^{33}$S values from +0.62‰ to +1.55‰ and $\Delta^{36}$S/$\Delta^{33}$S ratios falling within −0.9 to −1.5‰ (Figs. 2, 3, Supplementary Table 1). The $\delta^{18}$O values span + 2.2‰ to −19.5‰, with an average of −11.0‰ (VSMOW). The sulfur $\delta^{34}$S, $\Delta^{33}$S and $\Delta^{36}$S and oxygen $\delta^{18}$O measurements are obtained from trace barites of the lower Kazput Formation that were chemically extracted from drill core 3 of the Turee Creek Drilling Project (TCDP3) (Supplementary Figs. 1 and 2). Although barites were observed in acid-digested residues, the grains were too small (<3 μm) to be identified via standard transmitted light microscopy, and thus only imaged via SEM (Supplementary Figs. 3 and 4).

**Geological and geochemical context**. The analysed barites are from the Kazput Formation of the Turee Creek Group, an overall-shallowing succession with Mn-enriched units and decreasing iron contents that have been interpreted in favor of increasing atmospheric oxygen[30]. Recently reported nitrogen isotope and iron speciation evidence from the Kazput Formation further indicates free oxygen availability during its deposition[31]. An age of 2.25 Ga was assigned, as based on detrital zircon U-Pb dating and estimated sedimentation rates[28]. While this age may be

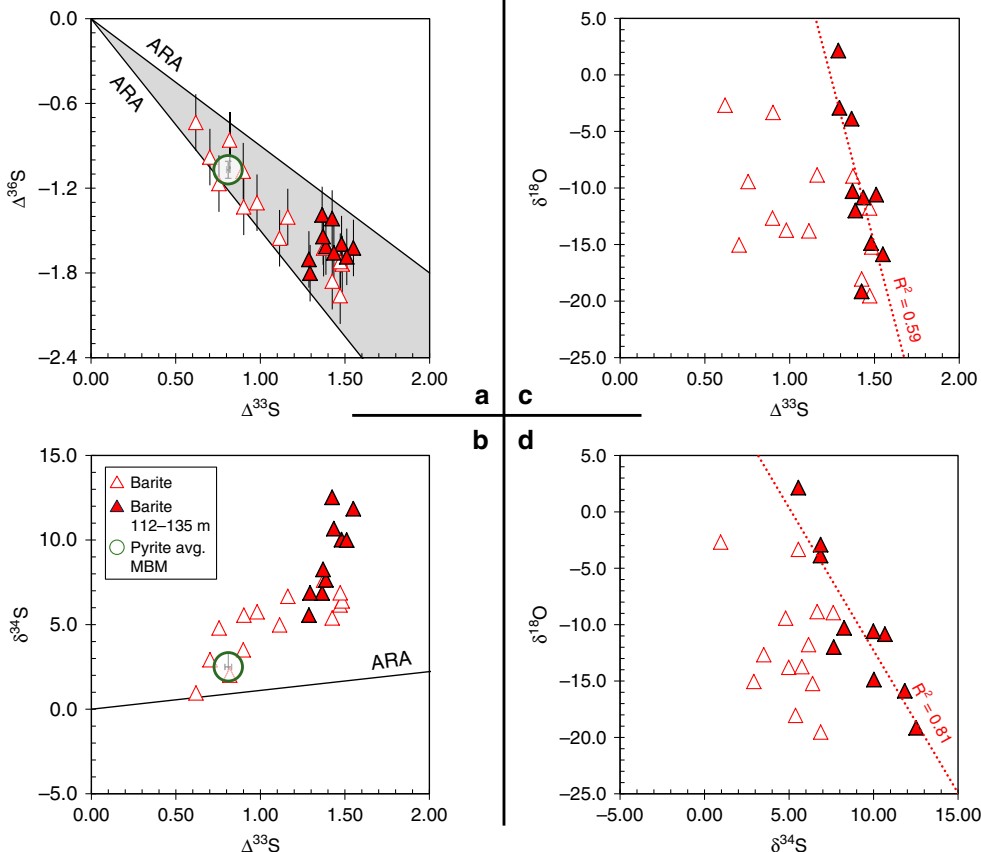

**Fig. 3** sulfur and oxygen isotope systematics of barite data are shown from the Kazput Formation carbonate. The two left-hand plots, **a**, **b** show only sulfur data, while **c**, **d** show both oxygen and sulfur data. Analytical uncertainties are the same as in Fig. 2. ARA: Archaean Reference Array;[68] pyrite avg. MBM: average isotope composition of pyrite from the Meteorite Bore Member diamictite[16]. The ARA describes quadruple sulfur isotope correlations during the Archaean attributed to mass independent processes, where $\Delta^{33}S = 0.9 \times \delta^{34}S$, and $\Delta^{36}S = -0.9$ or $-1.5 \times \Delta^{33}S$[68]. See Methods for isotope notation definitions and the Source Data file for data

appropriate, in order to be conservative with respect to what may be a remarkably young record of isotopically anomalous sulfur, here we simply assume that the Kazput Formation is younger than 2.31 Ga, the age constraint from the lower Meteorite Bore Member diamictite[16]. Isotope compositions of Kazput Formation carbonates are $\delta^{13}C = -2‰$ to $+1.5‰$ and $\delta^{18}O = -16.63‰$ to $-8.13‰$ (VPDB), suggesting that diagenesis has not erased their record of primary seawater chemistry[32]. Geologic maps locating the TCDP cores (Supplementary Fig. 1), core photographs (Supplementary Fig. 2), and transmitted light photomicrographs with complementary XRF analyses (Supplementary Figs. 5, 6) are provided in the Supplementary Information. The lower Kazput contains finely laminated mud- and silt-stones that shallow into laterally variable oolitic and stromatolitic carbonates with deltaic influence[32]. Our examined carbonates are fine-grained with mm- to cm-scale laminations, resembling facies association A of Barlow and co-authors[33], who further described local occurrences of dome-shaped stromatolites. The carbonate horizon gradates into, and out of, finely cross-laminated carbonate-rich siltstones (Fig. 2).

The strong $^{18}O$-depletion of the Kazput barites appear consistent with sulfide-derived $SO_4^{2-}$ being the carrier of S-MIF signals to the W. Australia sedimentary record after 2.31 Ga. Such low $\delta^{18}O$ values in sulfates are only known from terrestrial settings (e.g., glacial, high latitude, or lacustrine) (Fig. 1b) that have distinctly $^{18}O$-depleted water sources[23]. The significant S-MIF of the barite that falls within the Archaean $\Delta^{36}S/\Delta^{33}S$ array

(ARA, Fig. 3a) suggests that the sulfur source was originally fractionated in an atmosphere containing <0.001% PAL oxygen. Importantly, the barite S isotope compositions closely overlap with those of previously reported early diagenetic pyrites in the same TCDP3 core (Fig. 2), which implies that the sulfur precursor was mostly unmodified during its conversion to reduced (sulfide) or oxidised (sulfate) mineral phases, or that one phase was derived from the other without isotopic fractionation. Because, under various conditions, it is possible for low temperature redox reactions to proceed without S-isotope fractionation between sulfate and reduced sulfur phases[34], by itself, the S-isotope match between Kazput barite and pyrite cannot be used to diagnose a specific process. Broadly, however, the S-MIF in Kazput barite requires an atmospheric sulfur source that originated under an atmosphere with <0.001% PAL $O_2$, occurring within one of two scenarios: either the capture of such atmospheric sulfur under an essentially oxygen-free atmosphere existing at <2.31 Ga, or the reworking/recycling of an archival atmospheric sulfur source that was generated any time before 2.31 Ga. Understanding how and when these barites inherited their atmospherically derived sulfur is of fundamental importance to understanding Earth's oxygenation and its imprints on the minor sulfur isotope record.

**Assessing the origin of Kazput barite.** Given that pyrite-derived $SO_4^{2-}$ can closely match the sulfur isotope composition of its pyrite precursor under abiotic or biologically mediated oxidation[34,35], the nearly identical $\delta^{34}S$ (and $\Delta^{33}S$) compositions

of Kazput barite and pyrite could be attributed to low temperature pyrite oxidation during sample preparation in the laboratory. While pyrite oxidation produces dissolved $SO_4^{2-}$ as its immediate product, barite formation further requires the capture of $SO_4^{2-}$ with barium, which makes it more difficult to accidentally produce in the lab during sample handling. This requirement is important because fluids can be concentrated in barium or $SO_4^{2-}$, but not both, due to barite's low solubility[36]. We performed extensive tests of pyrite oxidation during barite extraction, as well as during attempted extractions of CAS, where pyrite oxidation during sample processing was excluded (see Methods).

A metamorphic origin for the barite must also be considered, as it could explain the occurrence of their S-MIF signals, irrespective of the host rock's age. For example, post-depositional metamorphic processes have been implicated in the remobilization and dilution of S-MIF signals in sulfur from N. American Palaeoproterozoic Huronian sections[37]. The metamorphism of the Turee Creek Group is constrained to prehnite-pumpellyite–epidote facies up to 300 °C[38]. Additional constraint from a SIMS study on the Turee Creek Group yields a maximum metamorphic temperature of 240 °C that was estimated from sharp sulfur isotope gradients of $\delta^{34}S$ ranging over 30‰ across a <4 μm transect in pyrite[39]. More importantly, previous work on trace element abundances in pyrites from TCDP drill cores has demonstrated the early diagenetic origin of their sulfur[16]. Considering the S isotope match between the Kazput barite and pyrite, we conclude that the barites are, likewise, of low temperature origin without significant metamorphic alteration.

A detrital origin for Kazput barites could provide another explanation for the unexpected young age, <2.31 Ga, of their sulfur isotope anomalies. Evidence for an anoxic atmosphere that is found in sedimentary rocks are rounded detrital grains of pyrite and uraninite that were not dissolved during riverine transport due to low oxygen availability[40]. The pyrite in the Kazput Formation is not detrital, and instead appears to have grown in situ. Kazput pyrites occur as microcrystalline aggregates, euhedral crystals, and elongated layer-parallel concretions in a range of sizes (Supplementary Figs. 5 and 6). In contrast, sedimentary reworking of Kazput barites is difficult to ascertain due to the barite size, <10 μm, and disseminated occurrence (Supplementary Figs. 3 and 4), but they share a tight S-isotopic match with the co-occurring diagenetic pyrites. Therefore, the mutual sulfur source for the barite and pyrite, despite different mode of occurrence, also excludes a detrital origin for the barite.

Secondary oxidation of pyrite to $SO_4^{2-}$ could occur at low temperature within the rock during its burial history, for example, via pyrite oxidation by pore waters. Firstly, it is feasible to oxidize pyrite under anoxic conditions via radiolysis of water that is accompanied by an enrichment of 1.5–3.4‰ in the $\delta^{34}S$ values of $SO_4^{2-}$ versus pyrite[41], but this scenario is ruled out because Kazput barite does not show such S-isotope enrichment versus the previously reported pyrite (Fig. 2). Meanwhile, if late oxidation by infiltrating fluids occurred, then carbonate, which is very susceptible to diagenetic alteration of its oxygen isotope composition[42], should register alteration by the same $^{18}O$-depleted waters implicated in the oxidation of pyrite to form the barite. Instead, comparison of $\delta^{18}O$ data from Kazput carbonate with our barite reveals separate environments of formation for the carbonate and $SO_4^{2-}$. The Kazput carbonate has an average $\delta^{18}O_{carb} = 18.4‰$ (here converted to the VSMOW scale)[33] that is typical for marine carbonates of similar age, but 10‰ lower than present-day marine carbonates (Fig. 1b). As reviewed by Gomes and Johnston[34], the oxygen isotope fractionation between pyrite-derived $SO_4^{2-}$ and water spans a range of $\delta^{18}O$ values between 0‰ and +20‰ for naturally

relevant conditions. Oxidation experiments using waters with $\delta^{18}O$ compositions >15‰ have produced sulfide-derived sulfates that have very low $\delta^{18}O$ compositions versus their source waters, as exemplified by an extreme case of −65‰ fractionation between pyrite-derived sulfate (+71‰) oxidised in isotopically labeled water (+127‰)[43]. Although the topic of sulfate oxygen isotope fractionation demands clarification, we contend that Kazput barites are well within the range of $\delta^{18}O$ values, below 15‰, of natural sulfates that are $^{18}O$-enriched versus their water oxidant sources. Regardless of assumptions concerning sulfate-water oxygen isotope fractionations, distinct water–oxygen sources are required to produce sulfate $\delta^{18}O$ compositions that are appreciably lower than those of coeval carbonates. These low sulfate compositions are apparent in non-marine sulfates as compared to concurrent marine carbonates at ca. 0.0, 0.6 and 1.4 Ga (Fig. 1b). Similarly, relative to coeval Kazput carbonates, the low $\delta^{18}O$ values of our barites require a meteoric-water–oxygen source.

A unique combination of environmental conditions likely contributed to the isotopic signatures of Kazput barite, but perhaps they were also preserved due to the sulfuretum, or consortium of sulfur-metabolising microbiota, active in the Turee Creek Basin during the Palaeoproterozoic. Besides prevalent microbial mats in the Kazput carbonate, it also contains microfossils that have been interpreted as sulfur-oxidizing filamentous bacteria[44]. What is remarkable about such bacteria from an isotopic perspective is that a sulfuretum of $SO_4^{2-}$-reducing microbes and sulfur-oxidizing bacteria can produce sulfide that gets re-oxidized to $SO_4^{2-}$, via elemental sulfur, with no net S isotope fractionation[45]. Sulfur-oxidizing bacteria may have contributed to the Kazput barites' oxygen isotope signals, perhaps by introducing Turee Creek Basinal water–oxygen to the $SO_4^{2-}$ produced during re-oxidation of MSR-produced sulfide, or by enhancing rates of sulfide oxidation on land. An MSR influence can be indicated by positive $\delta^{18}O$–$\delta^{34}S$ correlation in sulfate[46], however, the negative correlations between Kazput barite $\delta^{18}O$ and its sulfur isotope parameters (Fig. 3c, d) imply that source mixing, with sulfide weathering in meteoric waters, was much more important than MSR for setting these S and O isotope signatures. As the source-controlled barite sulfur and oxygen isotope variations mask microbial influence, further speculation about the biological role in Turee Creek sulfur cycling remains open.

**A continental source of isotopically light oxygen.** We suggest, given the previous evidence that has ruled out post-depositional processes and a detrital origin while pointing to a meteoric water oxygen source, that the O isotope signals preserved in Kazput barite are best explained by pyrite oxidation on land. Kazput barite $\delta^{18}O$, extending to −19.5‰, are among the lowest values in the sedimentary record, including other sulfate occurrences going down to −20.3‰ from a Neoproterozoic snowball Earth glacial diamictite deposited at 635 Ma[47], and in relatively more recent terrestrial deposits from within the Arctic Circle that reach down to −19.7‰[22] (Fig. 1b). Similarly, the range of oxygen isotope compositions of Kazput barite may require a glacial meltwater oxidant source. However, in contrast to those other examples of very low $\delta^{18}O$ in sulfate, Kazput barite $\delta^{18}O$ may not necessitate glacial or ice meltwater oxygen sources, and instead could reflect a hydrosphere anchored to seawater with a lower $\delta^{18}O$ than today. Meteoric waters evolve lower $\delta^{18}O$ compositions than seawater due to evaporation from seawater followed by Rayleigh distillation in cooling air masses[23]. The most evolved meteoric waters (and snow and ice melt) that can be found at high latitudes, high elevations and away from coastlines are

marked by the lowest $\delta^{18}O$ compositions relative to seawater. This relative relationship between seawater and more $^{18}O$-depleted meteoric waters is significant here, as it has been suggested that the $\delta^{18}O$ of seawater (0‰ at present[23]) may have changed through time, perhaps being as low as −10‰ in the Archaean[25]. Newly reported iron oxide $\delta^{18}O$ records, which are insensitive to temperature, convincingly support that seawater was lower in the past, having a $\delta^{18}O$ at near −8‰ by around 2.0 Ga[48]. Palaeoproterozoic seawater appears to be faithfully recorded in Kazput carbonate that is, as mentioned previously, on the order of 10‰ lower than modern carbonates but comparable to carbonates from around 2.3 Ga. Therefore, the Kazput barites recording the lowest $\delta^{18}O$ indicate their precursor sulfate was oxidised in the most evolved meteoric waters, placing this water–oxygen source for this sulfate firmly on land, as compared to a seawater $\delta^{18}O$ that was likely around −10‰. Although sulfate-water oxygen isotope fractionation during sulfide oxidation requires further study, we take a median sulfide-derived sulfate-water $\delta^{18}O$ fractionation value of +10‰[34], as compared to the barite, to roughly estimate that the water sources for Kazput sulfate may have ranged between −30‰ and −8‰. Considering a possible seawater composition around −10‰, this range of source water $\delta^{18}O$ is appropriate for meteoric sources. Alternately, if ca. 2.3 Ga seawater resembled a contemporary $\delta^{18}O$ composition, the estimated range of water $\delta^{18}O$ overlaps seawater while extending from meteoric to glacial waters. Regardless, the association between low $\delta^{18}O$ and high $\Delta^{33}S$ in the Kazput barites implicates continental weathering of sedimentary sulfides as the vector carrying S-MIF to the Turee Creek Basin.

**A model of Kazput barite formation.** Taken together, the S- and O-isotope systematics suggest a singular scenario for the origin of the Kazput barite. The barite and pyrite $\Delta^{33}S$ values reach a maximum in the studied drill core as compared to the two other Turee Creek Group drill cores in older underlying sediments[16]. The strongest expression of this sulfur source appears with maxima of $\Delta^{33}S$ and $\delta^{34}S$ coincident with the $\delta^{18}O$ minimum in barite, where negative correlations between $\delta^{18}O$ and $\Delta^{33}S$ and $\delta^{34}S$ data exist between 112 and 135 metre depth within the center of the carbonate unit (Fig. 3c, d). Such S–O isotope correlation may indicate the mixing of sulfide-derived $SO_4^{2-}$ sources with distinct S-isotope compositions but overlapping $\delta^{18}O$ ranges due to their oxidation in meteoric waters under variable humidity, as is observed in major river systems today[49,50]. The preservation of such a strongly riverine sulfate signal in marine sedimentary rocks is unlikely today due to buffering by high contemporary seawater sulfate concentrations. A previously identified source of sulfur in the Turee Creek Basin with a monotonous $\Delta^{33}S \approx 0.9‰$ is well expressed in the Meteorite Bore Member diamictite[16]. Mixing between this monotonous sulfur source and a different, continental-weathering, source with a higher $\Delta^{33}S \approx 1.6‰$ is observed in S–O isotope space (Fig. 3c). The first-order controls on the mix of sulfur sources are relative sea level and the exposure of sedimentary rocks to weathering. Minimal overprinting resulted in a largely faithful transfer of S- and O-isotope signatures as sulfate was captured as barite or reduced to form pyrite (0.32 weight % on average, Supplementary Table 2). Anoxic sediment porewaters favoring MSR would provide a flux of barium in sufficient concentration to precipitate $SO_4^{2-}$ as barite at the sediment-water interface[51]. Remaining porewater $SO_4^{2-}$ would be quantitatively reduced to sulfide, via closed system MSR, with capture by iron to form iron sulfides (pyrites) during early diagenesis in the sediment[16]. This is consistent with the lack of CAS-sulfate from the Kazput carbonate

that supports near-quantitative reduction of $SO_4^{2-}$ in porewaters. Previously reported Fe/Al mass ratios in core TCDP3 and the Kazput carbonate are 0.64 on average, excluding enriched intervals >1 (Fig. 2, Supplementary Table 3). The Fe/Al and total sulfur enrichments imply either transient water column anoxia, or movement of a chemocline during transgression-regression cycles that could in turn represent enhanced sulfate fluxes from land and their drawdown into anoxic sediments via microbial sulfate reduction into sulfide. The enrichments in Fe/Al and total sulfur observed at 164 and 182 meters depth in two muddy intervals below the carbonate are also likely pyrite ($FeS_2$) enrichments, though they were not directly quantified here.

**Atmospheric and sedimentary controls on anomalous sulfur.** Barites isolated from Kazput carbonate show the $\Delta^{36}S/\Delta^{33}S$ slope of −0.9 (Fig. 3a) that typifies much of the Archaean sulfur isotope record. This slope, however, steepens to near −1.5 within the upper and lower portion of the carbonate (Fig. 2). Close links between low $\delta^{13}C$ values and steeper $\Delta^{36}S/\Delta^{33}S$ slopes (−1.5) in Neoarchaean sedimentary rocks[52] have been interpreted as evidence for changes in atmospheric chemistry catalysed by the biosphere[53]. In comparison to these Neoarchaean records, however, Kazput barite show relatively small $\Delta^{36}S$ and $\Delta^{33}S$ ranges. Furthermore, recent iron-speciation and nitrogen isotope evidence imply that suboxic-to-oxic water column conditions prevailed beneath an atmosphere containing appreciable oxygen[31]. The apparent availability of significant free $O_2$ (>0.001% PAL), concurrent with mass independent sulfur isotope anomalies, strongly supports the latter being due to a memory effect carried by sulfate derived from oxidative weathering of S-MIF-bearing rocks[15]. It follows that the attendant $\Delta^{36}S/\Delta^{33}S$ variations in the barites may reflect control from oxidative weathering of pyrite in continental source rocks of various ages, whereas the S-MIF signal is simply tracking the integrated sulfur isotope composition of the rocks being weathered. As interpreted previously, the S-MIF systematics and their limited range in the Turee Creek Group may represent homogenisation during the weathering of older, more isotopically variable sulfur generated under earlier atmospheric states, together with dilution by addition of sulfate lacking S-MIF ($\Delta^{33}S \approx 0‰$)[16].

Despite being deposited in relative proximity, within similar equatorial intracratonic basins[7,12,30], W. Australian and S. African records display stark sulfur isotopic differences between 2.45 and 2.2 Ga. These differences may partly reflect spatial and temporal variations of $SO_4^{2-}$ concentrations. Geochemical constraints from the first basin-scale bedded evaporites at 2 Ga indicate a significant marine $SO_4^{2-}$ concentration of at least 10 mM[54], however, between 2.25 and 2.1 Ga, the $\delta^{34}S$ variability in extant CAS records imply a range of seawater $SO_4^{2-}$ concentrations of 5-20 mM[55]. Meanwhile, modern analog environments suggest late Archaean marine $SO_4^{2-}$ concentrations of <2.5 μM[56]. As a result of such sulfate concentrations that could span four orders of magnitude between 2.5 and 2.1 Ga, sulfur isotope variations would be very sensitive to local environmental controls. We suggest that such local control extends to the record of sulfur isotope anomalies. As compared to S. Africa, the W. Australian Turee Creek Group succession seems to require within-sediment closed system MSR to preserve its persistent S-MIF signals. Frequency distributions of $\delta^{34}S$ data support this assertion, with W. Australian sulfur $\delta^{34}S$ data displaying a broadly unimodal distribution, centered near 5‰ (Fig. 4a); while S. African data show a more bimodal $\delta^{34}S$ distribution (Fig. 4b), reflecting the preferential incorporation of $^{32}S$ into sulfide at the expense of sulfate. These different behaviors are also manifested in their respective $\Delta^{33}S$ distributions. Despite similar ranges of $\Delta^{33}S$

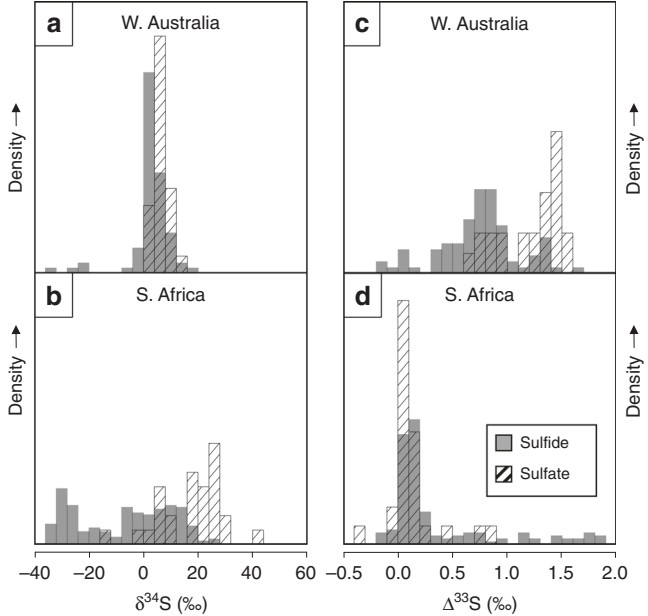

**Fig. 4** Histogram density plots of bulk sulfur isotope data of sulfides and sulfates from S. Africa and W. Australia of ca. 2.2–2.45 Ga age. The data are organised by $\delta^{34}S$ (**a**, **b**) and $\Delta^{33}S$ (**c**, **d**). The plots of S. African $\delta^{34}S$ (**b**) and $\Delta^{33}S$ (**d**) data ($\delta^{34}S_{sulfide}$ $n = 217$, $\delta^{34}S_{sulfate}$ $n = 28$, $\Delta^{33}S_{sulfide}$ $n = 164$ and $\Delta^{33}S_{sulfate}$ $n = 26$) are from[6,12,13,69]. W. Australian $\delta^{34}S$ (**a**) and $\Delta^{33}S$ (**c**) sulfide data ($\delta^{34}S_{sulfate}$ and $\delta^{34}S_{sulfide}$ $n = 75$) are from[16], while sulfate data ($\delta^{34}S_{sulfate}$ and $\Delta^{33}S_{sulfate}$ $n = 24$) are from this study. Source data are compiled in the Source Data file. Note that the $\Delta^{33}S$ range truncates the maximum value observed from the S. African records (7.35‰). We refrain from comparing against time-equivalent N. American data due to its sulfur isotope records being possibly compromised by metamorphic overprinting[37]

values, an important source of mass-dependent sulfur ($\Delta^{33}S \approx 0$‰, Fig. 4d) is indicated for S. Africa, while two sources of S-MIF-bearing sulfur ($\Delta^{33}S = +0.9$‰ and $+1.6$‰, Fig. 4c) are observed from W. Australia. Closed system MSR is also suggested by the $\delta^{34}S$-$\Delta^{33}S$ systematics. Similar to previous observations from microbial carbonates[57], the $^{34}S$-enrichment versus the ARA seen in Kazput pyrites and barites (Fig. 3b) could reflect the isolation of the sediment from the water column due to carbonate precipitation. Indeed, stromatolites and carbonate are prevalent throughout the Turee Creek Group, especially in the Kazput Formation. Finally, to help elucidate the possibility that the Turee Creek Basin may have been relatively more isolated from the global ocean as compared to contemporaneous S. African basins, the $\delta^{18}O$ compositions of CAS from S. Africa may be useful but are yet to be measured. As suggested by the S-isotope evidence for open system MSR, it is likely that sulfate from S. Africa is relatively enriched in $^{18}O$ as compared to the Kazput barite.

**Implications for constraining atmospheric oxygenation**. Beyond its production in an essentially oxygen-free atmosphere, the preservation of S-MIF is susceptible to second-order biotic and abiotic controls. Consequently, several hypotheses have been proposed to explain the temporal structure of the S-MIF record (Fig. 1a). For example, the enhanced preservation of large magnitude $\Delta^{33}S$ compositions in the lead up to 2.5 Ga have been explained by a shift in the locus of MSR from euxinic water columns to sediment porewaters due to oxygenation of shallow surface oceans[58], or by the changing oxidation state of gas

influxes[59]. Conversely, contraction of $\Delta^{33}S$ signals, such as observed in pyrite between 2.5 and 2.3 Ga, could be caused by dilution of sulfur isotope anomalies with sulfide biologically produced from mass-dependent sulfur sources[58]. The ca. 2.5–2.3 Ga weak signals of S-MIF could reflect millions of years of weathering out and dilution of anomalous sulfur that had previously been stockpiled in more reduced, pre-GOE (i.e., older), sediments[14,15], as is likely the case for the Turee Creek Group[16]. Another possibility is that oscillating atmospheric oxygen levels after 2.5 Ga[7] permitted intervals of primary atmospheric S-MIF production and preservation as oxygen intermittently dipped below 0.001% PAL. In any case, it remains difficult to clearly recognize these various scenarios because the signals they leave behind in sediments can be ambiguous. Perhaps more detailed examination of linked $\delta^{13}C$ and $\Delta^{36}S/\Delta^{33}S$ slope variations from ca. 2.5 to 2.3 Ga will reveal more about the possibly of oscillating levels of atmospheric gases in that interval. That oxidative weathering should be the main control on marine sulfate after atmospheric oxygenation seems intuitive, but as mentioned, even under the high atmospheric oxygen concentration of the modern Earth, surface weathering does not directly control marine sulfate S-isotope values in lieu of MSR. The dominance of positive $\Delta^{33}S$ values in the Kazput barites implies the $SO_4^{2-}$ originated from the oxidative weathering of pre-GOE sedimentary rocks. Our results are at odds with the conclusion, based on contemporary weathering of old exposures, that the weathering of Archaean continents would yield sulfate with $\Delta^{33}S$ summing to nearly 0‰[17]. Perhaps this discrepancy is because present-day rock exposures simply are not comparable in composition and weathering susceptibility to the exposed sedimentary rocks at Earth's surface in the Palaeoproterozoic. Furthermore, small positive $\Delta^{33}S$ have indeed been measured in riverine sulfate from catchments in S. Africa and Ontario today[60], which suggests that recycling of S-MIF signals at the Earth surface could occur at least locally. From the Archaean to the the earliest Palaeoproterozoic ca. 2.5–2.3 Ga, when sulfur was being delivered by incipient oxidative weathering and/or atmospheric inputs, the marine $SO_4^{2-}$ reservoir was smaller and its fraction buried as pyrite approached unity[2]. Under these conditions, it is possible that the S-isotope compositions of marine sulfate and sulfide records could approach those of S-MIF-bearing riverine $SO_4^{2-}$ sources. In turn, early Palaeoproterozoic sulfur records, especially from basins with an unknown degree of connection to the global ocean, can be expected to show substantial variability. As spatial and temporal coverage of Palaeoproterozoic sulfide and sulfate records increases, a clearer composite portrait should emerge that will allow for more clearly differentiated local versus global symptoms of atmospheric oxygenation being recognized in the sulfur cycle at different scales.

Our findings suggest that the oxygen isotope composition of sulfate may prove as insightful as multiple sulfur isotope data in evaluating the imprint of atmospheric oxygenation within the sedimentary record. The combined negative $\delta^{18}O$ and positive $\Delta^{33}S$ compositions of Kazput barite imply weathering, as opposed to atmospheric, control on sulfur input fluxes by ~2.31 Ga in the Turee Creek Basin. It follows that the onset of atmospheric oxygenation and its nascent imprint on the sulfur record must be older. It has been suggested that $SO_4^{2-}$ flux to the oceans increased in advance of the GOE due to oxygen production and its consumption by sulfide weathering on land[61,62] (Fig. 1c). The conspicuous absence of negative $\Delta^{33}S$ anomalies from rocks younger than 2.42 Ga (Fig. 1a) may mark the turning point when weathering derived $SO_4^{2-}$ fluxes, with predominantly positive $\Delta^{33}S$ anomalies, first dominated the surficial sulfur cycle. By using oxygen isotope compositions to identify sulfide-derived sulfates of continental weathering origin from the rock record, it may be possible to directly constrain the timing of early weathering fluxes

of $SO_4^{2-}$ and their relative significance for the late Archaean surficial sulfur and oxygen cycles.

## Methods

**CAS extraction.** Carbonate-associated sulfate extractions were completed at the Laboratoire Géosciences Océan at IUEM in Plouzané, France. Drill core samples were first manually crushed in a tungsten-carbide piston chamber before powdering in an agate ring and puck mill. Following the CAS extraction sequence tested by Wotte et al.[63], soluble sulfate phases were removed from powdered carbonate samples via triplicate 10% NaCl leaches. The carbonate component was then dissolved by slow addition of 12 M HCl until no reaction was observed, thus liberating CAS into solution. The CAS was then precipitated as $BaSO_4$ upon addition of a supersaturated $BaCl_2$ solution. The resulting trivial CAS yields could be readily compromised by laboratory-induced pyrite oxidation from Kazput samples (~0.32 weight % pyrite, Supplementary Table 2). Therefore, we developed a modified CAS extraction protocol, using the reducing agent hydroxylamine hydrochloride to inhibit pyrite oxidation during CAS extraction. Unfortunately, the hydroxylamine hydrochloride itself was found to contain sufficiently concentrated trace sulfate to contaminate the CAS sample target. The result of CAS extraction with hydroxylamine hydrochloride on Kazput carbonate sample gave a final yield of <2 ppm whole-rock CAS, all of which could be attributed to contamination from the hydroxylamine hydrochloride. Thus, it was concluded that CAS in these samples is too low for precise bulk S- and O-isotope measurements, and CAS extractions on these samples was not pursued further.

**Barite extraction.** Our barite extraction technique utilises a chelating agent that at once enables dissolution of barite while inhibiting the oxidation of pyrite. Again, the barite extractions were completed at the Laboratoire Géosciences Océan at IUEM in Plouzané, France. As a test for pyrite oxidation in our extraction technique, finely ground pyrite was stirred in the chelating solution for a week followed by attempted recovery of pyrite-derived $SO_4^{2-}$ from the filtered solution, with no resulting sulfate yield, likely because the chelation of iron and other metal cations also served to prevent pyrite oxidation. In addition, we intentionally oxidized pyrite in the same laboratory-distilled water that is used in the barite extracting solution followed by measurement of the oxygen isotope composition of the produced sulfate, which was isotopically distinct from that of the measured samples (Supplementary Fig. 7).

Barite in TCDP3 was only observed by scanning electron microscopy, where barites <10 μm were observed in both sample residues (Supplementary Figs. 3 and 4) and in thin sections. Barite was not observed during conventional optical petrographic observations of thin sections. Barite extractions were performed on ~100 g of powdered drill core rock sample using a modified barite purification technique (the DDARP method) from Bao 2006[64] that was originally developed to purify sulfate samples for triple oxygen isotope measurements. Sample powders were decarbonated in HCl-acidified solution, rinsed in distilled water, and then treated for 3 days with constant stirring in a 0.05 M Diethylenetriaminepentaacetic acid (DTPA) and 1 M NaOH solution to dissolve barium sulfate. The supernatant was then separated from the sample residue by vacuum filtration. After acidification (pH < 2) with HCl, the barite-housed sulfate was precipitated out of the supernatant as $BaSO_4$ via addition of supersaturated $BaCl_2$ solution. Co-precipitation of silicates dissolved in the basic DTPA solution renders the precipitated $BaSO_4$ impure, requiring an additional purification step. Here, residual DTPA must first be removed by triplicate washing with distilled water, centrifugation, and decanting of the supernatant. After heated overnight at 70 °C in a 2 M NaOH solution, the sample was again centrifuged and the supernatant decanted. The sample was once again redissolved in 0.05 M DTPA and 1 M NaOH solution by agitating overnight then re-precipitated by acidification with HCl (to pH < 2) and addition of $BaCl_2$ solution. The $BaSO_4$ precipitate is finally washed in triplicate and dried at 70 °C overnight, then ready for isotope analysis.

**Oxygen isotope (δ18O) measurements.** Purified barite samples were measured in duplicate on an Elementar vario PYRO cube elemental analyzer in-line with an Isoprime 100 mass spectrometer in continuous flow mode at the University of Burgundy in Dijon, France. Oxygen isotope data are expressed in delta notation, $\delta \equiv R_{sample}/R_{standard} - 1$, where $R$ is the mole ratio of $^{18}O/^{16}O$ and reported in units per mille (‰, i.e.×1000). The $\delta^{18}O$ data are reported with respect to the international standard Vienna Standard Mean Ocean Water (VSMOW). Analytical errors are ±0.4‰ (2σ) based on replicate analyses ($n = 21$) of the international barite standard NBS-127 (Supplementary Table 4), which was used for data correction via standard-sample-standard bracketing.

**Quadruple sulfur isotope (δ34S, Δ33S and Δ36S) measurements.** Purified barite samples required additional wet chemistry for quadruple sulfur isotope analysis, all done at the Institut de Physique du Globe de Paris in Paris, France. Approximately 3 mg of each barite sample was converted to $Ag_2S$ by heating in an anoxic distillation apparatus in a sub-boiling solution of HCl, HI, and $H_3PO_2$ after Thode, Monster and Dunford[65] and Pepkowitz and Shirley[66]. Barite was thus converted to $H_2S$ that was then flushed by $N_2$ gas to an awaiting trap of silver nitrate solution where the $H_2S$ was precipitated as $Ag_2S$. The sample $Ag_2S$ precipitate was washed in triplicate in distilled water, oven-dried overnight, then ready for sulfur isotope analysis.

Quadruple sulfur isotope analyses were made by first heating ~3 mg sample $Ag_2S$ under an excess of $F_2$ gas (~300 torr) at 350 °C in nickel bombs overnight. The produced $SF_6$ gas was then purified via cryogenic trapping and separation by gas chromatograph before introduction to the mass spectrometer, a Thermo Finnigan MAT 253 running in dual inlet mode, for determination of quadruple sulfur isotope composition. Delta notation is used to report $^{34}S$, with $\delta \equiv R_{sample}/R_{standard} - 1$, where R is the mole ratio of $^{34}S/^{32}S$ and reported in units per mille (‰). The minor isotopes $^{33}S$ and $^{36}S$ are reported in capital delta notation, where $\Delta^{33}S = \delta^{33}S - ((\delta^{34}S/1000 + 1)^{0.515} - 1) \times 1000‰$ and $\Delta^{36}S = \delta^{36}S - ((\delta^{34}S/1000 + 1)^{1.89} - 1) \times 1000‰$. The $\delta^{34}S$, $\Delta^{33}S$ and $\Delta^{36}S$ data are reported with respect to the Vienna Canyon Diablo Troilite (VCDT) international standard, with analytical errors of ±0.1‰, ±0.01‰ and ±0.2‰ (2σ), respectively, as based on the long-term reproducibility of analyses of the international standard IAEA S-1 (Supplementary Table 5).

**Reporting summary.** Further information on research design is available in the Nature Research Reporting Summary linked to this article.

## Data availability

All data generated or analysed during this study are included with this published article in its Supplementary Information, whereas both the original and compiled data used in Figs. 1–4 are provided as a Source Data file.

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

## Acknowledgements

We thank Nelly Assayag, Amaury Bouyon, Théophile Cocquerez, and Céline Liorzou for assistance in the lab, James Farquhar for discussions and Tim Lyons, Guillaume Paris and Gareth Izon for their highly constructive reviews that improved this manuscript. This project has received funding from the European Union's Horizon 2020 research and innovation programme under the Marie Skłodowska-Curie grant agreement No 708117. This work was also supported by grants from the UnivEarthS Labex Programme at Université Sorbonne Paris Cité (ANR-10-LABX-0023 and ANR-11-IDEX-0005-02), as well as a Region of Brittany Strategy of Attractivity Grant (SAD Project S-GEOBIO, No 0461/14007339/00001041 and 0461/14007349/00001041). P.P. acknowledges support from the São Paulo Research Foundation (FAPESP, grant 2015/16235-2).

## Author contributions

The study was conceived by B.K., S.L., P.S., P.P. and P.C. Samples were provided by P.P. with the assistance of Turee Creek Drilling Project collaborators. B.K. prepared samples with the assistance of S.L. and P.S. B.K. analyzed samples with the assistance of P.C. and C.T. All authors interpreted data. B.K. wrote the paper with contributions from all co-authors.

## Competing interests

The authors declare no competing interests.
