## [Peer Review File · Nature Communications]

Reviewers' comments:

Reviewer #1 (Remarks to the Author):

In the article submitted for review entitled "Oxygen isotope evidence for a transitional marine sulphur cycle at the onset of atmospheric oxygenation", Killingsworth and coauthors present new isotopic data measured from trace barites found in the Kazput formation in western Australia. The authors measured sulfur isotopic compositions characterized by unambiguous positive $\Delta^{33}\text{S}$ values and $\Delta^{36}\text{S}/\Delta^{33}\text{S}$ in agreement with Archean compositions that certainly indicate that the sulfur present in those barites was exposed to the Archean atmosphere. These isotopic compositions are in strict agreement with pyrite data published by some members of the same group in a Nature Communication paper earlier this year (Philippot et al., 2018). The novelty here lies in the $\delta^{18}\text{O}$ values of the sulfates varying between -20 and +2‰ on the SMOW scale, which is much lighter than sulfate in the modern ocean or the Phanerozoic (e.g. Claypool et al., 1980 - only cited in the SOM; Turchyn and Schragg, 2004, not cited) or the Mesoproterozoic barite (Bao et al., 2007 - only cited in the SOM) but very similar to a dataset from barites measured in the Neoproterozoic Kaiyang diamictites (Peng et al., 2013, cited in this submitted article).

The analytical methods used here for S and O isotopes are well established and provide robust results. It would be good to provide standard data as well. The dataset is clearly of interest, yet the fact that most of the geological background (including the age and sulfur isotopic data for pyrites) has been recently published in Nature Communication dampens the impact of the current manuscript. There is no word on the geological context, and almost no paleoenvironmental or lithological descriptions except for one picture and a mention of lamination. This makes the paper feel like an addendum to the previously published Philippot et al. study. In addition, Philippot et al. already suggested oxidative weathering. Furthermore, clues of oxidative weathering have been reported previously in papers not mentioned here (Reinhardt et al., 2009; Stüeken et al., 2012; Gaschnik et al., 2014). For the paper to be of real interest to a broad audience, further work on both the bibliography and a broader and deeper geological context is necessary.

Although the measured oxygen isotopic signature is unarguably of interest the interpretation contains possible shortcomings that would benefit from being addressed for the paper to reach its full impact.

The authors make an interpretation of what is certainly a very unusual signal primarily through the data provided by modern cycles. The S isotopic composition reflects that of older Archean sulfur, weathered and oxidized (on land and the unusually low isotopic composition of oxygen reflects the water source, which the authors conclude is meteoric. Unfortunately the crucial role of metabolisms in biogeochemical cycles is not discussed much, neither is the possibility of different dominating metabolisms 2.3 Ga ago, especially when it comes to the endless diversity of metabolisms, pathways and species involved in the sulfur cycle.

Unlike Peng and collaborators, who simply stated that the "ambient water" where the sulfate formed must have had an usually low $\delta^{18}\text{O}$ value and therefore likely reflected the presence of glacial meltwater (possibly in porewaters), the authors jump in and interpret their dataset as a reflection of oxidative weathering of reduced sulfur on land. Low $\delta^{18}\text{O}$ of sulfate can certainly suggest oxygen coming from continental waters, although a robust demonstration of this would require additional bibliographical references, as relatively recent papers exist on the topic (eg. Calmels et al., 2007; Turchyn et al., 2014- the question is crucial to the demonstration of the authors' thesis yet the only references made are to a paper quoting earlier, less directly relevant, studies and a self-reference). Such $\delta^{18}\text{O}$ signature of sulfate cannot be seen in the modern ocean because this signature is buffered by much higher sulfate concentration.

This point requires further discussion. Do the authors think that the whole Paleoproterozoic oceanic sulfate reservoir has an unusual signature? How would this value be maintained? In other words, is the barite authigenic (therefore all oceanic sulfate has an unusual $\delta^{18}\text{O}$ signature) or precipitated on land and transported to the ocean? I can foresee difficulties with

either interpretation.

The Paleoproterozoic ocean was similarly concentrated in sulfate compared to the Neoproterozoic, possibly even more if we follow the published $[SO_4]$ of Planavsky et al. (2012) around the 2.3-2.1 Ga Magundi excursion, or take into account the evidence for 2.32 Ga sulfate evaporites in the Gordon Lake Formation of the Huronian Supergroup or the Kona Dolomite of the Chocoma Group (e.g. Bekker et al., 2006). The latter suggests that sulfate concentration was high at 2.31 Ga based on the age correlation between the Gordon Lake formation and the Meteorite Bore Member (Philippot et al.; 2018) so prior to the deposition of the Kazput formation. As a result, even if sulfate was lower than today, it was probably not as low as the authors seem to suggest (and higher than in the case of Peng et al.'s study) and if oxygen was scarcer than today, then oceanic MSR might already be quite active and influencing sulfate $\delta^{18}O$ values. If the authors assume an entire oceanic sulfate reservoir being influenced by continental sulfate it seems necessary to provide a numerical quantification of how much "continental" sulfate would be required to overcome the expected "marine" $\delta^{18}O$ signature. If not, can we assume the basin to be, similarly to the Neoproterozoic formation of Peng et al., more easily influenced by continental inputs? Would that be in agreement with the body of literature on the Kazput formation geology?

Overall, the authors don't seem to solve that ambiguity: where did the sulfate ion form and where did the barite precipitate? An intriguing yet undiscussed aspect of this research is the extreme similarity between pyrite and barite isotopic composition, especially the $\delta^{34}S$. If the barite precipitated in the basin from sulfate ions that were formed on land, would it make sense that the S composition of local barite and pyrite overlap perfectly? It seems that in this case, the pyrite would be formed from sulfide resulting of microbial sulfate reduction and would most likely have lower $\delta^{34}S$ than sulfate (as is it the case for most barite-pyrite comparisons, Archean or not). This overlap instead suggests that the barite results from local sulfide oxidation (possibly in porewater?), but then how to explain the $\delta^{18}O$ values? This scenario seems however more likely than sulfate being delivered to the basin as a dissolved species and afterwards either buried as barite or reduced as sulfide with no S isotopic fractionation.

The barite itself could have precipitated on land much earlier than the deposition of the Kazput formation. Had the barite been transported from land, it would be harder to interpret the age and meaning of the signal. Alternatively, could the barite have formed later in situ, through, say, local oxidative fluid circulation, which might carry a very different oxygen isotopic signature? This is another possible explanation of why the barite is so similar to the pyrite (from a S isotope point of view) that remains unexplored.

Given that there is also another significant option for interpreting the data, as oxygen in sulfate can also reflect metabolic pathways, I think that the statement in line 74-75 attempting to put this option aside oversimplifies the actual situation (particularly as there is no reference to support the statement that $\delta^{18}O$ of sulfate uniquely indicates whether the sulfate was formed on land or in the ocean). Continental and oceanic waters do have different oxygen isotopic compositions but sulfate formation can be more complicated. In addition, if the oxygen can be sourced from the water, it can also come from O_2 depending on the pathway. The authors don't acknowledge an abundant body of literature on the topic of oxygen isotopic fractionation during sulfate formation. Gomes and Johnston 2017 provide a review of the influence of the pathways, whether biotic or not, on the oxygen isotopic composition of the resulting sulfates. If, as stated by Killingsworth and collaborators, the oxygen isotopic composition of the newly formed sulfate is not buffered by oceanic sulfate, is there a reason for not assuming, for example, that instead of oxidative weathering, what they observe is biologically mediated anaerobic oxidation of sulfide in a closed system (porewaters, for example)?

L. 70 references would be good

L.73 primarily yes, but possibly significantly from O_2

L. 74-75: this is true only if the involved metabolisms and pathways are known and constrained.

L. 126 A ref. would be good

L. 140: different fractionations for water-sulfate exist depending on the pathway so things are maybe not as straightforward as suggested here.

L. 148 further support: ref?

L. 156 and following: The authors suggest that their data are at odds with Torres et al. I agree, they seem to be but their way of dismissing the Torres et al. paper is not substantiated and seems misleading. They have two arguments: "present day rivers are strongly buffered by atmospheric S inputs, and the year round dry deposition of atmospheric sulfur was also neglected." Present day rivers are not necessarily "strongly buffered" by atmospheric S input, even though they are influenced by them. Such a strong statement, not supported by any reference, could appear as dismissing decades of work on the chemical and isotopic compositions of rivers, especially on sulfur cycle. Moreover, atmospheric S inputs, as well as "urban" runoff, are considered by Torres et al. and corrected for. In addition, is there a reason to expect that dry deposition would be much higher than wet deposition? Do the authors have references of data supporting their claim, in particular for this part of Canada? Given that there is over an order of magnitude in sulfate concentrations for catchments that are more or less right next to each other, through what mechanism(s) do the authors think that dry (or wet, for that matter) deposition would be so spatially heterogeneous?

L. 163: if pyrite burial approached indeed unity, wouldn't the remaining sulfate be nonetheless strongly fractionated? Would that explain that pyrite and barite carry the exact same sulfur signatures? In addition, seawater sulfate concentration in Canfield and Farquhar 2009 is not in agreement with later published scenarios (Planavsky et al., 2012)

L. 183 "oxidative conditions" is vague and a bit tautologic. All sulfate on Earth's surface results from oxidation at some point as sulfur in the mantle or sulfur emitted by volcanoes is not as oxidized as sulfate, and oxidation of course always requires oxidative conditions. But oxidized species, including sulfate, can be found even through the Archean, so oxidative conditions on the surface of the Earth have existed for a long time before the Kazput formation. What the Kazput formation barite signature seems to require is glacial meltwaters and/or specific metabolic pathways.

Figure 1: references in the main text would be better. Turchyn and Schragg , 2004 is missing.
SOM:

fig. S1 is hard to read (especially the spectra)

reference is misspelled (p. 10) and page numbering is wrong towards the end.

Izon's Review of Killingsworth *et al.*, NCOMMS-18-31276-T: "Oxygen isotope evidence for a transitional marine sulphur cycle at the onset of atmospheric oxygen".

Bryan and colleagues,

We are all toying with the same fundamental question(s): what does the S-MIF record mean and how do we translate this into some coherent narrative of atmospheric chemistry(?). That's still not resolved. An alternate approach, however, is interesting, timely and warranted. I expressed this to the editor and I hope this enthusiasm carries through the pages that follow—Sorry! Before you read further, I want to apologise for the number of words that will likely follow. I have an annoying habit of rewriting stuff and I don't want to put my slant on your science. Sometimes, however, we get too close and forget that others cannot read between the lines. I'm a simple human and hope that some my comments, asides and questions help streamline the manuscript and make it more accessible to those of us that are not so smart... I also recommend that you take advantage of the subheadings that *Nature Communications* offers. This will allow you to easily add extra information where necessary to present a balanced appraisal.

Anyway, regardless of the editorial decision (that's their job) please feel free to email or call regarding any of the following. I think this is a solid piece of work that, with some changes, really adds to the developing discussion of the Palaeoproterozoic S-MIF record. I am happy to help if I can, and I look forward to seeing this in print

Regards,

Gareth

Main text:

Line 39–56 I hate to say this, but the first paragraph is a little ropey. Not bad. Just ropey. I do not object to it on a scientific level, I think the issue here is you're trying to crowbar a tonne of information into a very limited space leaving the reader, or at least me, [a little] confused! I think simplicity is the key here and maybe splitting the prose into an extra paragraph. The key point you are trying to get across, I think, is that atmospheric oxygen availability has governed the evolution of the sulphur cycle. Therefore, the mention of oxygen in line 44, for example, is a little premature! The points that I think are worth pulling out are, against an oxygenated backdrop, (1) oxidative weathering makes sulphate → (2) MSR favours ^{32}S → (3) in concert with FeS_2 burial driving seawater heavy → (4) small isotope effect in Cap-delta → (5; NEW PARAGRAPH) Look at MIF = very different... *I think a little simplification and rewrite maybe warranted here.* Is the following more in line with what you are thinking? Take what you want but it will need citations...

Today, oxygen constitutes 21% of Earth's atmospheric gas budget, driving a marine sulphur cycle dominated by oxidative weathering, generating substantial pyrite-derived sulphate fluxes, augmented by the dissolution of evaporites. Sulphate reduction is associated with a large isotope effect, preferentially converting ^{32}S to sulphide where it can be preserved upon reaction with iron [or organic matter], eventually forming pyrite [sulfurized organic matter]. Consequently, global pyrite burial results in a pronounced isotopic offset between the riverine precursor and seawater sulphate, enriching the latter in ^{34}S ... [then small $\Delta^{33}\text{S}$, maybe? – see following comment]

[This might be better as a new paragraph?] By contrast, the Archean and the early Palaeoproterozoic sedimentary S-isotope record reflects a substantially different sulphur cycle, with large magnitude $\Delta^{33}\text{S}$ values ($\Delta^{33}\text{S} \approx \delta^{33}\text{S} - 0.515 \times \delta^{34}\text{S}$), ranging from -4 to 14‰, typifying mass-independent fractionation (MIF) produced from an anoxic atmosphere (Figure 1a). Here, while $p\text{O}_2$ remained below 10^{-5} times the present atmospheric level (PAL), photochemically induced S-MIF resulted in at least two geochemically distinct exit channels that escaped homogenization, with insoluble S_0 and soluble SO_4 thought to carry positive and negative $\Delta^{33}\text{S}$ values, respectively.

Line 45: Caution, this is a scientific objection. Non-zero cap-delta-33 does not mean that its S-MIF. S-MIF has a distinctive magnitude (> 0.4) and a distinctive slope (-0.9 to -1.5?). This makes "near-zero, but, very small, signal of mass independent fractionation" an incorrect statement. Please ensure that this is removed from the amended manuscript.

Line 54–56: I'm not sure that we know that there were only two atmospheric exit channels. I agree that more than one is a requisite, but the upper number is unknown given we don't understand the genesis of geologically analogous S-MIF. The sign of each phase is also not known with certainty – See discussion of Claire et al. (2014). I would deemphasise these points a little. I've tried to give an example in the rewritten text above.

Line 59 and throughout: Can we add fully propagated uncertainties to all the quoted ages. This is important for the rationale that you're presenting. If the chronological constraints are imprecise then we may not need the more complex interpretation.

Line 60: I hate < and > in prose. Can you remove these throughout and use appropriate words? This makes the text a lot easier to follow. This is also true of a lot of chemical symbols that are not necessary. Also, I know that Genming used the term "rapidly" in his abstract but, in fairness, this was on the 1–10 million-year time-scale. This is not rapid compared with atmospheric residence times. Can you add these timescales in parentheses so the reader is informed?

Line 61: Please remove the quotations. I do not think they are needed. This is well known terminology. In time, it may be convincingly demonstrated that the GOE is an oxymoron but, for me, *event* has definite linguistic connotations. Also, "An alternate hypothesis exists," could be simplified to "Alternatively" if you need to lose words(?).

Line 62: I don't think that Gyr has been defined and I could be wrong but I think you should use Ga rather than Gyr. I think it should read "~2.5 billion-years (Ga)", for example. Maybe this is semantics and/or maybe that I'm completely wrong. Moreover, I'm sure the typesetters will resolve the debate but the following website might be useful? https://www.ideo.columbia.edu/~ncb/Selected_Articles_all_files/25_Stratigraphy.6.100.pdf

Line 63: "Could be" is perhaps better written "could have been"?

Line 65: The end of the sentence seems odd and the sentence is also humongous. Perhaps close the sentence after ref 10 and open a new one with something like: "Subsequently, supporting^{11,12} and conflicting¹³ arguments regarding the importance of this crustal memory effect have emerged but a consensus is yet to be reached..."?

Line 68: I don't think you need "within the interval". This could be omitted if words were needed.

Line 70: There's nothing wrong here, it's perhaps just not as punchy as it could be. How about: "Coupled oxygen and sulphur isotope measurements from sedimentary sulphates offer a direct assessment of continental sulphide weathering. Sulphate produced via surficial sulphide oxidation derives its oxygen primarily from the ambient water at the locus of oxidation, experiencing negligible isotopic exchange thereafter¹⁴. Consequently, seawater-derived ¹⁸O-enriched sulphates can be distinguished from their ¹⁸O-depleted continental counterparts..."

Line 74: Question. Is sulphate oxygen exchange immune to modification via biological cycling? For example, via microbial redox? I haven't read the reference you cite but could sulphate availability (or other nutrients) play a part? Maybe some additional clarification would be beneficial to help the non-experts like me.

Line 75: Question. Can you indicate the ranges in the enrichments/depletions?

Line 76: This would flow better with a linking phrase. For example: "It follows, therefore, sulphate..." Also, do you mean deposited or formed? I would the latter because it does not care that its deposited. As soon as it is formed then it should inherit the signal that you describe. I also favour the latter because I would not describe CAS as being deposited. Finally, maybe it's me, but you use the word sulphate a lot. Given the subject of the manuscript this is understandable but do you want to discriminate between solid and dissolved sulphate? Maybe include barite and carbonate associated sulphate in parenthesis?

Line 79–86: Again, these are long sentences that seem to lose their impact. How about "The paucity of 2.5–2.3 Ga evaporites, in concert with the generally low-level of CAS in appropriately aged carbonate successions, has, thus far, precluded this two-pronged isotope approach. Barite, however, is highly insoluble and is much more resistant to diagenetic modification than CAS, providing an alternate and attractive sulphate repository¹⁵".

Line 88: Given that most of the delta denotations have not been defined then I think that the opening part of the sentence would be better written: "Here, we report quadruple sulphur (QSI; $\delta^{34}\text{S}$, $\Delta^{33}\text{S}$, $\Delta^{36}\text{S}$) and oxygen ($\delta^{18}\text{O}$) isotope data from trace barites that were chemically extracted from..." I also think that there is too little description of the materials that you have analysed. This is conspicuously absent from the supplement, also. Figure 2 seems to show you targeted multiple lithologies, yet the text simply says "finely-laminated and fine-

grained” does this describe the sandstones at the base of the succession? Please can you clarify which core you exploited in the manuscript and add some more detailed text. Using subheadings will allow this to fit more smoothly

Line 90: I’m sure Gyr should be Ga.

Line 91: Sentence beginning “It is noted” reads like a kind of afterthought. To make this read more intentional perhaps: “Barite is known to precipitate under sulphate impoverished conditions when barium is concentrated by organic matter¹⁶, a scenario we envisage for the deposition of the Kazput Formation given the extremely low CAS (< 2 ppm) and crust-like whole-rock barium concentrations (< 600 ppm; SOM).” *Specific questions:* (1) Do you want to mention the relatively $\delta^{34}\text{S}$ enriched pyrite as a line of low sulphate availability? (2) I flagged that the Ba concentrations are crust-like above, but I think they are a bit high. Is this consistent with the suggested mechanism? Low sulphate but lots of sulphate weathering? Is this OK?

Line 95: Again, another sentence that could be streamlined for effect. Better as: “Radiometric age constraints (Re-Os and zircon U-Pb¹⁷) from the Turee Creek Group, Western Australia, demonstrate that S-MIF persists in strata younger than $2.31 \pm X \text{ Ga}$, in marked contrast to QSI records from other cratons where S-MIF is apparently lost at $2.33 \pm X \text{ Ga}$ ⁹”? Apparently may be key in this sentence when I get around to writing my manuscript...

Line 98: “Its persistence” is unsupported and should be avoided. This is better written “The persistence of S-MIF”, however, perhaps “The asynchronous demise of S-MIF between cratons is reconciled with a crustal memory effect and differential recycling in different basins; however, the notion of asynchrony is based on poorly developed chronostratigraphic frameworks and remains equivocal” is punchier? Worth noting that cratons is perhaps not the best expression given Vaalbara... Is this, in fact, a basinal phenomenon?

Line 101: Please change “with $\Delta^{33}\text{S}$ from” to “featuring $\Delta^{33}\text{S}$ values from”. You can also delete “are further characterized by low” it’s subjective and not necessary. Another observation, the prose is a hybrid between American and English, check for consistency and remove all the unnecessary Zs from the prose and the figures. After all, *Nature* is British.

Line 103: “The $\Delta^{33}\text{S}$ data”, should read “These $\Delta^{33}\text{S}$ data”. Remember, data are always plural. How this sentence is written speaks to the most special data measured over the last 2.5 billion-years, I think you mean that “These $\Delta^{33}\text{S}$ data constitute the highest values reported from sedimentary sulphates younger than 2.5 Ga, while the $\delta^{18}\text{O}$...”.

Line 107: Again, there’s nothing wrong, but the following makes it more easy to read: “The exceptionally low $\delta^{18}\text{O}$ values preserved in the barite from the Kazput Formation indicates that the precursor sulphate did not have a typical marine origin.” This is definitely an optional edit.

Line 108: This is not well explained; how do your $\delta^{18}\text{O}$ data rule out sulphide oxidation? There is only a figure and no documentation of what you’ve done besides a figure caption. You should expand on this. Perhaps adding something like: “Oxygen isotope measurements of sulphate sourced from laboratory water, as well as from artificial pyrite oxidation experiments, preclude laboratory-induced oxidation during handling, in turn necessitating a palaeoenvironmental interpretation.” in the main text?

Line 109: Again a few tweaks could clarify. “The temporal compilation of sedimentary sulphate $\delta^{18}\text{O}$ data demonstrates that that $\delta^{18}\text{O}$ values greater than 7‰ typify minerals that derive their oxygen from seawater (Figure 1b). Sulphate with $\delta^{18}\text{O}$ values below -15‰, however, ...”

Line 113: I am not sure this makes sense but I do not know enough to correct it. Do you mean from “high altitude, high latitude or glacial settings”? How it was written, I’m not sure what the “or likely” meant, I think that’s why I’m confused. Additionally, the following sentence is not straight forward, a reiteration is: “The lowest $\delta^{18}\text{O}$ value reported from the sedimentary record (-20.3‰) was obtained from a glacial diamictite deposited at 635 Ma during a Neoproterozoic Snowball Earth event.”

Line 113: Should “formation” be capitalised?

Line 117: Between the commas beginning after 500m, swap the text with: “and the available age constraints (2.31–2.21 Ga^{12,19}) demonstrate deposition was potentially synchronous with the glacial strata preserved within the Huronian Supergroup, Canada²⁰,”

Line 119: This is best clarified as “...was the oxygen source for the barites preserved in the Kazput Formation”.

Line 121: Again, some ‘word-smithing’ would make this snappier. See: “Discriminating between temperature-derived signals and true seawater $\delta^{18}\text{O}$ variability remains the principle challenge for interpreting ancient sedimentary $\delta^{18}\text{O}$ records. For example, Precambrian oceans are thought to be warmer than today with temperature estimates at $60 \pm 20^\circ\text{C}$, while carbonate-based seawater $\delta^{18}\text{O}$ estimates are as low as -10% ²¹. The average $\delta^{18}\text{O}$ value of the carbonates within the Kazput Formation is 18.4% (*versus* VSMOW)²², which, despite being typical for Siderian carbonates, is 10% lower than present-day carbonates (Figure 1b). Thus, during the deposition of the Kazput Formation, the $\delta^{18}\text{O}$ of seawater may have been 10% lower than its contemporary value (0%). Except for the extreme range of $\delta^{18}\text{O}$ values associated with the Neoproterozoic Snowball Earth events, the $\delta^{18}\text{O}$ values of sulphates fall between 7 and 18% regardless of whether they were the product of relatively cold contemporary oceans or their warmer ancient counterparts (Figure 1b).

Line 134: The previous comment is subjective, grounded on my personal preferences. In fairness, it doesn’t need to be implemented if you disagree. I do feel however that your data is not sufficiently introduced and the prose in the second half is difficult to follow! For example, you refer to “negative barite $\delta^{18}\text{O}$ values” but this is the first time the data has been introduced. I feel that the data should more comprehensively introduced before they are discussed. Thus, the second half of this paragraph could use a rework. Here’s my attempt: “Barite isolated from the Kazput Formation is depleted in ^{18}O , with $\delta^{18}\text{O}$ values averaging -11.0 ± 10.8 (2σ). Irrespective of the uncertainties surrounding Precambrian seawater temperatures and oxygen isotope compositions, these distinctively negative $\delta^{18}\text{O}$ values require a distinct and non-seawater-derived oxygen source. Sulphate produced during pyrite oxidation is enriched in ^{18}O by 10% relative to the parent water²³, suggesting that the $\delta^{18}\text{O}$ of the initial source waters ranged from -8 to -30% . It remains to be resolved whether these barites sourced their oxygen from ^{18}O -depleted Palaeoproterozoic seawater ($\sim 10\%$ ²¹), glacial meltwater (-34% ²¹) or a mixture of the two. Regardless, both cases require a meteoric water source, implying a continental origin for this sulphate.” These data are so light given the record. I would have thought this would have warranted more discussion. This seems like a significant finding

Line 148: I ran out of steam a little here. There’s editorial changes that will help here but perhaps revisions to the structure and content is more pressing. This paragraph is very long and I have several questions/reservations concerning under developed claims that will need to be fortified as you move forward. These are:

1. **Line 155:** Why does the positive $\Delta^{33}\text{S}$ values indicate the precursor sulphate was derived from 2.5 Ga rocks? It looks like equal magnitude rocks are present in younger rocks. How do you prelude that this is not derived from atmospheric chemistry with the $\Delta^{33}\text{S}$ values derived from the atmosphere with mixing from terrestrially-derived ^{18}O depleted sulphate?
2. **Line 158:** Your results being “at odds” seem like an overly strong statement that are introduced too early. I feel that more complete description of the S data is warranted first. Maybe it is not applicable to all Archean crust? It is also not clear what you’re trying to say in lines 158–160. Perhaps rewrite this to make the point explicitly clear and reorder. Basinal variability?
3. **Line 163:** How do we know all the sulphate was being removed as pyrite and there was no re-oxidation? In your record, at least, you have trace barite demonstrating that pyrite burial was non-quantitative. Are there more up-to-date references. Perhaps some iron speciation data? Thinking about it the muted $\delta^{34}\text{S}$ data could be used as an argument but this contrasts with the south African record. Do you have thoughts on this difference? I would expect light pyrite if the sulphate if pyrite weathering was pervasive?
4. **Line 172:** Are the two ARA lines in panels a and b -0.9 and -1.5 ? If so, it seems that the data plots on the steeper of the two. This is different from most Archean rocks and to my knowledge my 2017 dataset is the best for illustrating these discrete slope changes? What do you think, are there some of these -1.5 in the catchment?
5. **Line 175:** “Section” should be “succession”.
6. **Line 175:** What is the significance of the 0.5% offset you describe here?

Line 181: “from our sulphates can be omitted”.

Line 183: “it is notable that” can also be pruned. I agree with the sentence requiring changes but perhaps expand and refer to the figure. I can’t help but think Fig 2 is beautiful yet under used. I would love to see this described more thoroughly and the implications discussed. The interesting thing here is that you seem to have systematic changes in the $\Delta^{33}\text{S}$ and $\Delta^{36}\text{S}$ data. What do you think this means? How does the slope evolve?

Line 199: Would you expect to see a correction in Fig 3c?

Line 200: What about ground-level oxygenation a la Lalonde? Can there be oxygen production fuelling weathering beneath an anoxic atmosphere? Can you preclude this with your data? If not, then I think you should be more balanced in your arguments.

Line 201: My understanding is that the dataset features relatively few negative values and thus this could represent a bias within the record. Do you agree?

Figure 1: Is TCDP and TC defined? American English in the legends. Green arrows in 1a are misleading please alter.

Figure 2: Is nice there's trends in this data could be interpreted(?). Also, there seems to be a strong dependence on lithology. Is this true and important? Grey lines?

Figure 1: The caption is confusing move the definition of the lines upwards and the definitions down. BSR should be MSR as its not just bacteria check throughout. It would be nice to acknowledge the 1.5 slope was demonstrated convincingly in Izon et al., 2017. I'm not sure I follow the significance of the dashed lines. Please clarify.

Line 288 on: Not checked but a quick scan reveals errors so please check.

Supplementary material: The way the supplement has been written is sometimes confusing. Additionally, the structure doesn't follow as well as it might. I have *attempted* to reword and shorten the text to make it flow better. I have also cut some of the descriptive text that people that are active in this field are intimately familiar with. These cuts will allow you to expand on the bits that are missing. In fairness, I don't expect the authors to take these edits verbatim unless, of course, they wish to do so. This was intended more as more of a guide. Anyway, I have appended the text in a track changes format; however, no doubt, that will get lost in the review system, so please encourage the authors to drop me an email if they feel this would be of benefit.

My specific questions or issues that are perhaps worth clarifying are:

1. The first section (CAS Extraction) seemed strange and repetitive given that it didn't work. I tried to improve this.
2. Can you clarify the strength of the acid you used to liberate the CAS and decarbonate the samples?
3. Can you add the concentrations of the CAS and blanks so others can potentially benefit from your efforts?
4. I find that the expression of your delta ratios is weird. Can you check that I've not messed them up? I try to follow the Coplen paper.
5. Can you specify how many standards you measured to get your precision. Also, can you include the various ratios that you obtained relative to the relevant international standard. I don't doubt your data is good but reporting standards allows us all to see where we're all at.
6. The data sources paragraph was also hard to read. I've tried to amend. Please consider these changes.
7. At least in my image the quality of the EDS images in Figure S1 was *bad*. I couldn't make out what the spectral peaks are! Can these be resized or tweaked to make them look better? Any change the French can be made English? The latter isn't a big deal...
8. Figure S2. Perhaps this doesn't need to be in the MS or clarified but some random Qs: Why is there no data from this sample? No barite? Seems odd to include a section from which there is no corresponding isotope data. What are the textures in the oldest third of the slide? Is this slumping? Is that pyrite crossing the depositional fabric, what stage do you think that is does this have any effect on your interpretation? Is a section from which there's data more appropriate?
9. Can you make the tables easy to use? I know the publishers make this difficult but if you can get an excel file at a supplement that would be awesome. We should push publishers for this!! I also plotted some of the data and the decimals in the averages were slightly out. This could be a rounding error but worth checking.
10. Why did you do it like that? I would have done the weight loss via decarbonation, which itself is not perfect. I'd ask you to note in the caption these are top-end estimates. Also, there's no mention in the methods of the SEM nor the carbonate abundance. Perhaps a few lines for completion?

11. There is no in-depth discussion of the sample cores, the regional geology, where they are archived *et cetera*. This is necessary background, especially the age constraints, and could be added up-front in brief or here in more detail. A map is useful, also.
12. Please check the references. There's a few issues with their formatting, especially with weird characters that are likely issues with super- and sub-scripts

Reworked Supplementary Information:

CAS Extraction:

Carbonate-associated sulphate (CAS) extractions were attempted on samples from drill core 3 from the Turee Creek Drilling Project (TCDP) at the Laboratoire Géosciences Océan at IUEM, Plouzané, France, following a CAS extraction sequence initially verified by Wotte et al.¹ Here, crushed and powdered carbonate samples were leached in triplicate using a 10% NaCl solution, removing any soluble sulphate phases. Dissolution of the carbonate-component using ?? then liberated the CAS, which was then precipitated as BaSO₄ upon addition of a supersaturated BaCl₂ solution. Low CAS yields (<X ppm) from the Kazput samples, in concert with relatively high pyrite abundances (>X ppm or Wt.%), could have compromised the isotopic integrity of the CAS via laboratory-induced pyrite oxidation. Consequently, to test this hypothesis, we modified the CAS extraction protocol, using a reducing agent, hydroxylamine hydrochloride, to inhibit pyrite oxidation during CAS extraction. Unfortunately, the hydroxylamine hydrochloride itself was found to contain trace-levels of sulphate that were sufficiently high (Xppm) to dominate the sulphate recovered as BaSO₄. Accordingly, it was concluded that the CAS content of these samples was too low to enable accurate and precise bulk S- and O-isotope measurements and CAS extractions were not pursued further.

Barite extraction:

Barite extractions were performed on an ~ 100g aliquot of homogenised drill core using a modified barite purification technique (the DDARP method²) that was originally designed to purify sulphate for triple oxygen isotope measurements. Again, the barite extractions were completed at the Laboratoire Géosciences Océan at IUEM. Briefly, drill core samples were manually crushed in a tungsten-carbide piston chamber before powdering in an agate ring and puck mill. Sample powders were decarbonated in HCl-acidified solution, rinsed in distilled water, and then stirred for 3 days in a mixed 0.05 M Diethylenetriaminepentaacetic acid (DTPA) and 1 M NaOH solution to dissolve barium sulphate. The supernatant was then separated from the sample residue by vacuum filtration. After acidification (pH < 2) with HCl, the barite-housed sulphate was precipitated as BaSO₄ via the addition of a supersaturated BaCl₂ solution. Co-precipitation of silicates dissolved in the DTPA solution renders the precipitated BaSO₄ impure, requiring an additional purification step. Here, residual DTPA is removed via repeated rinsing (n=3) with distilled water followed by centrifugation and decanting of the supernatant. After heating the precipitate at 70 °C overnight with 2 M NaOH, the sample was again centrifuged and the supernatant decanted. The BaSO₄ was once again redissolved via overnight agitation in a mixed 0.05 M DTPA and 1 M NaOH solution and precipitated as BaSO₄ via the addition of acidified BaCl₂ solution. The final precipitate was washed in triplicate and dried at 70 °C overnight ready for isotope analysis.

Oxygen isotope measurements:

The oxygen isotope composition of the purified barite samples were measured in duplicate using an Elemental vario PYRO cube elemental analyser interfaced with an Isoprime 100 mass spectrometer in continuous flow mode at the University of Burgundy in Dijon, France. Oxygen isotope data are expressed in delta notation, $\delta \equiv R_{\text{sample}}/R_{\text{standard}} - 1$, where R is the mole ratio of ¹⁸O/¹⁶O and reported in per mille (‰). The $\delta^{18}\text{O}$ data are reported with respect to the international standard Vienna Standard Mean Ocean Water (VSMOW). Analytical errors are $\pm 0.4\text{‰}$ (2 σ) based on replicate analyses (n = X) of the international barite standard NBS-127, which was used to bracket unknowns.

Quadruple sulphur isotope ($\delta^{34}\text{S}$, $\Delta^{33}\text{S}$, & $\Delta^{36}\text{S}$) measurements

To determine for quadruple sulphur isotope combustion of the purified barites, an additional wet chemistry stage was required. Here, at the Institut de Physique du Globe de Paris, Paris, France a 3 mg aliquot of purified barite was converted to Ag₂S via sub-boiling distillation with HCl, HI, and H₃PO₂³⁻⁵), reducing the sulphur to H₂S,

which was ultimately trapped upon reaction with silver nitrate. The precipitated Ag_2S precipitate was washed three times in distilled water and dried in an oven overnight ready for fluorination and sulphur isotope analysis. Approximately 3mg of Ag_2S was converted SF_6 gas via overnight reaction under an excess of F_2 gas at 250 °C in nickel bombs. Sulphur Hexafluoride was then purified via cryogenic trapping and separation by gas chromatography before introduction to the mass spectrometer—a Thermo Finnigan MAT 253 running in dual inlet mode. Conventional delta notation is used to report ^{34}S data, where $\delta \equiv R_{\text{sample}}/R_{\text{standard}} - 1$, where R is the mole ratio of $^{34}\text{S}/^{32}\text{S}$ and reported in units per mille ‰. Minor isotope data, ^{33}S and ^{36}S are reported in capital delta notation, where $\Delta^{33}\text{S} = \delta^{33}\text{S} - ((\delta^{34}\text{S}/1000 + 1)^{0.515} - 1)$ and $\Delta^{36}\text{S} = \delta^{36}\text{S} - ((\delta^{34}\text{S}/1000 + 1)^{1.89} - 1)$. The $\delta^{34}\text{S}$, $\Delta^{33}\text{S}$, & $\Delta^{36}\text{S}$ data are reported with respect to the Vienna Canyon Diablo Troilite (VCDT) international standard, with analytical errors of $\pm 0.1\text{‰}$, $\pm 0.01\text{‰}$ and $\pm 0.2\text{‰}$ (2σ), respectively, as estimated via replicate analyses of the international standard IAEA S-1 ($n = X$).

Data Sources in Figure 1

The temporal record of $\Delta^{33}\text{S}$ (Figure 1a) is predominantly derived from Havig et al. (2017)⁶. Data sourced from non-sedimentary processes (i.e., hydrothermal⁷ and thermochemical sulphate reduction⁸) have been removed and the $\Delta^{33}\text{S}$ values originally from Ref. 9 have been corrected after inverted substitution for $\delta^{33}\text{S}$ data⁶. Data from the Boolgeeda formation¹⁰ have been adjusted for the new age constraints and correlations presented by Philippot et al. (2018)¹¹. The temporal compilation of sedimentary $\delta^{18}\text{O}$ data (Figure 1b) is derived from chert, carbonate, and shale, which was published by Bindeman et al. (2016)¹² and augmented with additional sulphate $\delta^{18}\text{O}$ data¹³⁻¹⁶.

Figure S1. Scanning electron micrographs and EDS spectra of barite isolated from sample CAS-9 (96.53 m). To concentrate barite for imaging, the silicate fraction was removed from the decarbonated sample residue using hydrofluoric acid and the residual material was dried prior to mounting.

Figure S2. A scanned image of petrographic thin section from 107.8 m depth in the Kazput Formation, TCDP core 3. The approximate field of view is 20 x 40 mm. The direction of younging is parallel to the page with the youngest material at the top. The fine-grained and finely-laminated texture seen in this thin section typifies the central shaley carbonate discussed in the main text and the TCDP drill core 3 as a whole. The black rectangular shape in the centre-left is a euhedral pyrite.

Figure S3. The $\delta^{18}\text{O}$ of dissolved sulphate in tap water, sulphate derived from the deliberate oxidation of pyrite in distilled laboratory water, and the average for barium sulphates extracted from TCDP core 3. The sulphate in tap water was precipitated from municipal water in the laboratory in Plouzané, France. The sulphate derived from the oxidation of pyrite was generated by stirring crushed and powdered pyrite for >1 day in laboratory distilled water, using the same distilled water source that was used during the barite extractions.

[Table 1 OK]

Table S2 (% carbonate). Calculated carbonate abundances for TCDP core 3. These estimates combine CaO oxide abundances (Wt.%) and loss on ignition (assumed to entirely represent CO_2) data to approximate carbonate abundance. Initial X measurements were performed by ACTLABS using alkaline fusion total digest of sample powders.

Reviewer #3 (Remarks to the Author):

I very much appreciate the goals of this paper and the approach taken. This paper is absolutely appropriate for publication in Nat. Comm. The potential impact will be high. The motivations and methods are clever and robust. I hope it will be accepted. But there are things to consider first.

- (1) The choice of samples from the Turee Creek basin means that there is a wealth of previous work that elevates the chronological control, environmental context, and broad reader interest. That said, we aren't given much environmental detail, at least not in the primary text. The connection to glaciation is only loosely established (lines 116-119): "the Kazput formation overlies the Meteorite Bore Member glacial diamictite by ~500 m, (and therefore) it is plausible that glacial meltwater was an oxygen source for Kazput barite." Does 500 m of stratigraphic separation really establish an environmental linkage to glaciation? I'm pressing here because the potential role of glaciation is a common theme throughout the paper, yet the purported close temporal and spatial relationships are at best anecdotal. More generally, a better environmental context (links the oceans, glacial runoff, deposition under restricted shelf/continental conditions, etc.) will give the reader better access to evaluating the likelihood of local versus global controls and signals.
- (2) Line 94: We are only told that trace barite is present and that this observation is consistent with low sulfate levels (in the local setting? oceans?). I was hoping for more discussion about the barite (in the primary text, given the importance). For example, can we be sure that the barite reflects syndepositional, low temperature conditions? Might hydrothermal process have played a role, which would impact the oxygen isotope data? Could later remobilization/overprint scenarios have been important? Simply put, the sedimentological context of the barite and specifically arguments supporting its primary, low-T origin are important but missing.
- (3) The basic premise of the paper, which I like very much, is that combined S and O oxygen isotope approaches allow for recognition of oxidative weathering of continental sulfides, with the S data capturing recycling of older, Archean sulfides bearing a MIF-S signal. In other words, the S signal is inherited, and oxygen's is not—capturing instead the meteoric waters involved during oxidation of the continents at the time of deposition of the barite—presumably beneath an atmosphere that, by then, was too oxygenated to yield a preserved MIF signal. The possibility of recycled/inherited MIF-S signals in younger rocks deposited beneath a relatively O₂-rich atmosphere is important. Reinhard et al., as the authors note, argued for such processes and suggested that MIF-S recycling could complicate our interpretations of those isotope records. Torres et al. challenged that assertion, and now Killingsworth et al. shed new light supporting the possibility of recycling/inheritance as captured at least locally. This is important.

However, two things come to mind: (i) This overarching story and the related implications could be spelled out more clearly in the paper. Now, only insiders will truly grasp the potential importance of the paper in terms of our ability to constrain the timing and pattern of initial oxygenation, etc. (ii) I am not convinced that the authors have made a case for their data requiring recycling of older sulfur beneath a relatively well oxygenated (post-GOE) atmosphere.

Evidence for oxidative weathering of continental sulfides in combination with a MIF-S signal could, as the authors suggest, reflect recycling of older S retaining its inherited MIF behavior. On the other hand, the combination could rather reflect sulfide oxidation under very low O₂ conditions—low enough to preserve syndepositional MIF-S processes in the atmosphere. This second possibility was suggested by Reinhard et al. in 2009 in their take on the Mt. McRae whiff-of-oxygen story. Critically, the Reinhard 2009 paper argued that sulfate was sourced to the oceans by weathering even under very low, MIF-sustaining (whiff) oxygen conditions. If this latter model is correct, Killingsworth's notion of oxygen isotope signals reflecting continental weathering could be an Archean, pre-GOE feature as well. They just don't have the data to argue one way or the other. It is certainly possible that oxidative weathering was an early feature only enhanced at the GOE and sulfate was sourced to the ocean, even before the GOE, by oxidative weathering in addition to atmospheric sources.

We suffer now because the authors don't have earlier data, and we don't really know the possible magnitude/impact of the sulfate-oxygen signal (that is, possible expressions in the global ocean). The carbonate data certainly don't support the idea of very low $\delta^{18}\text{O}_{\text{H}_2\text{O}}$ for global seawater. And I struggle to imagine large amounts of sulfate forming in those water. (Are exchange reactions likely to have dominated?) More could be said about what was possible/likely isotopically for the early ocean and what the barite data most likely record and why we should care.

I am rambling. My point, though, is that we need more discussion on the likely local versus global signals and, most importantly, what these data really tell us about the GOE in comparison to likely sulfate sources and their isotopic signals during the earlier Archean. Arguments like those in this paper about presumed temporal shifts despite a lack of earlier data (a pre-shift baseline) are problematic.

- (4) Line 163: "Sulphide generated at the sediment-water interface would have experienced little to no re-oxidation in poorly oxygenated bottom waters." I agree, and the following speaks to this quantitatively: Oxidative sulfide dissolution on the early Earth, CT Reinhard, SV Lalonde, TW Lyons, *Chemical Geology* 362, 44-55 (2013).
- (5) Line 33: In light of all these comments, how do the data convincingly "indicate a transitional oceanic sulfur cycle dominated by oxidative weathering in the aftermath of atmospheric oxygenation?" I suspect this is true, but we need a more convincing and/or balanced perspective aimed at the key implications and arguments for a transition—with a clearer statement on the major conclusions relative to previous interpretations, the data before and after the GOE, etc.

This is a wonderful data set. Another round of revision will likely elevate the impact. I hope to see it in print soon.

Tim Lyons, UC-Riverside

Point by point response to reviews of manuscript NCOMMS-18-31276A (revised version of NCOMMS-18-31276-T) titled "Constraining the rise of oxygen with oxygen isotopes"

Original reviews are in black, responses are in blue, quoted manuscript sections are in blue and preceded by line numbers (L)

REVIEW 1

Reviewers' comments:

Reviewer #1 (Remarks to the Author):

In the article submitted for review entitled “Oxygen isotope evidence for a transitional marine sulphur cycle at the onset of atmospheric oxygenation”, Killingsworth and coauthors present new isotopic data measured from trace barites found in the Kazput formation in western Australia.

The authors measured sulfur isotopic compositions characterized by unambiguous positive $\Delta^{33}\text{S}$ values and $\Delta^{36}\text{S}/\Delta^{33}\text{S}$ in agreement with Archean compositions that certainly indicate that the sulfur present in those barites was exposed to the Archean atmosphere. These isotopic compositions are in strict agreement with pyrite data published by some members of the same group in a Nature Communication paper earlier this year (Philippot et al., 2018). The novelty here lies in the $\delta^{18}\text{O}$ values of the sulfates varying between -20 and +2‰ on the SMOW scale, which is much lighter than sulfate in the modern ocean or the Phanerozoic (e.g. Claypool et al., 1980 - only cited in the SOM; Turchyn and Schragg, 2004, not cited) or the Mesoarchean barite (Bao et al., 2007 - only cited in the SOM) but very similar to a dataset from barites measured in the Neoproterozoic Kaiyang diamictites (Peng et al., 2013, cited in this submitted article). . The analytical methods used here for S and O isotopes are well established and provide robust results.

It would be good to provide standard data as well.

Standard data is now provided in Tables 3 and 4 of the Supplementary Information

The dataset is clearly of interest, yet the fact that most of the geological background (including the age and sulfur isotopic data for pyrites) has been recently published in Nature Communication dampens the impact of the current manuscript. There is no word on the geological context, and almost no paleoenvironmental or lithological descriptions except for one picture and a mention of lamination. This makes the paper feel like an addendum to the previously published Philippot et al. study.

The geological context is now much expanded.

First, the "big picture" points about the geological context are introduced:

L108-117: The 2.45 Ga{Trendall, 2004 #3405;Caquineau, 2018 #3110} to 2.2 Ga{Müller, 2005 #3407} Turee Creek Group sedimentary succession from W. Australia is ideal for seeking barite records to characterize the surface sulphur cycle around the time of the GOE. Multiple sulphur isotope records ($\delta^{34}\text{S}$, $\Delta^{33}\text{S}$, $\Delta^{36}\text{S}$) in early diagenetic pyrites from the lower Kazput Formation, in drill core 3 from the Turee Creek Drilling Project (TCDP3) (Supplementary Figs. 1 and 2), have shown the persistence of S-MIF in sulphides that are younger than expected (<2.31 Ga) for sediments deposited under a globally oxygenated atmosphere{Philippot, 2018 #3192}. Furthermore, isotope compositions of Kazput Formation carbonates are $\delta^{13}\text{C} = -2\text{‰}$ to $+1.5\text{‰}$ and $\delta^{18}\text{O} = -16.63\text{‰}$ to -8.13‰ (VPDB), suggesting that diagenesis has not erased their record of primary seawater chemistry{Martindale, 2015 #2448}.

More detailed geologic background is then given in the discussion:

L133-148: The barite of the Kazput Formation was studied within an overall-shallowing succession of the Turee Creek Group that shows evidence for increasing atmospheric oxygenation in its Mn-enriched units and decreasing iron content{Van Kranendonk, 2015 #2451}. The lower Kazput contains finely laminated mud- and silt-stones before shallowing into laterally variable oolitic and stromatolitic carbonates with deltaic influence{Martindale, 2015 #2448}. Barlow and authors{Barlow, 2016 #2879} previously conducted detailed lithostratigraphic analysis of the ~350 m Kazput Formation. They described the diversity and prevalence of stromatolites, designating 5 facies associations according to lithologic and microbial morphologic variations. The lowermost Kazput carbonate from the TCDP3 core used in our study is consistent with their facies association A. Geologic maps with TCDP core locations (Supplementary Fig. 1), TCDP3 core photographs (Supplementary Fig. 2), and thin section micrographs and XRF scans (Supplementary Figs. 5 & 6) provide additional details. The lower Kazput carbonate in TCDP3 grades into carbonated siltstones above and below (Fig. 2), with fine cross-laminations observed throughout. Although barite was observed in rock digestion residues (Supplementary Figs. 3 & 4), it is not easily found in place in the rock, with grains of less than 3 μm in size only detected by SEM scan of a full thin section.

As pointed out above in the main text, new supplemental figures have been added of geologic maps (Supp. Fig. 1), core photos (Supp. Fig. 2), and new photomicrographs and XRF scans of select elements (Supp. Fig. 5), that augment the previous thin section figure (Supp. Fig. 6), and SEM detection of barites (Supp. Figs. 3 & 4)

In addition, Philippot et al. already suggested oxidative weathering. Furthermore, clues of oxidative weathering have been reported previously in papers not mentioned here (Reinhardt et al., 2009; Stüeken et al., 2012; Gaschnik et al., 2014). For the paper to be of real interest to a broad audience, further work on both the bibliography and a broader and deeper geological context is necessary.

As above, the geologic context is much expanded now. Further, referenced discussion and material has been added about oxidative weathering, where an important change is

made to Fig. 1 that now includes, in a new added panel, the time series modeled sulphur weathering fluxes (in Fig. 1c) from the above-mentioned Stueken et al., 2012 paper.

Indeed, as Reviewer 1 points out, oxidative weathering has been discussed by others. What we add here, with the application of oxygen isotopes in sulphate, is a way to interrogate this further in the sulphur cycle instead of only relying on the sulphide record, which is, after all, the reduced phase of sulphur. It makes sense to now provide more information about oxidative sulphur cycling with a detailed look at the oxidized sulphur species, sulphate, and further, to look at the oxidant in that species, oxygen, to better reveal oxidation in the early Earth.

The last point of the paper speaks directly to the above concerns..

L391-394: It has been suggested that SO_4^{2-} flux to the oceans increased in advance of the GOE due to oxygen production and its consumption by sulphide weathering on land {Stüeken, 2012 #513; Lalonde, 2015 #522} (Fig. 1c). Through their oxygen isotope records, it may be possible to constrain this early flux of SO_4^{2-} in the Neoproterozoic.

A deeper geologic context is also now provided in the discussion of the barite $\delta^{18}\text{O}$ signals with respect to Paleoproterozoic glaciations:

L281-301: Kazput barite $\delta^{18}\text{O}$, which extend to -19.5‰ , are among the lowest values in the sedimentary record, which points to oxidation in glacial meltwater rather than just meteoric water. This is consistent with the lowest $\delta^{18}\text{O}$ reported from the sedimentary record reaching -20.3‰ in sulphates recovered from a Neoproterozoic snowball Earth glacial diamictite deposited at 635 Myr {Peng, 2013 #2018} (Fig. 1b). One other case of such low $\delta^{18}\text{O}$ in sulphate from the sedimentary record is observed in more recent terrestrial deposits within the Arctic Circle that reach down to -19.7‰ (Fig. 1b). In either case, sulphate $\delta^{18}\text{O}$ approaching -20‰ appears to require snow or ice meltwater as an oxidant. The age of the Kazput Formation falls between 2.31 to 2.21 Ga {Philippot, 2018 #3192}, placing it firmly within a period of global glaciation, with up to four glacial episodes between 2.45 and 2.2 Ga {Gumsley, 2017 #2809}. Three Paleoproterozoic glacial episodes are recognized in W. Australia, with minimum detrital zircon U-Pb ages of $<2.460 \pm 0.009$ Ga for the Boolgeeda glacial diamictite and $<2.340 \pm 0.022$ Ga for the lower Meteorite Bore Member diamictite {Caquineau, 2018 #3110}, with the latter further constrained by a Re-Os age of 2312.7 ± 5.6 Ma from pyrites within core TCDP2 {Philippot, 2018 #3192}. The youngest Paleoproterozoic glaciation, only recognized in S. Africa thus far, is constrained between 2250-2240 Ma {Schröder, 2016 #3383} to 2256 ± 6 Ma {Rasmussen, 2013 #508}. It might be possible for snowmelt water from high altitude mountainous regions to contribute to ^{18}O -depleted signatures in SO_4^{2-} , but we believe that this is unlikely due to the significant flux of such sulphate, and its preservation, that is required by the Kazput barite record. Considering the time window of deposition for the Kazput Formation, its barite $\delta^{18}\text{O}$ might be linked to meltwaters from the final, ca. 2.25 Ga, Paleoproterozoic glaciation but this will require future study.

Broader geological and geochemical context is now provided in the comparison between W. Australia and S. Africa sulphur cycling during the GOE interval, which is accompanied by the newly added Figure 4:

L318-344: Sulphur isotope distributions compared between S. Africa and W. Australia (Fig. 4) reveal preservational controls on S-MIF during the GOE interval, ca. 2.45 to 2.2 Ga. The depositional settings were intracratonic basins for both S. Africa {Bekker, 2004 #268} and W. Australia {Van Kranendonk, 2015 #2451}, near the equator {Gumsley, 2017 #2809}. Despite similarities between their basins, S. Africa and W. Australia sulphur isotope records are distinct for the GOE period. In W. Australia, including Kazput barite, the spread of sulphur $\delta^{34}\text{S}$ data are mostly distributed around one mode, slightly above 0‰. In contrast, data from S. Africa show a bimodal ^{34}S distribution more typical of open-system MSR, with isotopically lighter sulphur preferentially incorporated into sulphide with negative $\delta^{34}\text{S}$ and sulphate from CAS records with average values around +25‰. These different behaviors are manifested in their respective $\Delta^{33}\text{S}$ distributions, where the range from each basin is similar, but data from S. Africa indicate a strong source of mass-dependent sulphur ($\Delta^{33}\text{S}$ near 0‰) compared to two sources of sulphur with S-MIF at around +0.9‰ and +1.6‰ in W. Australia. The Kazput barite $\delta^{34}\text{S}$ - $\Delta^{33}\text{S}$ relationship of ^{34}S -enrichment versus the ARA (Fig. 3b) is matched in pyrite {Philippot, 2018 #3192}. Such enrichment in S-MIF-bearing pyrite has previously been observed in microbialitic carbonates where carbonate precipitation effectively sealed off the sediment from the water column, promoting closed-system MSR {Ono, 2009 #3388}. Stromatolites and carbonate are indeed prevalent throughout the Turee Creek Group, and especially in the Kazput Formation carbonates. The $\delta^{18}\text{O}$ compositions of CAS from S. Africa are yet to be measured, but it is likely that they are relatively enriched in ^{18}O as compared to the Kazput barite. As discussed here, the open system MSR suggested by S isotope evidence from S. Africa would also exert strong control on the $\delta^{18}\text{O}$ of SO_4^{2-} . In order to preserve the persistent S-MIF signals recorded throughout the Turee Creek Group succession, as compared to S. Africa, in W. Australia it seems that closed system MSR, within the sediment, was required. Further, we suggest that low global marine sulphate concentrations and an intracratonic basin setting made the capture of weathering input fluxes, and their SO_4^{2-} isotope compositions, possible in the Turee Creek Basin.

Although the measured oxygen isotopic signature is unarguably of interest the interpretation contains possible shortcomings that would benefit from being addressed for the paper to reach its full impact.

The authors make an interpretation of what is certainly a very unusual signal primarily through the data provided by modern cycles. The S isotopic composition reflects that of older Archean sulfur, weathered and oxidized (on land and the unusually low isotopic composition of oxygen reflects the water source, which the authors conclude is meteoric.

Point noted about our interpretations being made in the context of modern cycles. We argue that establishing a context with respect to modern cycles is a common trope but useful because 1) we know more about modern cycles, and 2) it is better to root the discussion in what is known, first, before venturing into speculation.

That there is a meteoric water oxygen source for the Kazput barites is unequivocal, and this is even emphasized further with the recent publication of new sulphate d18O data in a large time series compilation (Crockford et al., 2019, cited, and with their new data included in the compilation Fig. 1b), whereas marine sulphates are d18O >6‰ through time, in comparison to the Kazput barite that reaches down to nearly -20‰. Further, d18O values for different water sources are now given in the text (please see below). We emphasize here that there is an ongoing debate about the oxygen isotope composition of seawater in the past - was its d18O near 0‰ or quite different? Yes, we are comparing our Paleoproterozoic data against modern cycles because the relationship between meteoric water and seawater should remain the same through time, as the same process of Rayleigh distillation that sets the range of meteoric water oxygen isotope compositions will not change. What may change is seawater oxygen isotope composition - the source of meteoric water, essentially. However, it is very instructive that clearly marine sulphates (e.g., evaporites in the Crockford et al., 2019 comp) through time register consistently heavier values above 6‰ d18O (whereas, today's seawater sulphate is 9‰ in d18O) that implies seawater has not changed that much since the Paleoproterozoic to now. This is a tricky issue that is still not settled, and simply put, out of the scope of this study. As such, we try to handle it carefully without directly weighing in on this debate, as below...

L95-106: We suggest that oxygen isotope constraints from sedimentary sulphates such as barites can provide critical insight into early Earth sulphide weathering. The majority of oxygen atoms in SO_4^{2-} generated from sulphide oxidation derive from ambient water at the locus of oxidation (e.g., $\delta^{18}\text{O}$ values of seawater $\approx 0\text{‰}$, meteoric waters $\approx 0\text{‰}$ to -20‰ , snow and ice $\approx -20\text{‰}$ to -60‰ {Luz, 2010 #1858}), with a lesser contribution from other oxidants such as O_2 , followed by negligible isotopic exchange {Bao, 2015 #1022}. For example, sulphates in sediments are normally enriched in ^{18}O versus their water oxygen source, where seawater at $\delta^{18}\text{O} \approx 0\text{‰}$ results in marine SO_4^{2-} being $\geq 6\text{‰}$ through time (Fig. 1b). Meanwhile, MSR leaves a diagnostic positive correlation between the $\delta^{34}\text{S}$ and $\delta^{18}\text{O}$ of affected sulphate {Antler, 2017 #2816}. Thus, after its formation, the largely conservative behavior of sulphate and the characteristic fingerprints from MSR permit discrimination between different environmental origins and transformations of SO_4^{2-} .

Unfortunately the crucial role of metabolisms in biogeochemical cycles is not discussed much, neither is the possibility of different dominating metabolisms 2.3 Ga ago, especially when it comes to the endless diversity of metabolisms, pathways and species involved in the sulfur cycle.

As above, we do our best to introduce the modern biological influence on the S cycle (in the introduction, now and as it was in the earlier manuscript version) before speculating, although based on some interesting evidence, on the biological role in the sulfur cycle that could have influenced the isotopic signatures in our study.

Important new discussion has been added on the potential biological role (where MSR = microbial sulphate reduction):

L231-247: A unique combination of environmental conditions likely contributed to the isotopic signatures to Kazput barite, but perhaps they were also preserved due to the sulphuretum, or consortium of sulphur-metabolizing microbiota, active in the Turee Creek Basin during the Paleoproterozoic. Besides prevalent microbial mats in the Kazput carbonate, it also contains microfossils that have been interpreted as sulphur-oxidizing filamentous bacteria {Schopf, 2015 #2544}. What is remarkable about such bacteria from an isotopic perspective is that a sulphuretum of SO_4^{2-} -reducing microbes and sulphur-oxidizing bacteria can produce sulphide that gets re-oxidized to SO_4^{2-} , via elemental sulphur, with no net S isotope fractionation {Fry, 1988 #3432}. We can imagine that sulphur-oxidizing bacteria may have contributed to Kazput barite signals, perhaps by introducing oxygen isotope signatures of Turee Creek Basin water to SO_4^{2-} produced during re-oxidation of MSR-produced sulphide, or by enhancing rates of sulphide oxidation on land. The negative correlations between Kazput barite $\delta^{18}\text{O}$ and its sulphur isotope parameters (Fig. 3c & 3d) excludes overprint from MSR because, in contrast, MSR is indicated by positive $\delta^{18}\text{O}$ - $\delta^{34}\text{S}$ correlation {Brunner, 2012 #3362}. Further speculation about the biological role in Turee Creek sulphur cycling is open, as the barite sulphur and oxygen isotope signatures are dominated by a strong source effect that mask possible microbial influence.

L332-342: Such enrichment in S-MIF-bearing pyrite has previously been observed in microbialitic carbonates where carbonate precipitation effectively sealed off the sediment from the water column, promoting closed-system MSR {Ono, 2009 #3388}. Stromatolites and carbonate are indeed prevalent throughout the Turee Creek Group, and especially in the Kazput Formation carbonates. The $\delta^{18}\text{O}$ compositions of CAS from S. Africa are yet to be measured, but it is likely that they are relatively enriched in ^{18}O as compared to the Kazput barite. As discussed here, the open system MSR suggested by S isotope evidence from S. Africa would also exert strong control on the $\delta^{18}\text{O}$ of SO_4^{2-} . In order to preserve the persistent S-MIF signals recorded throughout the Turee Creek Group succession, as compared to S. Africa, in W. Australia it seems that closed system MSR, within the sediment, was required.

Unlike Peng and collaborators, who simply stated that the “ambient water” where the sulfate formed must have had an usually low $\delta^{18}\text{O}$ value and therefore likely reflected the presence of glacial meltwater (possibly in porewaters), the authors jump in and interpret their dataset as a reflection of oxidative weathering of reduced sulfur on land. Low $\delta^{18}\text{O}$ of sulfate can certainly suggest oxygen coming from continental waters, although a robust demonstration of this would require additional bibliographical references, as relatively recent papers exist on the topic (eg. Calmels et al., 2007; Turchyn et al., 2014- the question is crucial to the demonstration of the authors’ thesis yet the only references made are to a paper quoting earlier, less directly relevant, studies and a self-reference).

A new section of discussion, "Context and origin of the Kazput barite" (L132-278) is added that focuses on the origin of the Kazput barite, specifically interrogating different possible origins before arriving at our conclusion of sulfide weathering on land. The prior version of the manuscript was greatly condensed so we appreciate the opportunity to expand the discussion in this regard.

The above-mentioned papers on riverine sulfate (Calmels et al., 2007 and Turchyn et al., 2014) are good references (and familiar) but we find them to be less directly applicable to this study, however we add the Calmels reference for its use of $\delta^{18}\text{O}$ to demonstrate a pyrite-derived origin of riverine sulphate. Both the above-mentioned studies are geographic transects of river basins sampled within a limited time frame - much different than sampling one location over a long period of time, where the latter is more akin to the type of information in a rock core. Therefore, although it is a self-reference, the 4-year study on Mississippi River sulfate (Killingsworth et al., 2018, cited in the text) is more directly applicable here because it is on a continental scale and long term - so it at once covers a large scale and its isotope compositions are well constrained so we suggest it is a more convincing demonstration for the purposes of this work. Calmels citation added, as below...

L271-273: Such mixing, between sulphide sources with distinct S-isotope compositions but similar overlapping ranges of $\delta^{18}\text{O}$ values of their resulting SO_4^{2-} , has been observed in major river systems today {Calmels, 2007 #35; Killingsworth, 2018 #3193}.

Again, we emphasize that Fig. 1b demonstrates what the $\delta^{18}\text{O}$ of sulphates from different origins looks like through time - seawater sulphate $\delta^{18}\text{O}$ is above 6‰. Sulphate that reaches down to -20‰ is only observed in 3 cases so far, including Kazput barite - and all of them seem to snow/ice meltwater.

Such $\delta^{18}\text{O}$ signature of sulfate cannot be seen in the modern ocean because this signature is buffered by much higher sulfate concentration.

This point requires further discussion. Do the authors think that the whole Paleoproterozoic oceanic sulfate reservoir has an unusual signature? How would this value be maintained? In other words, is the barite authigenic (therefore all oceanic sulfate has an unusual $\delta^{18}\text{O}$ signature) or precipitated on land and transported to the ocean? I can foresee difficulties with either interpretation.

We do not think that the whole ocean sulphate reservoir had such low $\delta^{18}\text{O}$ signatures in the Paleoproterozoic. Hopefully the new discussion addresses this more clearly, and we appreciate being able to clarify this aspect, as pasted from the text of comparison against S. Africa further above (L332-342), and here, where it is emphasized that what is captured is likely the sulphate from continental weathering:

L342-344: Further, we suggest that low global marine sulphate concentrations and an intracratonic basin setting made the capture of weathering input fluxes, and their SO_4^{2-} isotope compositions, possible in the Turee Creek Basin.

Furthermore, along with the more detailed discussion of the Kazput barite origin, we propose a model for the barite formation that distinguishes between the formation of sulphate versus the capture of that sulphate with barium to form barium sulphate (barite). We suggest that the formation of the sulphate was indeed separate from the formation of barite from that sulphate, as below....

L249-266: We suggest that the context of Kazput barite and its isotopic signatures are explained by pyrite oxidation on land that produced SO_4^{2-} which was delivered by riverine and/or groundwater input to a nearshore environment of high bioproductivity. The first-order controls on these signals are relative sea-level and the exposure of sedimentary rocks to weathering that should be expressed as a strong source effect on the S-isotope compositions of SO_4^{2-} . The Fe/Al ratios in core TCDP3 are 0.33 on average, excluding enriched intervals >1 (Fig. 2, Supplementary Table 2), which is consistent with oxic water columns whose Fe/Al are close to or less than average shale at 0.5 {Lyons, 2006 #1861}. The enrichments in Fe/Al and total sulphur observed in two muddy intervals below the carbonate are also likely pyrite (FeS_2) enrichments although this was not quantified directly, while the few measured pyrite concentrations are 0.32 weight % on average (Supplementary Table 2). The Fe/Al and total sulphur enrichments imply either transient water column anoxia, or movement of a chemocline during transgression-regression cycles. Anoxic sediment porewaters favoring MSR would also provide a flux of barium in sufficient concentration to precipitate SO_4^{2-} as barite at the sediment-water interface {Magnall, 2016 #3413}. Remaining porewater SO_4^{2-} would be quantitatively reduced to sulphide, via closed system MSR, with capture by iron to form iron sulphides (pyrites) during early diagenesis in the sediment {Philippot, 2018 #3192}.

The Paleoproterozoic ocean was similarly concentrated in sulfate compared to the Neoproterozoic, possibly even more if we follow the published [SO₄] of Planavsky et al. (2012) around the 2.3-2.1 Lumagundi excursion, or take into account the evidence for 2.32 Ga sulfate evaporites in the Gordon Lake Formation of the Huronian Supergroup or the Kona Dolomite of the Chocoma Group (e.g. Bekker et al., 2006). The latter suggests that sulfate concentration was high at 2.31 Ga based on the age correlation between the Gordon Lake formation and the Meteorite Bore Member (Philippot et al.; 2018) so prior to the deposition of the Kazput formation. As a result, even if sulfate was lower than today, it was probably not as low as the authors seem to suggest (and higher than in the case of Peng et al.'s study) and if oxygen was scarcer than today, then oceanic MSR might already be quite active and influencing sulfate $\delta^{18}\text{O}$ values. If the authors assume an entire oceanic sulfate reservoir being influenced by continental sulfate it seems necessary to provide a numerical quantification of how much "continental" sulfate would be required to overcome the expected "marine" $\delta^{18}\text{O}$ signature. If not, can we assume the basin to be, similarly to the Neoproterozoic formation of Peng et al., more easily influenced by continental inputs? Would that be in agreement with the body of literature on the Kazput formation geology?

Overall, the authors don't seem to solve that ambiguity: where did the sulfate ion form and where did the barite precipitate?

First, the Peng et al., 2013 study is not relevant here in a discussion about sulphate concentrations because that study shows a case of sulphide minerals (pyrite) being oxidized *in situ* with meltwaters to form sulphate adjacent within the same rock. Therefore, their results, while being applicable in discussion with respect to very low sulphate d18O values down to -20‰, do not make for a comparison of aqueous sulphate concentrations.

Further, we have clarified that we do not think that our results represent a global seawater sulphate pool, as was discussed in the previous answer above.

Other than that, seawater sulphate concentrations in the earliest Paleoproterozoic are still an open question. We add new referenced discussion in the introduction that makes this point clear, while at the same time it emphasizes the difficulty in finding sulphate records during that time, for which barite, as we found, is an excellent alternative to evaporite and CAS, as below...

L76-93: During the Archaean Eon and into the early Paleoproterozoic, the sparse sulphate record has prevented closer examination of the oxidative side of the early Earth sulphur cycle. Evaporites and bedded barites are discontinuous, while more continuous records are hindered by weakly concentrated carbonate-associated sulphates (CAS) that are vulnerable to contamination {Peng, 2014 #1266}, diagenetic alteration {Rennie, 2014 #1267}, and loss during metamorphic recrystallization {Fichtner, 2017 #3128}. Ideally, CAS concentrations in rocks should be proportional to original ambient seawater SO_4^{2-} concentrations, but the latter may be trivial in early environments. In the late Archaean, before 2.5 Ga, marine SO_4^{2-} may have been less than 2.5 μM , as inferred from modern analog environments {Crowe, 2014 #1257}. Between 2.25 to 2.1 Ga, estimated seawater SO_4^{2-} concentrations were 5-20 mM, as based on the $\delta^{34}\text{S}$ variability in extant CAS records {Planavsky, 2012 #507}. While at 2 Ga, the SO_4^{2-} concentration was at least 10 mM, as geochemically constrained with the first basin-scale bedded evaporites {Blättler, 2018 #3137}. During the GOE interval, between 2.5 to 2.3 Ga, however, it can be difficult to obtain enough sulphate from sediments for sulphur isotope measurements, where up to 1500 grams of carbonate may be necessary for sufficient CAS {Guo, 2009 #2455}. Such low sulphate sample yields may prohibit the additional measurement of oxygen isotopes. However, barite (BaSO_4) is a highly insoluble and diagenesis-resistant alternative that provides a more robust record of sulphate S and O isotope compositions {Griffith, 2012 #1642}.

An intriguing yet undiscussed aspect of this research is the extreme similarity between pyrite and barite isotopic composition, especially the $\delta^{34}\text{S}$. If the barite precipitated in the basin from sulfate ions that were formed on land, would it make sense that the S composition of local barite and pyrite overlap perfectly? It seems that in this case, the pyrite would be formed from sulfide resulting of microbial sulfate reduction and would most likely have lower $\delta^{34}\text{S}$ than sulfate (as is it the case for most barite-pyrite comparisons, Archean or not).

Indeed, this is a key aspect - the close S-isotopic match between co-occurring pyrite and our barite - for which the discussion is now expanded. Also, we now make more effort throughout the text to distinguish between dissolved sulphate (SO_4^{2-}) versus barite, as it is critical (and was not clear enough before) to consider their separate formation. We consider that the SO_4^{2-} was formed on land, transported to the marine/lagoonal environment, where it was captured as barite and remaining SO_4^{2-} reduced to sulphide and retained as Fe-sulphide mineral (as already mentioned and pasted further above, L249-266).

Finally, after considering different possibilities, we suggest that the correlations between $\delta^{18}\text{O}$ and S-isotopes, particularly the $\Delta^{33}\text{S}$ signal, is most parsimoniously described as pointing to the provenance of this unusual (for its time, at <2.31 Ga) S-isotope signal being from oxidation of sulphide on land. The point is that, considered alone, the barite and pyrite S-isotope signals do indeed indicate that they came from the same sulphur pool with little modification between them. The oxygen isotope $\delta^{18}\text{O}$ adds another level of interpretative power to parse the different possibilities. Relevant discussion pasted below.

L189-206: Secondary oxidation of pyrite to SO_4^{2-} could also occur at low temperature within the rock during its burial history. If late oxidation by infiltrating fluids occurred, then carbonate, which is very susceptible to diagenetic alteration of its oxygen isotope composition {Ryb, 2018 #3199}, should register alteration by the same ^{18}O -depleted waters implicated in the oxidation of pyrite to form the barite. Instead, comparison of $\delta^{18}\text{O}$ data from Kazput carbonate with our barite reveals separate environments of formation for the carbonate and SO_4^{2-} . The Kazput carbonate has an average $\delta^{18}\text{O}_{\text{carb}} = 18.4\text{‰}$ (here converted to the VSMOW scale) {Barlow, 2016 #2879} that is typical for marine carbonates of similar age, but 10‰ lower than present-day marine carbonates (Fig. 1b). Meanwhile, the oxygen isotope fractionation between pyrite-derived SO_4^{2-} and water spans a range of $\delta^{18}\text{O}$ values falling between 0‰ and +20‰ for aerobic and anaerobic conditions {Gomes, 2017 #3154}. Regardless of assumed fractionations versus their water sources, the only occasions when sulphate $\delta^{18}\text{O}$ values are distinctly low as compared to coeval carbonates are when they do not share the same water-oxygen sources, such as for sulphate from terrestrial settings versus marine carbonates (Fig. 1b). Additionally, it is feasible to oxidize pyrite under anoxic conditions via radiolysis of water that is accompanied by an enrichment of 1.5‰ to 3.4‰ in the $\delta^{34}\text{S}$ values of SO_4^{2-} versus pyrite {Lefticariu, 2010 #3430}, but this scenario is ruled out because Kazput barite does not show such S-isotope enrichment versus previously reported pyrite (Fig. 2).

L220-229: Whether the Kazput barite water-oxygen came from meteoric waters that were, in turn, derived from seawater with a lower $\delta^{18}\text{O}$ composition than at present, such as -10‰ {Kasting, 2006 #3107}, or from Paleoproterozoic glacial meltwaters as low as -43‰ {Herwartz, 2015 #2799}, in both cases a strongly ^{18}O -depleted non-marine water oxygen source is required. Kazput barite $\delta^{18}\text{O}$ data show negative correlation with $\Delta^{33}\text{S}$ and $\delta^{34}\text{S}$ data (Fig. 3c and 3d), which, given the evidence, is most parsimoniously explained by a shared provenance of oxygen and sulphur in low temperature oxidation of pyrite to form SO_4^{2-} . We suggest that the negative isotopic correlation, where low $\delta^{18}\text{O}$ is

associated with high $\Delta^{33}\text{S}$, fingerprint the source of stronger S-MIF signals being delivered to the Turee Creek Basin as coming from the weathering of sedimentary rock exposures on land.

This overlap instead suggests that the barite results from local sulfide oxidation (possibly in porewater?), but then how to explain the $\delta^{18}\text{O}$ values? This scenario seems however more likely than sulfate being delivered to the basin as a dissolved species and afterwards either buried as barite or reduced as sulfide with no S isotopic fractionation.

The barite itself could have precipitated on land much earlier than the deposition of the Kazput formation. Had the barite been transported from land, it would be harder to interpret the age and meaning of the signal. Alternatively, could the barite have formed later *in situ*, through, say, local oxidative fluid circulation, which might carry a very different oxygen isotopic signature? This is another possible explanation of why the barite is so similar to the pyrite (from a S isotope point of view) that remains unexplored.

These exact possibilities - localized sulphide oxidation (within the sediment/rock) versus delivery of sulphate as a dissolved species to the basin - are addressed in the new discussion of barite origin, as already discussed above. We appreciate being able to clarify the different possibilities that Reviewer 1 highlights.

The possibility of detrital barite is also now discussed in the text, as below.

L208-218: A detrital origin may be indicated by the unexpected young age, <2.31 Ga, of the Kazput barite sulphur isotope anomalies. Evidence for an anoxic atmosphere that is found in sedimentary rocks are rounded detrital grains of pyrite and uraninite that were not dissolved during riverine transport due to low oxygen availability {Reinhard, 2013 #1165}. The pyrite in the Kazput Formation is not detrital, and instead appears to have grown *in situ*. Kazput pyrites occur as microcrystalline aggregates, euhedral crystals, and elongated layer-parallel concretions in a range of sizes (Supplementary Figs. 5 and 6). In contrast, Kazput barites are difficult to ascertain due to their size, $<10\mu\text{m}$, and disseminated occurrence (Supplementary Figs. 3 and 4), but share a tight S-isotopic match with the co-occurring diagenetic pyrites. Therefore, as compared to the diagenetic pyrite, the different mode of occurrence but mutual sulphur source for the barite precludes its detrital origin.

Given that there is also another significant option for interpreting the data, as oxygen in sulfate can also reflect metabolic pathways, I think that the statement in line 74-75 attempting to put this option aside oversimplifies the actual situation (particularly as there is no reference to support the statement that $\delta^{18}\text{O}$ of sulfate uniquely indicates whether the sulfate was formed on land or in the ocean).

Continental and oceanic waters do have different oxygen isotopic compositions but sulfate formation can be more complicated. In addition, if the oxygen can be sourced from the water, it can also come from O_2 depending on the pathway. The authors don't acknowledge an abundant body of literature on the topic of oxygen isotopic fractionation during sulfate formation. Gomes and Johnston 2017 provide a review of the influence of

the pathways, whether biotic or not, on the oxygen isotopic composition of the resulting sulfates.

The reference mentioned by Reviewer 1, Gomes and Johnston 2017, is indeed an important review of S- and O-isotope fractionations that we substituted into the manuscript and adjusted the text accordingly, as below:

L197-199: Meanwhile, the oxygen isotope fractionation between pyrite-derived SO_4^{2-} and water spans a range of $\delta^{18}\text{O}$ values falling between 0‰ and +20‰ for aerobic and anaerobic conditions {Gomes, 2017 #3154}.

Although the S-isotope variations, as discussed in the text, may indicate some biological transformation, the O-isotopes do not show this. Evidence of biological transformations in the early sulphur cycle can be elusive (as discussed in Dave Johnston's 2011 review paper, for example). In the newly added text we attempt to do the discussion of microbial influence on S-isotopes in Turee Creek Basin more justice (as already discussed above).

Oxygen isotopes in sulphate are complicated, more so than S-isotopes, but here the low d18O of Kazput barite and the negative correlation between d18O and both d34S and $\Delta 33\text{S}$ indicates that a source effect is the first order consideration.

As for whether sulphate d18O can indicate sulphate formation on land or in the ocean, it is perhaps unknown territory that the present study attempts to open up because we feel that it will offer important interpretive power when looking at sulphate O-isotope records even before atmospheric oxygenation. There is not a real reference for this idea, this is our assertion, and we feel that it is self-evident when looking at the d18O of sulphate through time (Fig. 1b), which shows marine sulphate being >6‰, as already discussed above. What we can offer, more concretely, is a better demonstration of this idea, as now the introduction text includes ranges of water d18O values, mention of other oxidants such as O_2 . Although, given the low d18O of our barite and a typically dominant water signal in sulphate, we do not go into detail about O_2 being a direct oxidant - also because there is even the question of O_2 in the atmosphere at the time of having relatively strong $\Delta 33\text{S}$ signals that indicate anoxia. Relevant text was pasted earlier, L95-106, before which we our assertion was made clear:

L95-96: We suggest that oxygen isotope constraints from sedimentary sulphates such as barites can provide critical insight into early Earth sulphide weathering.

If, as stated by Killingsworth and collaborators, the oxygen isotopic composition of the newly formed sulfate is not buffered by oceanic sulfate, is there a reason for not assuming, for example, that instead of oxidative weathering, what they observe is biologically mediated anaerobic oxidation of sulfide in a closed system (porewaters, for example)?

This is already addressed above - we do not consider that Kazput barite capture a global oceanic sulphate pool. The negative correlation between barite d18O and S-isotopes

indicates a source effect that is most parsimoniously explained by pyrite oxidation on land with meteoric water, as opposed to anaerobic oxidation of sulphide in porewaters.

L. 70 references would be good

Original lines 70-72: Direct assessment of continental sulphide oxidation is made possible here by measurement of the oxygen and multiple sulphur isotope compositions of sedimentary sulphate.

This line has been removed, and the newly modified discussion is in L95-106 that was pasted further above, where we make it clear that this is our assertion.

L.73 primarily yes, but possibly significantly from O₂

We now mention O₂ as an oxidant, in the L95-106 text further above.

L. 74-75: this is true only if the involved metabolisms and pathways are known and constrained.

Original L74-75: which permits discrimination between sulphate originating in seawater (relatively ¹⁸O-enriched) and continental waters (relatively ¹⁸O-depleted).

There is an important difference between marine sulphate d¹⁸O through time (>6‰, as discussed multiple times above), and a dramatically different signal of sulphate d¹⁸O that reaches down to -20‰. Again, we assert that a source effect dominates and that we expand the discussion of microbial influence on the Kazput barite S- and O-isotopes (as already discussed further above), but going into more even more detail than we have in the new discussion would be speculative. We do not see clear evidence of microbial transformation of Kazput barite S- and O-isotopes.

L. 126 A ref. would be good

References were there, indeed. Here, with the new line numbers:

L195-197: The Kazput carbonate has an average $\delta^{18}\text{O}_{\text{carb}} = 18.4\text{‰}$ (here converted to the VSMOW scale) {Barlow, 2016 #2879} that is typical for marine carbonates of similar age, but 10‰ lower than present-day marine carbonates (Fig. 1b).

L. 140: different fractionations for water-sulfate exist depending on the pathway so things are maybe not as straightforward as suggested here.

Yes, absolutely. Previously we considered ~10‰ enrichment of d¹⁸O in sulphate versus its ambient water oxidant as being best representative of sulphide oxidation on a large scale. For example, marine sulphate today is ~+9‰ versus seawater at 0‰. Mississippi River sulphate is ~+3‰ on average versus the river water average ~-6‰. In both those cases, the enrichment of sulphate is +9‰ versus ambient water. We do not consider an

absolute value, just one that is representative, which is further demonstrated in marine sulphates through time that are $>6\%$. The important point is that we don't have the evidence, nor are likely to find, to argue for a water-sulphate fractionation different from a generalized value but this is not critical to the key arguments we make. The relevant discussion is in new lines L95-106 that were pasted further above.

L. 148 further support: ref?

This is not in the new version

L. 156 and following: The authors suggest that their data are at odds with Torres et al. I agree, they seem to be but their way of dismissing the Torres et al. paper is not substantiated and seems misleading. They have two arguments: “present day rivers are strongly buffered by atmospheric S inputs, and the year round dry deposition of atmospheric sulfur was also neglected.” Present day rivers are not necessarily “strongly buffered” by atmospheric S input, even though they are influenced by them. Such a strong statement, not supported by any reference, could appear as dismissing decades of work on the chemical and isotopic compositions of rivers, especially on sulfur cycle. Moreover, atmospheric S inputs, as well as “urban” runoff, are considered by Torres et al. and corrected for. In addition, is there a reason to expect that dry deposition would be much higher than wet deposition? Do the authors have references of data supporting their claim, in particular for this part of Canada? Given that there is over an order of magnitude in sulfate concentrations for catchments that are more or less right next to each other, through what mechanism(s) do the authors think that dry (or wet, for that matter) deposition would be so spatially heterogeneous?

Point taken. Perhaps we were too flippant in how we dismissed the Torres et al evidence. The discussion has been rewritten, as below:

L368-375: Our results are at odds with the conclusion, based on modern weathering of old exposures, that the weathering of Archaean continents would yield sulphate with $\Delta^{33}\text{S}$ summing to nearly 0% {Torres, 2018 #3222}. Perhaps this discrepancy is because present-day rock exposures simply are not comparable in composition and weathering susceptibility to the sedimentary rocks exposed at Earth's surface in the Paleoproterozoic. Furthermore, small positive $\Delta^{33}\text{S}$ have indeed been measured in riverine sulphate from catchments in S. Africa and Ontario today {Asael, 2017 #3447}, which suggests that recycling of S-MIF signals at the Earth surface could occur at least locally.

L. 163: if pyrite burial approached indeed unity, wouldn't the remaining sulfate be nonetheless strongly fractionated? Would that explain that pyrite and barite carry the exact same sulfur signatures? In addition, seawater sulfate concentration in Canfield and Farquhar 2009 is not in agreement with later published scenarios (Planavsky et al., 2012)

Absolutely, remaining sulphate would be strongly fractionated. However, we do not see, or claim to see, such remaining sulphate in our barite records. The above points have both been addressed here in previous discussion and pasted manuscript text further above.

L. 183 "oxidative conditions" is vague and a bit tautologic. All sulfate on Earth's surface results from oxidation at some point as sulfur in the mantle or sulfur emitted by volcanoes is not as oxidized as sulfate, and oxidation of course always requires oxidative conditions. But oxidized species, including sulfate, can be found even through the Archean, so oxidative conditions on the surface of the Earth have existed for a long time before the Kazput formation. What the Kazput formation barite signature seems to require is glacial meltwaters and/or specific metabolic pathways.

This refers to (new lines) L275-278: It is unclear whether the sulphur isotope compositions from our barites reflect specific fractionations during sulphide oxidation on the continent or in an incipient oceanic SO_4^{2-} reservoir, but both require oxidizing conditions.

As it is framed in the text, this is not about local or isolated oxidizing conditions, but pervasive oxidizing conditions attributable to atmospheric oxygenation, as opposed to production of sulphate before 2.5 Ga that would occur despite an anoxic atmosphere. The discussions relevant to Kazput barite origin are now expanded, as compared to before, and hopefully address the concerns stated by Reviewer 1 above, as previously answered here.

Figure 1: references in the main text would be better. Turchyn and Schragg , 2004 is missing.

We do not have enough room to put the compilation citations in the main text so they are in the Supplementary Information. Turchyn and Schrag, 2004 is now included in the $\delta^{18}\text{O}$ compilation (Fig. 1b)

SOM:

fig. S1 is hard to read (especially the spectra)

reference is misspelled (p. 10) and page numbering is wrong towards the end.

Barite SEM detection figures are now redone with more legible spectra, Supplementary Figs. 3 and 4. Page number and misspelling is now fixed.

REVIEW 2

Main text:

Line 39–56 I hate to say this, but the first paragraph is a little ropey. Not bad. Just ropey. I do not object to it on a scientific level, I think the issue here is you're trying to crowd a tonne of information into a very limited space leaving the reader, or at least me, [a little] confused! I think simplicity is the key here and maybe splitting the prose into an extra paragraph. The key point you are trying to get across, I think, is that atmospheric oxygen availability has governed the evolution of the sulphur cycle. Therefore, the mention of oxygen in line 44, for example, is a little premature! The points that I think are worth pulling out are, against an oxygenated backdrop, (1) oxidative weathering

makes sulphate à $\delta^{34}\text{S}$ MSR favours $\delta^{32}\text{S}$ à $\delta^{34}\text{S}$ in concert with FeS₂ burial driving seawater heavy à $\delta^{34}\text{S}$ small isotope effect in Cap-delta à $\delta^{34}\text{S}$; NEW PARAGRAPH) Look at MIF = very different... *I think a little simplification and rewrite maybe warranted here.* Is the following more in line with what you are thinking? Take what you want but it will need citations...

Today, oxygen constitutes 21% of Earth's atmospheric gas budget, driving a marine sulphur cycle dominated by oxidative weathering, generating substantial pyrite-derived sulphate fluxes, augmented by the dissolution of evaporites. Sulphate reduction is associated with a large isotope effect, preferentially converting $\delta^{32}\text{S}$ to sulphide where it can be preserved upon reaction with iron [or organic matter], eventually forming pyrite [sulfurized organic matter]. Consequently, global pyrite burial results in a pronounced isotopic offset between the riverine precursor and seawater sulphate, enriching the latter in $\delta^{34}\text{S}$... *then small $\delta^{33}\text{S}$, maybe? – see following comment]

[This might be better as a new paragraph?] By contrast, the Archean and the early Palaeoproterozoic sedimentary S-isotope record reflects a substantially different sulphur cycle, with large magnitude $\Delta^{33}\text{S}$ values ($\delta^{33}\text{S} \approx \delta^{33}\text{S} - 0.515 \times \delta^{34}\text{S}$), ranging from -4 to 14‰, typifying mass-independent fractionation (MIF) produced from an anoxic atmosphere (Figure 1a). Here, while $p\text{O}_2$ remained below 10⁻⁵ times the present atmospheric level (PAL), photochemically induced S-MIF resulted in at least two geochemically distinct exit channels that escaped homogenization, with insoluble S₀ and soluble SO₄ thought to carry positive and negative $\delta^{33}\text{S}$ values, respectively.

With Reviewer 2's suggestions taken into account, the first section has been rewritten and split into two paragraphs, as below:

L39-50: Today, Earth's oxygen-rich (21% O₂ by volume) atmosphere drives a marine sulphur cycle dominated by oxidative weathering inputs. Rivers supply the ocean with substantial loads of dissolved sulphate (SO₄²⁻) derived from the oxidation of pyrite and the dissolution of sulphate minerals. Once in the ocean, the sulphur and oxygen isotope composition of SO₄²⁻ is modified during microbial sulphate reduction (MSR) processes. During MSR, the partial reduction of SO₄²⁻ preferentially converts $\delta^{32}\text{S}$ to sulphide that is mostly re-oxidized, whereas a fraction of the sulphide product is retained as iron sulphide minerals {Canfield, 2009 #3116}. The mass-dependent sulphur fractionations induced by MSR lead to marine sulphide having a large, but relatively $\delta^{34}\text{S}$ -depleted, range in $\delta^{34}\text{S}$ (see Methods for isotope notations) and a $\Delta^{36}\text{S}/\Delta^{33}\text{S}$ slope of ~ -9 {Johnston, 2007 #2682}. Simultaneously, marine SO₄²⁻ is concentrated at 28 mM, enriched in $\delta^{34}\text{S}$ ($\delta^{34}\text{S} = 21$ ‰ {Paytan, 1998 #50}), and has a small $\delta^{33}\text{S}$ enrichment ($\Delta^{33}\text{S} < 0.1$ ‰).

L52-60: Before oxygenation, the surface sulphur cycle was controlled by atmospheric inputs that can be traced within Archaean age, 4.0 to 2.5 Ga, sedimentary rocks displaying strong sulphur mass independent fractionation (S-MIF) of $\Delta^{33}\text{S}$ values between +14‰ to -4‰ {Farquhar, 2000 #2456; Guo, 2009 #2455; Gumsley, 2017 #2809} (Fig. 1a). There are lingering geochemical {Paris, 2014 #1263} and quantitative {Claire, 2014 #481} challenges still to be understood about the preservation and generation of Archaean S-MIF signals. However, it is generally accepted that S-MIF results from atmospheric photochemical reactions operating under low O₂ of <0.001% of the present

atmospheric level of oxygen, causing surface sulphur fluxes of insoluble S_0 with $\Delta^{33}S > 0\text{‰}$ and soluble SO_4^{2-} with $\Delta^{33}S < 0\text{‰}$ that were not homogenized during their transfer into sedimentary rocks {Pavlov, 2002 #2464; Halevy, 2013 #2530}.

Line 45: Caution, this is a scientific objection. Non-zero cap-delta-33 does not mean that its S-MIF. S-MIF has a distinctive magnitude (> 0.4) and a distinctive slope (-0.9 to $-1.5?$). This makes “near-zero, but, very small, signal of mass independent fractionation” an incorrect statement. Please ensure that this is removed from the amended manuscript.

Point taken. This is now removed.

Line 54–56: I’m not sure that we know that there were only two atmospheric exit channels. I agree that more than one is a requisite, but the upper number is unknown given we don’t understand the genesis of geologically analogous S-MIF. The sign of each phase is also not known with certainty – See discussion of Claire et al. (2014). I would deemphasise these points a little. I’ve tried to give an example in the rewritten text above.

Agree. We generalize, but the generalizations are rooted in the rock record evidence, whereas our primary concern is how these atmospheric fluxes appear at the surface (the key here being “appear”). This part is rewritten to make that aspect clear, as pasted above, L52-60.

Line 59 and throughout: Can we add fully propagated uncertainties to all the quoted ages. This is important for the rationale that you’re presenting. If the chronological constraints are imprecise then we may not need the more complex interpretation.

Full uncertainties now added:

L64-67: There, $\Delta^{33}S$ decreases below 0.4‰ between 2316 to 2326 ± 7 million years (Ma) ago, a horizon that has been interpreted as the shift to $>0.001\%$ of present atmospheric oxygen, the Great Oxidation Event (GOE) itself {Bekker, 2004 #268; Guo, 2009 #2455; Luo, 2016 #2669}.

Line 60: I hate $<$ and $>$ in prose. Can you remove these throughout and use appropriate words? This makes the text a lot easier to follow. This is also true of a lot of chemical symbols that are not necessary. Also, I know that Genming used the term “rapidly” in his abstract but, in fairness, this was on the 1–10 million-year time-scale. This is not rapid compared with atmospheric residence times. Can you add these timescales in parentheses so the reader is informed?

This was rewritten, as above. Care was taken throughout the newly revised text to reduce the use of “and” when a more useful word could be used instead.

Line 61: Please remove the quotations. I do not think they are needed. This is well known terminology. In time, it may be convincingly demonstrated that the GOE is an

oxymoron but, for me, *event* has definite linguistic connotations. Also, “An alternate hypothesis exists,” could be simplified to “Alternatively” if you need to lose words(?).

Use of GOE was fixed (above, L64-70). Suggested use of "Alternatively" is now incorporated.

Line 62: I don't think that Gyr has been defined and I could be wrong but I think you should use Ga rather than Gyr. I think it should read “~2.5 billion-years (Ga)”, for example. Maybe this is semantics and/or maybe that I'm completely wrong. Moreover, I'm sure the typesetters will resolve the debate but the following website might be useful? https://www.ldeo.columbia.edu/~ncb/Selected_Articles_all_files/25_Stratigraphy.6.100.pdf

We changed all ages to "Ga", but defer to the editorial office if they prefer a different notation.... (but honestly, the correct Ga/Gyr usage remains unclear because it seems that they are used interchangeably in different journals, depending on the preference)

Line 63: “Could be” is perhaps better written “could have been”?

Changed to "is attributable", L68

Line 65: The end of the sentence seems odd and the sentence is also humongous. Perhaps close the sentence after ref 10 and open a new one with something like: “Subsequently, supporting^{11,12} and conflicting¹³ arguments regarding the importance of this crustal memory effect have emerged but a consensus is yet to be reached...”?

Now split into two, more digestible, sentences:

L67-71: Alternatively, the slow disappearance of $\Delta^{33}\text{S}$ signals from the rock record after 2.45 Ga is attributable to long term oxidative weathering of an older, S-MIF-bearing, continental sulphide reservoir whose anomalous isotope compositions (i.e., $\Delta^{33}\text{S} > 0.4\%$) would be transferred to sulphate until this source was exhausted {Farquhar, 2003 #2458}. Subsequent evidence has supported {Reinhard, 2013 #2850; Philippot, 2018 #3192} and challenged {Torres, 2018 #3222} this assertion.

Line 68: I don't think you need “within the interval”. This could be omitted if words were needed.

Now stated as L73-74: Paleoproterozoic surface sulphur cycle ca. 2.5 to 2.3 Ga

Line 70: There's nothing wrong here, it's perhaps just not as punchy as it could be. How about: “Coupled oxygen and sulphur isotope measurements from sedimentary sulphates offer a direct assessment of continental sulphide weathering. Sulphate produced via surficial sulphide oxidation derives its oxygen primarily from the ambient water at the locus of oxidation, experiencing negligible isotopic exchange thereafter¹⁴. Consequently,

seawater-derived ^{18}O -enriched sulphates can be distinguished from their ^{18}O -depleted continental counterparts....”

This has been rewritten

Line 74: *Question.* Is sulphate oxygen exchange immune to modification via biological cycling? For example, via microbial redox? I haven't read the reference you cite but could sulphate availability (or other nutrients) play a part? Maybe some additional clarification would be beneficial to help the non-experts like me.

Please see above. Yes this aspect was addressed.

Line 75: *Question.* Can you indicate the ranges in the enrichments/depletions?

As above, this was indeed addressed by adding (referenced) oxygen isotope ranges for water.

Line 76: This would flow better with a linking phrase. For example: “It follows, therefore, sulphate...” Also, do you mean deposited or formed? I would the latter because it does not care that its deposited. As soon as it is formed then it should inherit the signal that you describe. I also favour the latter because I would not describe CAS as being deposited. Finally, maybe it's me, but you use the word sulphate a lot. Given the subject of the manuscript this is understandable but do you want to discriminate between solid and dissolved sulphate? Maybe include barite and carbonate associated sulphate in parenthesis?

Changed to sulphate being "formed" or simply its "occurrence". Used linking word of "Accordingly". Rewritten text previously pasted, as above.

Great point about distinguishing between solid and dissolved sulphate. This is also important later in the discussion of the formation of dissolved sulphate versus the formation of barite. Throughout the text we now use SO_4^{2-} to denote dissolved sulphate versus sulphate minerals, for which we use the specific evaporite or CAS or barite.

Line 79–86: Again, these are long sentences that seem to lose their impact. How about “The paucity of 2.5–2.3 Ga evaporites, in concert with the generally low-level of CAS in appropriately aged carbonate successions, has, thus far, precluded this two-pronged isotope approach. Barite, however, is highly insoluble and is much more resistant to diagenetic modification than CAS, providing an alternate and attractive sulphate repository¹⁵”.

This whole section is now rewritten and expanded, in L76-93 that was previously pasted further above.

Line 88: Given that most of the delta denotations have not been defined then I think that the opening part of the sentence would be better written: “Here, we report quadruple

sulphur (QSI; d34S, D33S, D36S) and oxygen (d18O) isotope data from trace barites that were chemically extracted from..." I also think that there is too little description of the materials that you have analysed. This is conspicuously absent from the supplement, also. Figure 2 seems to show you targeted multiple lithologies, yet the text simply says "finely-laminated and fine-grained" does this describe the sandstones at the base of the succession? Please can you clarify which core you exploited in the manuscript and add some more detailed text. Using subheadings will allow this to fit more smoothly

The results are now reported in a very concise paragraph. Expanded description of the analyzed materials is now included, as well as generous new materials in the supplemental, including additional core photos, photomicrographs and XRF scans, as was previously discussed in response to Reviewer 1. Within the main text, more detailed discussion of the geologic context is woven throughout, also as previously discussed here.

Line 90: I'm sure Gyr should be Ga.

Fixed, as previously stated.

Line 91: Sentence beginning "It is noted" reads like a kind of afterthought. To make this read more intentional perhaps: "Barite is known to precipitate under sulphate impoverished conditions when barium is concentrated by organic matter¹⁶, a scenario we envisage for the deposition of the Kazput Formation given the extremely low CAS (< 2 ppm) and crust-like whole-rock barium concentrations (< 600 ppm; SOM)." *Specific questions:* (1) Do you want to mention the relatively d34S enriched pyrite as a line of low sulphate availability? (2) I flagged that the Ba concentrations are crust-like above, but I think they are a bit high. Is this consistent with the suggested mechanism? Low sulphate but lots of sulphate weathering? Is this OK?

This section is now changed and removed.

Line 95: Again, another sentence that could be streamlined for effect. Better as: "Radiometric age constraints (Re-Os and zircon U-Pb¹⁷) from the Turee Creek Group, Western Australia, demonstrate that S-MIF persists in strata younger than $2.31 \pm X$ Ga, in marked contrast to QSI records from other cratons where S-MIF is apparently lost at $2.33 \pm X$ Ga⁹?" Apparently may be key in this sentence when I get around to writing my manuscript...

This text is now changed, in previously pasted L108-117.

As for the last point, importantly, the idea of permanent/transient S-MIF loss is now left more open in the final discussion (as already pasted much further above), with what is hopefully a more balanced appraisal in the final discussion section **Implications for surface environment response to oxygenation.**

Line 98: “Its persistence” is unsupported and should be avoided. This is better written “The persistence of S-MIF”, however, perhaps “The asynchronous demise of S-MIF between cratons is reconciled with a crustal memory effect and differential recycling in different basins; however, the notion of asynchrony is based on poorly developed chronostratigraphic frameworks and remains equivocal” is punchier? Worth noting that cratons is perhaps not the best expression given Vaalbara... Is this, in fact, a basinal phenomenon?

This is rewritten, as follows:

L110-114: Multiple sulphur isotope records ($\delta^{34}\text{S}$, $\Delta^{33}\text{S}$, $\Delta^{36}\text{S}$) in early diagenetic pyrites from the lower Kazput Formation, in drill core 3 from the Turee Creek Drilling Project (TCDP3) (Supplementary Figs. 1 and 2), have shown the persistence of S-MIF in sulphides that are younger than expected (<2.31 Ga) for sediments deposited under a globally oxygenated atmosphere {Philippot, 2018 #3192}.

The idea of a basinal phenomenon is now discussed in the final discussion section, mentioned previously here (in response to Reviewer 1).

Line 101: Please change “with D33S from” to “featuring D33S values from”. You can also delete “are further characterized by low” it’s subjective and not necessary. Another observation, the prose is a hybrid between American and English, check for consistency and remove all the unnecessary Zs from the prose and the figures. After all, *Nature* is British.

Now changed, following the above suggestions:

L125-128: All barites register signals of S-MIF, featuring $\Delta^{33}\text{S}$ values from +0.62‰ to +1.55‰ and $\Delta^{36}\text{S}/\Delta^{33}\text{S}$ ratios falling within -0.9 to -1.5‰ (Figs. 2 and 3, Supplementary Table 1). The $\delta^{18}\text{O}$ values span +2.2‰ to -19.5‰, with an average of -11.0‰ (VSMOW).

Indeed, point taken about the American/English hybrid. Would appreciate editorial input on this, as we are used to writing in American English and if it is possible our preference is for sulfur/sulfide/sulfate without the sulphur "ph", but we are happy to carefully check for British English formatting, following editorial instructions because we are not familiar with the proper word British english spellings with s/z.

Line 103: “The D33S data”, should read “These D33S data”. Remember, data are always plural. How this sentence is written speaks to the most special data measured over the last 2.5 billion-years, I think you mean that “These D33S data constitute the highest values reported from sedimentary sulphates younger than 2.5 Ga, while the d18O...”.

This sentence no longer exists as such.

Line 107: Again, there's nothing wrong, but the following makes it more easy to read: "The exceptionally low d18O values preserved in the barite from the Kazput Formation indicates that the precursor sulphate did not have a typical marine origin." This is definitely an optional edit.

This sentence no longer exists as such.

Line 108: This is not well explained; how do your d18O data rule out sulphide oxidation? There is only a figure and no documentation of what you've done besides a figure caption. You should expand on this. Perhaps adding something like: "Oxygen isotope measurements of sulphate sourced from laboratory water, as well as from artificial pyrite oxidation experiments, preclude laboratory-induced oxidation during handling, in turn necessitating a palaeoenvironmental interpretation." in the main text?

This was handled in the text, in the Methods, and supplementary figure 7.

Main text:

L173-176: We performed extensive tests of pyrite oxidation during barite extraction, as well as during attempted extractions of CAS, where pyrite oxidation during sample processing was excluded (see Methods).

Methods:

(CAS extraction)

L406-417: Trivial CAS yields from Kazput samples, in combination with significant pyrite concentrations (~0.32 weight %, Supplementary Table 2), could compromise the sulphate yields via laboratory-induced pyrite oxidation. Therefore, we developed a modified CAS extraction protocol, using the reducing agent hydroxylamine hydrochloride to inhibit pyrite oxidation during CAS extraction. Unfortunately, the hydroxylamine hydrochloride itself was found to contain sufficiently concentrated trace sulphate to contaminate the CAS sample target. The result of CAS extraction with hydroxylamine hydrochloride on Kazput carbonate sample gave a final yield of <2 ppm whole-rock CAS, all of which could be attributed to contamination from the hydroxylamine hydrochloride. Thus, it was concluded that CAS in these samples is too low for precise bulk S- and O-isotope measurements, and CAS extractions on these samples was not pursued further.

(Barite extraction)

L423-431: As a test for pyrite oxidation in our extraction technique, finely ground pyrite was stirred in the chelating solution for a week followed by attempted recovery of pyrite-derived SO_4^{2-} from the filtered solution, with no resulting sulphate yield, likely because the chelation of iron and other metal cations also served to prevent pyrite oxidation. In addition, we intentionally oxidized pyrite in the same laboratory-distilled water that is used in the barite extracting solution followed by measurement of the oxygen isotope

composition of the produced sulphate, which was isotopically distinct from that of the measured samples (Supplementary Fig. 7).

The d18O isotope results mentioned above are given in the supplementary fig. 7, as before.

Line 109: Again a few tweaks could clarify. “The temporal compilation of sedimentary sulphate d18O data demonstrates that that d18O values greater than 7‰ typify minerals that derive their oxygen from seawater (Figure 1b). Sulphate with d18O values below -15‰, however, ...”

This is now rewritten and does not exist as such.

Line 113: I am not sure this makes sense but I do not know enough to correct it. Do you mean from “high altitude, high latitude or glacial settings”? How it was written, I’m not sure what the “or likely” meant, I think that’s why I’m confused. Additionally, the following sentence is not straight forward, a reiteration is: “The lowest d18O value reported from the sedimentary record (-20.3‰) was obtained from a glacial diamictite deposited at 635 Ma during a Neoproterozoic Snowball Earth event.”

This is now much-expanded and rewritten in the discussion, as previously pasted in response to Reviewer 1, L281-301.

Line 113: Should “formation” be capitalised?

Line does not exist in new version

Line 117: Between the commas beginning after 500m, swap the text with: “and the available age constraints (2.31–2.21 Ga^{12,19}) demonstrate deposition was potentially synchronous with the glacial strata preserved within the Huronian Supergroup, Canada²⁰,”

Rewritten within section L281-301, as above.

Line 119: This is best clarified as “...was the oxygen source for the barites preserved in the Kazput Formation”.

Rewritten within section L281-301, as above.

Line 121: Again, some ‘word-smithing’ would make this snappier. See: “Discriminating between temperature-derived signals and true seawater d18O variability remains the principle challenge for interpreting ancient sedimentary d18O records. For example, Precambrian oceans are thought to be warmer than today with temperature estimates at 60 ± 20 °C, while carbonate-based seawater d18O estimates are as low as -10‰²¹. The average d18O value of the carbonates within the Kazput Formation is 18.4‰ (*versus* VSMOW)²², which, despite being typical for Siderian carbonates, is 10‰ lower than

present-day carbonates (Figure 1b). Thus, during the deposition of the Kazput Formation, the $\delta^{18}\text{O}$ of seawater may have been 10‰ lower than its contemporary value (0‰). Except for the extreme range of $\delta^{18}\text{O}$ values associated with the Neoproterozoic Snowball Earth events, the $\delta^{18}\text{O}$ values of sulphates fall between 7 and 18‰ regardless of whether they were the product of relatively cold contemporary oceans or their warmer ancient counterparts (Figure 1b).

Rewritten and expanded, with the above line of argumentation now simplified for clarity without changing the key point(s), as previously pasted here in response to Reviewer 1, L189-206

Line 134: The previous comment is subjective, grounded on my personal preferences. In fairness, it doesn't need to be implemented if you disagree. I do feel however that your data is not sufficiently introduced and the prose in the second half is difficult to follow! For example, you refer to "negative barite $\delta^{18}\text{O}$ values" but this is the first time the data has been introduced. I feel that the data should more comprehensively introduced before they are discussed. Thus, the second half of this paragraph could use a rework. Here's my attempt: "Barite isolated from the Kazput Formation is depleted in ^{18}O , with $\delta^{18}\text{O}$ values averaging -11.0 ± 10.8 (2s). Irrespective of the uncertainties surrounding Precambrian seawater temperatures and oxygen isotope compositions, these distinctively negative $\delta^{18}\text{O}$ values require a distinct and non-seawater-derived oxygen source. Sulphate produced during pyrite oxidation is enriched in ^{18}O by 10‰ relative to the parent water²³, suggesting that the $\delta^{18}\text{O}$ of the initial source waters ranged from -8 to -30‰. It remains to be resolved whether these barites sourced their oxygen from ^{18}O -depleted Palaeoproterozoic seawater ($\sim 10\text{‰}^{21}$), glacial meltwater (-34‰^{21}) or a mixture of the two. Regardless, both cases require a meteoric water source, implying a continental origin for this sulphate." These data are so light given the record. I would have thought this would have warranted more discussion. This seems like a significant finding

This - that "your data is not sufficiently introduced and the prose in the second half is difficult to follow!" - was taken into account in the newly revised version and we greatly appreciate this feedback.

The data is introduced and discussed differently now, with care taken to expand and clarify the discussion, incorporating the Reviewer's advice above. The key things are that the results and discussion is now organized with an expanded consideration of the **Context and origin of the Kazput barite** (subheading), followed by a discussion of larger implications in **Implications for surface environment response to oxygenation** (subheading). We feel that this new structure builds the argumentation within a more logical sequence that is easier to follow.

Line 148: I ran out of steam a little here. There's editorial changes that will help here but perhaps revisions to the structure and content is more pressing. This paragraph is very long and I have several questions/reservations concerning under developed claims that will need to be fortified as you move forward. These are:

This section has been rewritten for a more balanced appraisal, as already mentioned here.

1. **Line 155:** Why does the positive $\Delta^{33}\text{S}$ values indicate the precursor sulphate was derived from 2.5 Ga rocks? It looks like equal magnitude rocks are present in younger rocks. How do you preclude that this is not derived from atmospheric chemistry with the $\Delta^{33}\text{S}$ values derived from the atmosphere with mixing from terrestrially-derived ^{18}O depleted sulphate?

Noted. We state it as pre-GOE instead, as below:

L366-368: The all-positive $\Delta^{33}\text{S}$ values of Kazput barite imply this SO_4^{2-} originated in the oxidative weathering of pre-GOE sedimentary rocks.

This is now preceded by:

L358-366: Another possibility is that oscillating atmospheric oxygen levels after 2.5 Ga {Gumsley, 2017 #2809} permitted intervals of primary atmospheric S-MIF production and preservation as oxygen intermittently dipped below 0.001% of present levels. In any case, it remains difficult to clearly recognize these various scenarios because the signals they leave behind in sediments may be ambiguous. That oxidative weathering should be the main control on marine sulphate after atmospheric oxygenation seems intuitive, but as mentioned, even under the high atmospheric oxygen concentration of the modern Earth, surface weathering does not directly control marine sulphate S-isotope values in lieu of MSR.

...thus now we present the possibility of the $\Delta^{33}\text{S}$ signal being atmosphere-derived instead of a secondary weathering signal. As the discussion stands now, this part above comes after it is already discussed and established that the origin of the sulphate is most parsimoniously described as being from sulphide weathering on land. Therefore the context of mentioning this atmospheric origin is for the possibility of differentiating these scenarios - atmospheric or secondary weathering - in records in the future. As discussed, the evidence at hand seems to strongly indicate a secondary weathering origin for the signals of $\delta^{18}\text{O}$ down to -20‰ and $\Delta^{33}\text{S}$ up to +1.6‰ in the Kazput barite. The negative correlation between $\delta^{18}\text{O}$ and $\Delta^{33}\text{S}$, where the strongest $\Delta^{33}\text{S}$ is associated with the most negative $\delta^{18}\text{O}$ values, is at odds with an atmospheric origin of the $\Delta^{33}\text{S}$ signals and instead associates the strongest S-MIF with the most "landward" $\delta^{18}\text{O}$, as primary and secondary atmospheric sulfates are enriched in ^{18}O in comparison to the low $\delta^{18}\text{O}$ values found in meteoric waters on land.

2. **Line 158:** Your results being “at odds” seem like an overly strong statement that are introduced too early. I feel that more complete description of the S data is warranted first. Maybe it is not applicable to all Archean crust? It is also not clear what you’re trying to say in lines 158–160. Perhaps rewrite this to make the point explicitly clear and reorder. Basinal variability?

This part is indeed rewritten, whereas now it comes much later in the discussion, after a secondary weathering origin is already established for Kazput barite S and O-isotope signals. Furthermore, we now make clear that we expect that basinal variability plays a major role here, where in the final section of discussion we compare S. Africa and W. Australia in L318-344 and include a new figure (Fig. 4) to those ends - and this discussion also comes before the text mentioned above.

The context for the "at odds" statement is now as follows...

L366-375: The all-positive $\Delta^{33}\text{S}$ values of Kazput barite imply this SO_4^{2-} originated in the oxidative weathering of pre-GOE sedimentary rocks. Our results are at odds with the conclusion, based on modern weathering of old exposures, that the weathering of Archaean continents would yield sulphate with $\Delta^{33}\text{S}$ summing to nearly 0‰ {Torres, 2018 #3222}. Perhaps this discrepancy is because present-day rock exposures simply are not comparable in composition and weathering susceptibility to the sedimentary rocks exposed at Earth's surface in the Paleoproterozoic. Furthermore, small positive $\Delta^{33}\text{S}$ have indeed been measured in riverine sulphate from catchments in S. Africa and Ontario today {Asael, 2017 #3447}, which suggests that recycling of S-MIF signals at the Earth surface could occur at least locally.

3. **Line 163:** How do we know all the sulphate was being removed as pyrite and there was no re-oxidation? In your record, at least, you have trace barite demonstrating that pyrite burial was non-quantitative. Are there more up-to-date references. Perhaps some iron speciation data? Thinking about it the muted d34S data could be used as an argument but this contrasts with the south African record. Do you have thoughts on this difference? I would expect light pyrite if the sulphate if pyrite weathering was pervasive?

As above, the S. Africa and W. Australia records are discussed/compared now, which is very important with respect to considerations of a lack of re-oxidation in W. Australia as compared to S. Africa.

It is the super low d18O that requires sulphate capture as barite without it being subject to microbial cycling that would also typically leave a positive d34S-d18O correlation in sulphate, which is the opposite of what is observed here. That said, we now address the possibility of some microbial S-cycling effects on the Kazput barite S and O isotopes, as previously discussed here, with passages of manuscript text, in response to Reviewer 1.

Also, we did not do iron speciation but we do now include Fe/Al ratios in Fig. 2, and some (limited) pyrite concentration data in Supplementary Table 2. The Fe/Al ratios support that the water column was oxic. Thus it appears, based on the plentiful pyrite but lack of CAS in the Kazput Formation in our drill core, that the water column was oxic while the sediment porewaters were anoxic and basically a closed system - all of this is now discussed in the text, as below:

L254-266: The Fe/Al ratios in core TCDP3 are 0.33 on average, excluding enriched intervals >1 (Fig. 2, Supplementary Table 2), which is consistent with oxic water

columns whose Fe/Al are close to or less than average shale at 0.5 {Lyons, 2006 #1861}. The enrichments in Fe/Al and total sulphur observed in two muddy intervals below the carbonate are also likely pyrite (FeS₂) enrichments although this was not quantified directly, while the few measured pyrite concentrations are 0.32 weight % on average (Supplementary Table 2). The Fe/Al and total sulphur enrichments imply either transient water column anoxia, or movement of a chemocline during transgression-regression cycles. Anoxic sediment porewaters favoring MSR would also provide a flux of barium in sufficient concentration to precipitate SO₄²⁻ as barite at the sediment-water interface {Magnall, 2016 #3413}. Remaining porewater SO₄²⁻ would be quantitatively reduced to sulphide, via closed system MSR, with capture by iron to form iron sulphides (pyrites) during early diagenesis in the sediment {Philippot, 2018 #3192}.

4. **Line 172:** Are the two ARA lines in panels a and b -0.9 and -1.5? If so, it seems that the data plots on the steeper of the two. This is different from most Archean rocks and to my knowledge my 2017 dataset is the best for illustrating these discrete slope changes? What do you think, are there some of these -1.5 in the catchment?

Indeed there are data that fall on both the -0.9 and -1.5 $\Delta^{36}\text{S}/\Delta^{33}\text{S}$ slopes. To look at this further, we also now plot the $\Delta^{36}\text{S}$ data in Fig. 2 as $\Delta^{36}\text{S}/\Delta^{33}\text{S}$ ratios. New relevant discussion text is below, whereas we cite Izon et al, 2017 for its in depth look into $\Delta^{36}\text{S}/\Delta^{33}\text{S}$ systematics with respect to changing ARA slopes:

L303-316: A characteristic $\Delta^{36}\text{S}/\Delta^{33}\text{S}$ slope of -0.9 (Fig. 3a) is observed during the Archean and in the Kazput carbonate, however this slope steepens to near -1.5 in the siltstone facies above and below the carbonate (Fig. 2). A close link between methanogenesis and steeper $\Delta^{36}\text{S}/\Delta^{33}\text{S}$ slopes near -1.5 have been inferred in Neoproterozoic sediments with low $\delta^{13}\text{C}$ values {Thomazo, 2013 #3143}, whose systematics have been explored for their possible direct link to atmospheric conditions {Izon, 2017 #2830}. In the case of Kazput barite, its $\Delta^{36}\text{S}/\Delta^{33}\text{S}$ variations may simply reflect a strong control from oxidative weathering of pyrite in continental source rocks of various ages, and therefore the S-MIF signal is simply tracking the sulphur isotope composition of the rocks being weathered. The S-MIF signals of the Kazput barite show relatively small $\Delta^{36}\text{S}$ and $\Delta^{33}\text{S}$ ranges in comparison to the larger ranges observed in Neoproterozoic sulphur, where this limited range of S-MIF signals in the Turee Creek Group has been interpreted in favor of homogenization during the weathering of older, more isotopically variable sulphur which was previously formed in an anoxic atmosphere {Philippot, 2018 #3192}.

.... note that the above discussion text comes after establishing a weathering origin for the Kazput barite S and O isotope signals

5. **Line 175:** “Section” should be “succession”.

Line no longer in the text

6. **Line 175:** What is the significance of the 0.5‰ offset you describe here?

Line no longer in the text

Line 181: “from our sulphates can be omitted”.

We keep it for now (as "our barites, bolded below), as the sentence follows discussion of riverine sulphate thus we want to avoid confusion, instead of starting a paragraph as before, in context as below:

L273-278: However, the preservation of such a strongly riverine sulphate signal in marine sedimentary rocks, such as in the Kazput barite, is highly unlikely today due to the buffering by high seawater sulphate concentrations. It is unclear whether the sulphur isotope compositions from **our barites** reflect specific fractionations during sulphide oxidation on the continent or in an incipient oceanic SO_4^{2-} reservoir, but both require oxidizing conditions.

Line 183: “it is notable that” can also be pruned. I agree with the sentence requiring changes but perhaps expand and refer to the figure. I can’t help but think Fig 2 is beautiful yet under used. I would love to see this described more thoroughly and the implications discussed. The interesting thing here is that you seem to have systematic changes in the D33S and D36S data. What do you think this means? How does the slope evolve?

The "it is notable that" is now removed.

Also, Fig. 2 is now used more extensively in the discussion, and we feel that the reporting of $\Delta 36\text{S}/\Delta 33\text{S}$ ratios in that figure increases its usefulness, as well as the Fe/Al ratios now shown in that same figure. Both are discussed, as already mentioned.

Line 199: Would you expect to see a correction in Fig 3c?

Correction? Assuming that correlation was meant here, which refers to the Kazput barite $\Delta 33\text{S}/\text{d}18\text{O}$... and yes, this is now more thoroughly discussed and highlighted in the new version of Fig. 3, which now has two panels for $\text{d}18\text{O}$ cross-plotted with $\Delta 33\text{S}$ as well as $\text{d}34\text{S}$, whereas both $\text{d}34\text{S}/\text{d}18\text{O}$ and $\Delta 33\text{S}/\text{d}18\text{O}$ do show some correlations. This was already discussed in previous responses here as well.

Line 200: What about ground-level oxygenation a la Lalonde? Can there be oxygen production fuelling weathering beneath an anoxic atmosphere? Can you preclude this with your data? If not, then I think you should be more balanced in your arguments.

Key point, and noted. This is now taken into account, as below that includes the Lalonde reference, and as discussed already here. Again, a significant new aspect is the inclusion of the Stueken et al., 2013 estimated S flux time series in Fig. 1c.

L391-394: It has been suggested that SO_4^{2-} flux to the oceans increased in advance of the GOE due to oxygen production and its consumption by sulphide weathering on land {Stüeken, 2012 #513; Lalonde, 2015 #522} (Fig. 1c). Through their oxygen isotope records, it may be possible to constrain this early flux of SO_4^{2-} in the Neoproterozoic.

Line 201: My understanding is that the dataset features relatively few negative values and thus this could represent a bias within the record. Do you agree?

This refers to $\Delta^{33}\text{S}$ records, and yes we agree that there is a bias toward positive $\Delta^{33}\text{S}$ in the record. However, despite that bias, it is very intriguing that the Paris et al., 2014 Neoproterozoic sulfate paper, although putting forth the argument that Neoproterozoic sulfate carried positive $\Delta^{33}\text{S}$, also had very negative sulfate $\Delta^{33}\text{S}$ that was only given in its supplemental data. We included that data in the compilation Fig. 1a, whereas the most negative $\Delta^{33}\text{S}$ is their sulfate data from around 2.55 Ga. The point is that in this context, that strongly negative $\Delta^{33}\text{S}$ was found in sulfate even up to 2.55 Ga, we place our argument below. Despite the bias in the record, there is something more to it that has yet to be satisfactorily resolved, which of course is out of the scope of what we can do here.

L387-391: There are only positive, and no negative, $\Delta^{33}\text{S}$ anomalies reported from rocks <2.42 Ga (Fig. 1a) and this may mark the threshold when fluxes of SO_4^{2-} from the oxidative weathering of older Proterozoic sediments, with their predominantly positive $\Delta^{33}\text{S}$ anomalies, dominated the oceanic sulphur cycle.

Figure 1: Is TCDP and TC defined? American English in the legends. Green arrows in 1a are misleading please alter.

Fixed. In legends, sulfate changed to sulphate

Figure 2: Is nice there's trends in this data could be interpreted(?). Also, there seems to be a strong dependence on lithology. Is this true and important? Grey lines?

Yes this is discussed more thoroughly now, as already mentioned here. The grey lines are at the 0‰ value, which is hopefully clear, as is.

Figure 1: The caption is confusing move the definition of the lines upwards and the definitions down. BSR should be MSR as its not just bacteria check throughout. It would be nice to acknowledge the 1.5 slope was demonstrated convincingly in Izon et al., 2017. I'm not sure I follow the significance of the dashed lines. Please clarify.

This seems to refer to Fig. 3 (typo?). The caption was reworked, BSR changed to MSR throughout the whole text. Izon et al., 2017 is referenced in the main text discussion about the -1.5 slope (already mentioned here). Dashed lines (which denoted MSR trajectories) are now removed.

Line 288 on: Not checked but a quick scan reveals errors so please check.

Rewritten now.

Supplementary material: The way the supplement has been written is sometimes confusing. Additionally, the structure doesn't follow as well as it might. I have *attempted* to reword and shorten the text to make it flow better. I have also cut some of the descriptive text that people that are active in this field are intimately familiar with. These cuts will allow you to expand on the bits that are missing. In fairness, I don't expect the authors to take these edits verbatim unless, of course, they wish to do so. This was intended more as more of a guide. Anyway, I have appended the text in a track changes format; however, no doubt, that will get lost in the review system, so please encourage the authors to drop me an email if they feel this would be of benefit.

The supplementary text now only has a note about the compilation (Fig. 1) references. Now, the text the Reviewer refers to above is in the Methods section of the main text, and has been reworked, with detailed responses below.

My specific questions or issues that are perhaps worth clarifying are:

1. The first section (CAS Extraction) seemed strange and repetitive given that it didn't work. I tried to improve this.

Yes the CAS extraction didn't yield CAS, but as a negative result it is important to include.

2. Can you clarify the strength of the acid you used to liberate the CAS and decarbonate the samples?

Fixed:

L404-405: Dissolution of the carbonate-component by slow addition of 12M HCl, until no reaction was observed, liberated the CAS into solution

3. Can you add the concentrations of the CAS and blanks so others can potentially benefit from your efforts?

This is provided, as below, where the blank would have had at least 2 ppm sulphate attributable to the hydroxylamine hydrochloride. Real CAS concentration is unknown - it has to be lower than the blank.

L413-417: The result of CAS extraction with hydroxylamine hydrochloride on Kazput carbonate sample gave a final yield of <2 ppm whole-rock CAS, all of which could be attributed to contamination from the hydroxylamine hydrochloride. Thus, it was concluded that CAS in these samples is too low for precise bulk S- and O-isotope measurements, and CAS extractions on these samples was not pursued further.

4. I find that the expression of your delta ratios is weird. Can you check that I've not messed them up? I try to follow the Coplen paper.

We don't use a linear capital delta definition, so the delta definitions are left as-is.

5. Can you specify how many standards you measured to get your precision. Also, can you include the various ratios that you obtained relative to the relevant international standard. I don't doubt your data is good but reporting standards allows us all to see where we're all at.

The precision for the d18O comes from the standards measured during the analysis session, where the standards are now reported in the supplemental. For the S-isotope data, the standards measured during sample analyses sessions are given, however, the reported precision is not based on these standards alone, it is based on long term precision on the standards.

6. The data sources paragraph was also hard to read. I've tried to amend. Please consider these changes.

Suggestions appreciated. Rewritten now.

7. At least in my image the quality of the EDS images in Figure S1 was *bad*. I couldn't make out what the spectral peaks are! Can these be resized or tweaked to make them look better? Any change the French can be made English? The latter isn't a big deal...

Now fixed and split into two figures - Supplementary Figs. 3 and 4, so they are resized and tweaked to have them looking better. Spectra are now larger, higher resolution, thus more visible. French changed to English now as well.

8. Figure S2. Perhaps this doesn't need to be in the MS or clarified but some random Qs: Why is there no data from this sample? No barite? Seems odd to include a section from which there is no corresponding isotope data. What are the textures in the oldest third of the slide? Is this slumping? Is that pyrite crossing the depositional fabric, what stage do you think that is does this have any effect on your interpretation? Is a section from which there's data more appropriate?

Note that figure S2 is now Supplementary Fig. 6. This thin section is not from the same set of samples used for CAS & barite extractions, for which we needed >200g each sample. This is why there is no data that corresponds to this thin section - it was taken separately for that purpose.

9. Can you make the tables easy to use? I know the publishers make this difficult but if you can get an excel file at a supplement that would be awesome. We should push publishers for this!! I also plotted some of the data and the decimals in the averages were slightly out. This could be a rounding error but worth checking.

Absolutely agree! Would love all supplemental data to be offered as excel tables.

As for the rounding error, unsure, but perhaps it comes from different definitions used for capital deltas?

10. Why did you do it like that? I would have done the weight loss via decarbonation, which itself is not perfect. I'd ask you to note in the caption these are top-end estimates. Also, there's no mention in the methods of the SEM nor the carbonate abundance. Perhaps a few lines for completion?

This refers to...

Supplementary Table 2 caption: The reported data for CaO oxide weight percent is added to loss on ignition (assumed to entirely represent CO₂) to give the percentage of carbonate shown, as carbonates were not specifically measured for this study. This accuracy of this determination of carbonate weight percent was confirmed gravimetrically on one sample (CAS-23) by weighing dry sample powder, decarbonating it in acid and drying, then weighing the remaining dry sample residue.

Above, we state why carbonate % was calculated this way. Normally, of course, carbonate weight percent is measured at the same time as carbonate δ¹³C, but this was not done for this study. We did not pursue CAS extractions on samples since early results showed that there was negligible CAS. Then, during barite extractions, after the decarbonation step the sample was washed in triplicate and progressed directly to the extraction step without drying it, so dry sample residues were not weighed.

11. There is no in-depth discussion of the sample cores, the regional geology, where they are archived *et cetera*. This is necessary background, especially the age constraints, and could be added up-front in brief or here in more detail. A map is useful, also.

Geologic maps added in Supplementary Fig. 1 - suggestion appreciated. Much more geologic context has been added in general, in the main text, as previously discussed here.

12. Please check the references. There's a few issues with their formatting, especially with weird characters that are likely issues with super- and sub-scripts

Checked and fixed.

Reworked Supplementary Information:

CAS Extraction:

Carbonate-associated sulphate (CAS) extractions were attempted on samples from drill core 3 from the Turee Creek Drilling Project (TCDP) at the Laboratoire Géosciences Occéaniques at IFREMER, Plouzané, France, following a CAS extraction sequence initially verified by Wotte et al. Here, crushed and powdered carbonate samples were leached in triplicate using a 10% NaCl solution, removing any soluble sulphate phases. Dissolution of the carbonate-component using HCl then liberated the CAS, which was then precipitated as BaSO₄ upon addition of a supersaturated BaCl₂ solution. Low CAS yields (<X ppm)

from the Kazput samples, in concert with relatively high pyrite abundances (>X ppm or Wt.%), could have compromised the isotopic integrity of the CAS via laboratory-induced pyrite oxidation. Consequently, to test this hypothesis, we modified the CAS extraction protocol, using a reducing agent, hydroxylamine hydrochloride, to inhibit pyrite oxidation during CAS extraction. Unfortunately, the hydroxylamine hydrochloride itself was found to contain trace-levels of sulphate that were sufficiently high (Xppm) to dominate the sulphate recovered as BaSO₄. Accordingly, it was concluded that the CAS content of these samples was too low to enable accurate and precise bulk S- and O-isotope measurements and CAS extractions were not pursued further.

Barite extraction:

Barite extractions were performed on an ~ 100g aliquot of homogenised drill core using a modified barite purification technique (the DDARP method²) that was originally designed to purify sulphate for triple oxygen isotope measurements. Again, the barite extractions were completed at the Laboratoire Géosciences Occéaniques at IUEM. Briefly, drill core samples were manually crushed in a tungsten-carbide piston chamber before powdering in an agate ring and puck mill. Sample powders were decarbonated in HCl-acidified solution, rinsed in distilled water, and then stirred for 3 days in a mixed 0.05 M Diethylenetriaminepentaacetic acid (DTPA) and 1 M NaOH solution to dissolve barium sulphate. The supernatant was then separated from the sample residue by vacuum filtration. After acidification (pH < 2) with HCl, the barite-hosted sulphate was precipitated as BaSO₄ via the addition of a supersaturated BaCl₂ solution. Co-precipitation of silicates dissolved in the DTPA solution renders the precipitated BaSO₄ impure, requiring an additional purification step. Here, residual DTPA is removed via repeated rinsing (n=3) with distilled water followed by centrifugation and decanting of the supernatant. After heating the precipitate at 70 °C overnight with 2 M NaOH, the sample was again centrifuged and the supernatant decanted. The BaSO₄ was once again redissolved via overnight agitation in a mixed 0.05 M DTPA and 1 M NaOH solution and precipitated as BaSO₄ via the addition of acidified BaCl₂ solution. The final precipitate was washed in triplicate and dried at 70 °C overnight ready for isotope analysis.

Oxygen isotope measurements:

The oxygen isotope composition of the purified barite samples were measured in duplicate using an Elementar vario PYRO cube elemental analyser interfaced with an Isoprime 100 mass spectrometer in continuous flow mode at the University of Burgundy in Dijon, France. Oxygen isotope data are expressed in delta notation, $\delta \equiv R_{\text{sample}}/R_{\text{standard}} - 1$, where R is the mole ratio of ¹⁸O/¹⁶O and reported in per mille (‰). The $\delta^{18}\text{O}$ data are reported with respect to the international standard Vienna Standard Mean Ocean Water (VSMOW). Analytical errors are $\pm 0.4\text{‰}$ (2 σ) based on replicate analyses (n = X) of the international barite standard NBS-127, which was used to bracket unknowns.

Quadruple sulphur isotope ($\delta^{34}\text{S}$, $\Delta^{33}\text{S}$, & $\Delta^{36}\text{S}$) measurements

To determine for quadruple sulphur isotope combustion of the purified barites, an additional wet chemistry stage was required. Here, at the Institut de Physique du Globe de Paris, Paris, France a 3 mg aliquot of purified barite was converted to Ag₂S via sub-boiling distillation with HCl, HI, and H₃PO₄, reducing the sulphur to H₂S,

which was ultimately trapped upon reaction with silver nitrate. The precipitated Ag₂S precipitate was washed three times in distilled water and dried in an oven overnight ready for fluorination and sulphur isotope analysis.

Approximately 3mg of Ag₂S was converted SF₆ gas via overnight reaction under an excess of F₂ gas at 250 °C in nickel bombs. Sulphur Hexafluoride was then purified via cryogenic trapping and separation by gas chromatography before introduction to the mass spectrometer—a Thermo Finnigan MAT 253 running in dual inlet mode. Conventional delta notation is used to report ³⁴S data, where $\delta \equiv R_{\text{sample}}/R_{\text{standard}} - 1$, where R is the mole ratio of ³⁴S/³²S and reported in units per mille ‰. Minor isotope data, ³³S and ³⁶S are reported in capital delta notation, where $\Delta^{33}\text{S} = \delta^{33}\text{S} - ((\delta^{34}\text{S}/1000 + 1)0.515 - 1)$ and $\Delta^{36}\text{S} = \delta^{36}\text{S} - ((\delta^{34}\text{S}/1000 + 1)1.89 - 1)$. The $\delta^{34}\text{S}$, $\Delta^{33}\text{S}$, & $\Delta^{36}\text{S}$ data are reported with respect to the Vienna Canyon Diablo Troilite (VCDT) international standard, with analytical errors of $\pm 0.1\text{‰}$, $\pm 0.01\text{‰}$ and $\pm 0.2\text{‰}$ (2σ), respectively, as estimated via replicate analyses of the international standard IAEA S-1 ($n = X$).

Data Sources in Figure 1

The temporal record of $\Delta^{33}\text{S}$ (Figure 1a) is predominantly derived from Havig et al. (2017)⁶. Data sourced from non-sedimentary processes (i.e., hydrothermal⁷ and thermochemical sulphate reduction⁸) have been removed and the $\Delta^{33}\text{S}$ values originally from Ref. 9 have been corrected after inverted substitution for $\delta^{33}\text{S}$ data⁶. Data from the Boolgeeda formation¹⁰ have been adjusted for the new age constraints and correlations presented by Philippot et al. (2018)¹¹. The temporal compilation of sedimentary $\delta^{18}\text{O}$ data (Figure 1b) is derived from chert, carbonate, and shale, which was published by Bindeman et al. (2016)¹² and augmented with additional sulphate $\delta^{18}\text{O}$ data¹³⁻¹⁶.

Figure S1. Scanning electron micrographs and EDS spectra of barite isolated from sample CAS-9 (96.53 m). To concentrate barite for imaging, the silicate fraction was removed from the decarbonated sample residue using hydrofluoric acid and the residual material was dried prior to mounting.

Figure S2. A scanned image of petrographic thin section from 107.8 m depth in the Kazput Formation, TCDP core 3. The approximate field of view is 20 x 40 mm. The direction of younging is parallel to the page with the youngest material at the top. The fine-grained and finely-laminated texture seen in this thin section typifies the central shaley carbonate discussed in the main text and the TCDP drill core 3 as a whole. The black rectangular shape in the centre-left is a euhedral pyrite.

Figure S3. The $\delta^{18}\text{O}$ of dissolved sulphate in tap water, sulphate derived from the deliberate oxidation of pyrite in distilled laboratory water, and the average for barium sulphates extracted from TCDP core 3. The sulphate in tap water was precipitated from municipal water in the laboratory in Plouzan , France. The sulphate derived from the oxidation of pyrite was generated by stirring crushed and powdered pyrite for >1 day in laboratory distilled water, using the same distilled water source that was used during the barite extractions.

[Table 1 OK]

Table S2 (% carbonate). Calculated carbonate abundances for TCDP core 3. These estimates combine CaO oxide abundances (Wt.%) and loss on ignition (assumed to entirely represent CO₂) data to approximate carbonate abundance. Initial X

measurements were performed by ACTLABS using alkaline fusion total digest of sample powders.

The above suggestions from Reviewer 2 were incorporated in the newly rewritten Methods section and much appreciated.

REVIEW 3

I very much appreciate the goals of this paper and the approach taken. This paper is absolutely appropriate for publication in Nat. Comm. The potential impact will be high. The motivations and methods are clever and robust. I hope it will be accepted. But there are things to consider first.

(1) The choice of samples from the Turee Creek basin means that there is a wealth of previous work that elevates the chronological control, environmental context, and broad reader interest. That said, we aren't given much environmental detail, at least not in the primary text.

In the previous response to Reviewer 1 here, the new important changes to the text were identified that give a deeper and broader context with respect to the Turee Creek Basin (text from manuscript pasted previously here "big picture" background: L108-117, detailed geologic context: L133-148) and how the controls on its sulphur cycle (that are identified in this work) fit within a larger global context as compared to the S. Africa sulphur records around the time of the GOE interval (L318-344) with a new Figure 4 added for this latter discussion. In addition, new supplementary figures have been added for deeper background context, as discussed.

The chronological controls are reported with more precision, as detailed in the response to Reviewer 2 (text pasted previously, for example L64-67)

The connection to glaciation is only loosely established (lines 116-119): "the Kazput formation overlies the Meteorite Bore Member glacial diamictite by ~500 m, (and therefore) it is plausible that glacial meltwater was an oxygen source for Kazput barite." Does 500 m of stratigraphic separation really establish an environmental linkage to glaciation? I'm pressing here because the potential role of glaciation is a common theme throughout the paper, yet the purported close temporal and spatial relationships are at best anecdotal. More generally, a better environmental context (links the oceans, glacial runoff, deposition under restricted shelf/continental conditions, etc.) will give the reader better access to evaluating the likelihood of local versus global controls and signals.

As discussed, with manuscript text (L281-301) pasted in prior responses to Reviewer 1 here, a section of detailed discussion on the potential connection to glaciation is now added.

(2) Line 94: We are only told that trace barite is present and that this observation is

consistent with low sulfate levels (in the local setting? oceans?). I was hoping for more discussion about the barite (in the primary text, given the importance). For example, can we be sure that the barite reflects syndepositional, low temperature conditions? Might hydrothermal process have played a role, which would impact the oxygen isotope data? Could later remobilization/overprint scenarios have been important? Simply put, the sedimentological context of the barite and specifically arguments supporting its primary, low-T origin are important but missing,

In response, we have added an extensive new section of discussion that evaluates different possibilities (e.g., in the lab, secondary low-T, secondary high-T, detrital) for the Kazput barite origin, **Context and origin of the Kazput barite**, L132-278 in the main text, as previously discussed here.

(3) The basic premise of the paper, which I like very much, is that combined S and O oxygen isotope approaches allow for recognition of oxidative weathering of continental sulfides, with the S data capturing recycling of older, Archean sulfides bearing a MIF-S signal. In other words, the S signal is inherited, and oxygen's is not—capturing instead the meteoric waters involved during oxidation of the continents at the time of deposition of the barite—presumably beneath an atmosphere that, by then, was too oxygenated to yield a preserved MIF signal. The possibility of recycled/inherited MIF-S signals in younger rocks deposited beneath a relatively O₂-rich atmosphere is important. Reinhard et al., as the authors note, argued for such processes and suggested that MIF-S recycling could complicate our interpretations of those isotope records. Torres et al. challenged that assertion, and now Killingsworth et al. shed new light supporting the possibility of recycling/inheritance as captured at least locally. This is important. However, two things come to mind: (i) This overarching story and the related implications could be spelled out more clearly in the paper. Now, only insiders will truly grasp the potential importance of the paper in terms of our ability to constrain the timing and pattern of initial oxygenation, etc.

Intensive efforts, in rewriting and reorganizing the text, were made to address these concerns of critical importance. The result is a deeper evaluation that is also more balanced than before. Though discussed in detail in prior responses here, we highlight that..

1) the new structure of the paper presents the important ideas in a more logical order that is now hopefully more accessible to those who have not followed the important links between oxygenation and its interpretation via the sulphur record thus far,

2) we lean less heavily on the global seawater sulphate sulphur isotope "memory effect" of Reinhard et al., 2013 simply because it appears that we have a story of more localized controls - in Turee Creek there are important regional/local controls that have affected its S and O isotope record of the barites we report - something that is made clear in comparison with the S isotope records around the GOE from S. Africa (main text pasted earlier here, L318-344).

(ii) I am not convinced that the authors have made a case for their data requiring recycling of older sulfur beneath a relatively well oxygenated (post-GOE) atmosphere. Evidence for oxidative weathering of continental sulfides in combination with a MIF-S signal could, as the authors suggest, reflect recycling of older S retaining its inherited MIF behavior. On the other hand, the combination could rather reflect sulfide oxidation under very low O₂ conditions—low enough to preserve syndepositional MIF-S processes in the atmosphere. This second possibility was suggested by Reinhard et al. in 2009 in their take on the Mt. McRae whiff-of-oxygen story. Critically, the Reinhard 2009 paper argued that sulfate was sourced to the oceans by weathering even under very low, MIF-sustaining (whiff) oxygen conditions. If this latter model is correct, Killingsworth's notion of oxygen isotope signals reflecting continental weathering could be an Archean, pre-GOE feature as well. They just don't have the data to argue one way or the other. It is certainly possible that oxidative weathering was an early feature only enhanced at the GOE and sulfate was sourced to the ocean, even before the GOE, by oxidative weathering in addition to atmospheric sources.

The above speaks to the tantalizing possibilities of oxygen isotope constraints on sulphur cycling before the GOE that is now taken into account in the newly revised manuscript.

We do not hang our arguments on whether the atmosphere was well oxygenated at ca. 2.31 or not, but instead simply make the case for weathering control on the S-MIF signals in Turee Creek Basin.

Now, the origin of the S and O isotope signals in Kazput barite is more critically evaluated, with different possibilities explored. Although we come to the same conclusion (as previously discussed in detailed responses here) - these signals are best explained by sulphide weathering on land - it is now supported within a stronger framework. Furthermore, although the above-mentioned Reinhard et al., 2009 is not cited, we do suggest (supported by other references) the possibility that O-isotope signals in sulphates from before the GOE may be able to indicate oxygen production (O₂ whiffs) that was consumed on land, as below in the final lines of the paper.

L391-394: It has been suggested that SO₄²⁻ flux to the oceans increased in advance of the GOE due to oxygen production and its consumption by sulphide weathering on land {Stüeken, 2012 #513; Lalonde, 2015 #522} (Fig. 1c). Through their oxygen isotope records, it may be possible to constrain this early flux of SO₄²⁻ in the Neoproterozoic.

As mentioned above, Fig. 1c is a new figure panel that shows O₂ concentration and the S flux curves from Stüeken et al., 2012 - seeing them in the same figure as the Δ³³S and δ¹⁸O time series compilations emphasizes the points above.

We suffer now because the authors don't have earlier data, and we don't really know the possible magnitude/impact of the sulfate-oxygen signal (that is, possible expressions in the global ocean). The carbonate data certainly don't support the idea

of very low $\delta^{18}\text{O}_{\text{H}_2\text{O}}$ for global seawater. And I struggle to imagine large amounts of sulfate forming in those water. (Are exchange reactions likely to have dominated?) More could be said about what was possible/likely isotopically for the early ocean and what the barite data most likely record and why we should care.

I am rambling. My point, though, is that we need more discussion on the likely local versus global signals and, most importantly, what these data really tell us about the GOE in comparison to likely sulfate sources and their isotopic signals during the earlier Archean. Arguments like those in this paper about presumed temporal shifts despite a lack of earlier data (a pre-shift baseline) are problematic.

Agree. As has been discussed here, the new version of the text more clearly describes our stance that the Turee Creek Basin must have had a regional control that was different than, for example, the Transvaal Basin in S. Africa. Thus, it is not implied that the low $\delta^{18}\text{O}$ of the Kazput barite is representative of the ~ 2.31 Ga ocean water, as it appears to capture a local, possibly glacial meltwater, signal. Further, we state that the (unmeasured) sulphate $\delta^{18}\text{O}$ compositions from S. Africa from around the same time will likely not have this same low $\delta^{18}\text{O}$ signal, which is implied by the different S isotope signatures in W. Australia versus S. Africa around the time of the GOE (Fig. 4). We paste the relevant text again here:

L318-344: Sulphur isotope distributions compared between S. Africa and W. Australia (Fig. 4) reveal preservational controls on S-MIF during the GOE interval, ca. 2.45 to 2.2 Ga. The depositional settings were intracratonic basins for both S. Africa {Bekker, 2004 #268} and W. Australia {Van Kranendonk, 2015 #2451}, near the equator {Gumsley, 2017 #2809}. Despite similarities between their basins, S. Africa and W. Australia sulphur isotope records are distinct for the GOE period. In W. Australia, including Kazput barite, the spread of sulphur $\delta^{34}\text{S}$ data are mostly distributed around one mode, slightly above 0‰. In contrast, data from S. Africa show a bimodal ^{34}S distribution more typical of open-system MSR, with isotopically lighter sulphur preferentially incorporated into sulphide with negative $\delta^{34}\text{S}$ and sulphate from CAS records with average values around +25‰. These different behaviors are manifested in their respective $\Delta^{33}\text{S}$ distributions, where the range from each basin is similar, but data from S. Africa indicate a strong source of mass-dependent sulphur ($\Delta^{33}\text{S}$ near 0‰) compared to two sources of sulphur with S-MIF at around +0.9‰ and +1.6‰ in W. Australia. The Kazput barite $\delta^{34}\text{S}$ - $\Delta^{33}\text{S}$ relationship of ^{34}S -enrichment versus the ARA (Fig. 3b) is matched in pyrite {Philippot, 2018 #3192}. Such enrichment in S-MIF-bearing pyrite has previously been observed in microbialitic carbonates where carbonate precipitation effectively sealed off the sediment from the water column, promoting closed-system MSR {Ono, 2009 #3388}. Stromatolites and carbonate are indeed prevalent throughout the Turee Creek Group, and especially in the Kazput Formation carbonates. The $\delta^{18}\text{O}$ compositions of CAS from S. Africa are yet to be measured, but it is likely that they are relatively enriched in ^{18}O as compared to the Kazput barite. As discussed here, the open system MSR suggested by S isotope evidence from S. Africa would also exert strong control on the $\delta^{18}\text{O}$ of SO_4^{2-} . In order to preserve the persistent S-MIF signals recorded throughout the Turee Creek Group succession, as compared to S. Africa, in W. Australia it seems that closed system MSR, within the sediment, was required. Further, we suggest that low

global marine sulphate concentrations and an intracratonic basin setting made the capture of weathering input fluxes, and their SO_4^{2-} isotope compositions, possible in the Turee Creek Basin.

(4) Line 163: “Sulphide generated at the sediment-water interface would have experienced little to no re-oxidation in poorly oxygenated bottom waters.” I agree, and the following speaks to this quantitatively: Oxidative sulfide dissolution on the early Earth, CT Reinhard, SV Lalonde, TW Lyons, *Chemical Geology* 362, 44-55 (2013).

The above reference is cited in the text, but not for this line (it was removed). The text has been rewritten because, in our more critical evaluation, while porewaters in Turee Creek Basin may have been anoxic, the water column may have been oxic. Fe/Al data (newly added to Fig. 2) supports this. Furthermore, oxic porewaters but a (relatively) closed system in porewaters is more consistent with the S isotope data in previously reported Turee Creek, W. Australia pyrites from Philippot et al., 2018. The implication of closed system MSR in Turee Creek is discussed in the text pasted in the answer just above (comparison with S. Africa).

The new discussion of oxic water column, but closed porewaters:

L254-267: The Fe/Al ratios in core TCDP3 are 0.33 on average, excluding enriched intervals >1 (Fig. 2, Supplementary Table 2), which is consistent with oxic water columns whose Fe/Al are close to or less than average shale at 0.5 {Lyons, 2006 #1861}. The enrichments in Fe/Al and total sulphur observed in two muddy intervals below the carbonate are also likely pyrite (FeS_2) enrichments although this was not quantified directly, while the few measured pyrite concentrations are 0.32 weight % on average (Supplementary Table 2). The Fe/Al and total sulphur enrichments imply either transient water column anoxia, or movement of a chemocline during transgression-regression cycles. Anoxic sediment porewaters favoring MSR would also provide a flux of barium in sufficient concentration to precipitate SO_4^{2-} as barite at the sediment-water interface {Magnall, 2016 #3413}. Remaining porewater SO_4^{2-} would be quantitatively reduced to sulphide, via closed system MSR, with capture by iron to form iron sulphides (pyrites) during early diagenesis in the sediment {Philippot, 2018 #3192}. The lack of CAS-sulphate from the Kazput carbonate further attests to nearly quantitative reduction of SO_4^{2-} in porewaters.

(5) Line 33: In light of all these comments, how do the data convincingly “indicate a transitional oceanic sulfur cycle dominated by oxidative weathering in the aftermath of atmospheric oxygenation?” I suspect this is true, but we need a more convincing and/or balanced perspective aimed at the key implications and arguments for a transition—with a clearer statement on the major conclusions relative to previous interpretations, the data before and after the GOE, etc. This is a wonderful data set. Another round of revision will likely elevate the impact. I hope to see it in print soon.

The quoted statement above is no longer, and we have backed off on the idea that our data speaks to the global "transitional oceanic sulfur cycle".

While we agree that the data do not necessarily speak to a global sulphur cycle, they do shed light on a transitional period for the global sulphur cycle that may have strong local controls. For example, the comparison between W. Australia and S. Africa S isotope records reveals why sulphur isotope anomalies may be preserved in W. Australia for longer (up to 2.31 Ga or longer) than in S. Africa (up to 2.33 Ga). Although we do not have, or claim to have, all the answers to exactly what is happening for the 2.5-2.3 Ga global sulphur cycle, what we offer is a way to constrain the effects of weathering on sulphur records for even before the GOE, >2.5 Ga, which is hopefully useful to the community, now presented with more balance and clarity.

Reviewer #2 (Remarks to the Author):

Since submission, the paper has received a thorough review and, in my opinion, most of the original reviewers' concerns have been adequately incorporated, rebutted and/or answered. Consequently, I believe content of the manuscript has been improved. Unfortunately, however, in dealing with these comments the message of the manuscript has been lost in places. Nevertheless, this often not major and typically concerns structure, word choice(s) and/or punctuation. Given I expect the remaining reviewer will reach largely the same conclusions, I have attempted to streamline their text in a track-changes format, including specific questions in the margin. Please forward this to the authors with the caveat that I attach these predominantly as a guide: I do not expect the authors to adopt these changes piecemeal, and I definitely do not want to change/influence the authors' interpretations or science. These changes, instead, show how I struggled and where their message is likely to be lost or confused by the more general audience they target. The questions I have made in the margins I would like to see incorporated into the manuscript but they are generally minor and likely will not need a formal review.

In summary, I believe the efforts of Killingsworth and colleagues have culminated in a largely credible, timely and provocative manuscript, that will be both well-read and -cited. Moreover, it encourages us to question our paradigms and search for evermore imaginative approaches to outstanding questions. I admit that I am not an expert in sulphate-oxygen isotope systematics but their approach is interesting and something that, moving forward, I aim to include in my own examination of Palaeoproterozoic rocks.

As always, I leave the ultimate decision of the manuscript to the editors but, personally, I am happy to support the ultimate publication of the appraised manuscript with some minor modification. I will take this opportunity to thank all involved for their patience and invite direct questions to expedite the outcome.

NOTE: Reviewer remarks in normal text, author responses in blue, *inserted manuscript text in blue italics*

Reviewer #1 (Remarks to the Author):

This is my second review of the submitted manuscript by Killingsworth and colleagues and I think it has significantly improved. First of all, I thank the authors for taking into account my comments and their careful response. At the request of the editor, I have also assessed the comments of reviewer 3.

I think this new version is much better. Providing additional geological context, scientific background and a more "in-depth" discussion make the manuscript more relevant to a broad audience review and will, I hope, increase the impact of the paper. It was much easier (and interesting) to read and the conclusions are much clearer to the reader. Not all questions are fully answered but this is also part of the paleoenvironmental game, especially on such old events.

I have no problem with their interpretation of seawater $\delta^{18}\text{O}$ and the idea of multiple phases for barite formation in an anoxic basin is an option. However, I still feel that the observation of the exact same S-isotope values between pyrite and barite is surprising. It is a clear improvement that this question is now openly addressed in the paper.

We appreciate the important points previously raised by Reviewer #1 that have improved this manuscript thus far, resulting in a more thorough assessment of the S and O isotope record in the barite and its paleoenvironmental context.

It is indeed surprising that the S-isotope values match so well from the bulk pyrite (as reported in Philippot et al., 2018¹) and the bulk barite extractions (from this study). Proceeding from that observation, we tested different possibilities that could explain this match, as previously elaborated in the main text.

We add a new sentence to confront the issue of S-isotope match head-on:

L168-171: Because, under various conditions, it is possible for low temperature redox reactions to proceed without S-isotope fractionation between sulphate and reduced sulphur phases², by itself, the S-isotope match between Kazput barite and pyrite cannot be used to diagnose a specific process.

However, quantitative pyrite formation does not necessarily imply that all pyrites have the same value; just that "bulk" pyrite has the same isotopic composition as the initial sulfate - but some pyrite grains must have different values, depending on when they form. How representative of "bulk" pyrite is the sample analyzed by Philippot et al.? What is the variability within a sample or a grain? This might go beyond the scope of this paper, but future in-situ investigation could be of interest and could help strengthen the "quantitative" hypothesis.

Agree that such future in-situ investigation like comparison of barite and pyrite in the same samples via SIMS S-isotope measurement would be worthwhile to understand their relationship in detail. As it stands, however, our sulfate study was focused on bulk measurements from the beginning, where carbonate-associated sulfate (CAS) was the original target but CAS was not extractable while barite was. Furthermore, detection of barite was challenging due to its trace occurrence as microns-scale grains - while bulk chemical extractions overcome this challenge, it will likely make SIMS work on the barites difficult.

Although the present study focuses on bulk barite, the bulk versus SIMS S-isotope data reported in Philippot et al., 2018 appears to show good agreement between the data despite their different scales of observation. However, as can be expected, the pyrite SIMS data do show greater variation. Here is Figure 2, showing pyrite S-isotope data from Philippot et al., 2018, where the upper core is the same as used in the present study; bulk data is shown in red, averages of SIMS data in white dots, and individual SIMS values within grey envelopes (stratigraphic column view) or grey points (cross plots):

Fig. 2 Sulphur content and $\delta^{34}\text{S}$ and $\Delta^{33}\text{S}$ profiles of sulphides in the Turee Creek drill cores and associated $\delta^{34}\text{S}$ - $\Delta^{33}\text{S}$ and $\Delta^{33}\text{S}$ - $\Delta^{36}\text{S}$ compositional diagrams. Red dots, bulk analysis; white dots, mean value of in situ analysis for each sample; grey dots, individual spot analysis using the SHRIMP-SI (error bars are standard deviation of 1σ , see Methods). The grey shaded areas in the $\delta^{34}\text{S}$ and $\Delta^{33}\text{S}$ profiles represent the range of individual spot analysis for each sample. The two blue-shaded areas labelled (i) below 130 m depth in T1 and (ii) below 145 m depth in T3 are discussed in the text. ARA, Archaean Reference Array. MDF, mass-dependent fractionation of sulphur isotopes. The stratigraphic section of the Turee Creek Group on the left is modified after Van Kranendonk and Mazumder¹⁹. Re-Os age of 2312.7 ± 5.6 Ma is from this study. Other age constraints are labelled accordingly. Abbreviations associated with the stratigraphic section are as follows: WR, Woongarra Rhyolite, B Fm, Boolgeeda Iron Fm., Kun, Kungarra Fm., MBM1 and 2, two diamictite horizons of the Meteorite Bore Member, KfM, Koolbye Fm., Ka, Kazput Fm., BRQ, Beasley River Quartzite, CSB, Cheela Springs Basalt

As presented in our manuscript, the "quantitative" hypothesis for pyrite formation is not only supported by the barite and pyrite S-isotope match, but also in comparison between

histograms of pyrite and sulfate S-isotope data from W. Australia vs S. Africa (Fig. 4). Even if the pyrite alone is considered, it is striking that in W. Australia the $\delta^{34}\text{S}$ mode is near zero while in S. Africa the values are more evenly distributed, with the maximum density being near -30‰ for the pyrite. The new version of Figure 4, with the dataset from Luo et al., 2016³ now included, has not changed appreciably from the previous version (i.e., the data distributions still show the same patterns) and is pasted below.

In addition, this hypothesis does not explain the small discrepancy (systematically slightly above error bars in D36S values presented in the previous version of the manuscript) between pyrite and barite.

Here is the previous version of Figure 2 that is referenced by Reviewer 1 above, which shows the $\Delta^{36}\text{S}$ data:

...and the current version of Figure 2 that reports $\Delta^{36}\text{S}/\Delta^{33}\text{S}$ ratios instead because using the ratio is more informative for the discussion:

To be fair, the $\Delta^{36}\text{S}$ of barite and pyrite do not systematically differ throughout the studied interval but it is certainly true that between ~ 130 to 110 m the pyrite show some (apparent) ^{36}S enrichments versus barite. The data are now represented as $\Delta^{36}\text{S}/\Delta^{33}\text{S}$ ratios (latter, and current, figure version above) instead of $\Delta^{36}\text{S}$ alone (former figure version above), but the $\Delta^{36}\text{S}$ difference between pyrite and barite can still be observed in the new version of the figure above.

We emphasize two points about comparing $\Delta^{36}\text{S}$ between sulfate and sulfide here, with the first point perhaps being most significant:

1) $\Delta^{36}\text{S}$ measurements are notoriously difficult because ^{36}S is the least abundant stable S isotope and there are significant isobaric interferences due to contaminants - a problem that is recognized in the lab (and is also attested to in the significantly larger error bars for ^{36}S measurements versus ^{33}S and ^{34}S) but has yet to be fully resolved. Therefore, while there is remarkably (and, as noted above, unexpectedly) good agreement between the S isotope results for bulk barites and pyrites that were measured over a year apart, some small, though indeed significant, difference in $\Delta^{36}\text{S}$ between pyrite and barite for part of the dataset might be due to unrecognized differences in running conditions between the pyrite and barite sessions, or in the chemical conversion of the pyrite and barite phases to the Ag_2S sulphur phase used for multiple S isotope measurements, but this is unclear.

2) Shen et al., 2009⁴ explored multiple S isotope differences between barite and pyrite from within North Pole, Australia, Archean barites where $\Delta^{36}\text{S}$ differences were usually observed in scenarios alongside significant $\delta^{34}\text{S}$ differences. They showed 2 scenarios of sulfide $\Delta^{36}\text{S}$ being higher than its precursor sulfate, and 2 scenarios where sulfide $\Delta^{36}\text{S}$ was lower than its precursor sulfate. In the case of our barites, the co-occurring sulfide (pyrite) is higher in $\Delta^{36}\text{S}$. This could be consistent with either of the upper two scenarios shown in Fig. 3 of Shen et al. below, where the trajectories in their figure are for sulfide being fractionated relative to sulfate...

Fig. 3. Plots of $\Delta^{36}\text{S}$ vs $\Delta^{33}\text{S}$. Unfilled circles are measurements of North Pole barite. Dashed grey line is $\Delta^{36}\text{S} = -\Delta^{33}\text{S}$ and represents approximation of a reference mass-independent fractionation array. Grey dotted lines indicate principal axes of plots. Vectors (dotted black lines with arrowheads) illustrate deviation from the mass-independent fractionation array for sulfide relative to sulfate that is related by (a) equilibrium isotope fractionation (yellow filled triangles) and two component mixing with sulfate (green line), (b) sulfur disproportionation (black diamonds) observed in laboratory culture experiments (Johnston et al., 2005a; 2007), (c) sulfate reduction as observed (blue triangles) in laboratory culture experiments (Farquhar et al., 2003a; Johnston et al., 2005a; 2007) and as predicted by network models (black line) experiments (Johnston et al., 2007; Farquhar et al., 2007b), and (d) pyrite from North Pole reported here (red filled squares). Error bars are estimates of 2 sigma uncertainties.

...however, pyrite with higher $\Delta^{36}\text{S}$ than sulfate seems to also require large $\delta^{34}\text{S}$ differences, as shown in Fig. 5 of Shen et al...

Fig. 5. Plot of the deviation of from the reference mass-independent fractionation array ($\Delta^{36}\text{S} + \Delta^{33}\text{S}$) (from Fig. 1 - $\Delta^{36}\text{S} = -\Delta^{33}\text{S}$) versus the isotopic fractionation of $^{34}\text{S}/^{32}\text{S}$ between sulfate and sulfide ($\delta^{34}\text{S}_{\text{sulfate}} - \delta^{34}\text{S}_{\text{sulfide}}$). Unfilled circles are measurements of North Pole barite referenced to estimated average composition for sulfate. Dashed grey line is $\Delta^{36}\text{S} = -\Delta^{33}\text{S}$ and represents approximation of a reference mass-independent fractionation array. Plots for (5a) equilibrium isotope fractionation (yellow filled triangles) and two component mixing with sulfate (green curves), (5b) sulfur disproportion observed in laboratory culture experiments (Johnston et al., 2005a; 2007) (black diamonds), (5c) sulfate reduction as observed in laboratory culture experiments (Farquhar et al., 2003a; Johnston et al., 2005a; 2007) (blue unfilled triangles) and as predicted by network models (Johnston et al., 2007; Farquhar et al., 2007b) (red lines) and (5d) pyrite from North Pole reported here (red filled squares). Vectors (dotted lines with arrowheads) in 5c and 5d illustrate the direction and limits of deviation from the mass-independent fractionation array for sulfide relative to sulfate that is found in experiments with cultures of microbial sulfate reducers. Error bars are estimates of 2 sigma uncertainties.

As seen in the above figure, 2-component mixing (panel a) and sulfur disproportionation (panel b) show small ^{36}S enrichments in sulfide relative to sulfate concurrent with strong ^{34}S depletions in sulfide relative to sulfate (noting that in the panels above, ^{34}S relationships are shown as sulfate-sulfide).

In comparison, Kazput barite and pyrite, although showing some ^{36}S enrichments in pyrite relative to barite, have $\delta^{34}\text{S}$ values that appear identical within error. Thus, we assert, though at present cannot yet test, that the observed $\Delta^{36}\text{S}$ differences between Kazput barite and pyrite may be an analytical artifact. Hopefully, the assessment of ^{36}S will be improved in the future due to technical advances in mass spectrometry, and the significance of small ^{36}S fractionations can be interpreted with confidence.

For now, we do not interpret the $\Delta^{36}\text{S}$ difference between Kazput barite and pyrite further due to the abovementioned reasons.

Gomes and Johnston report mostly 18eps between 0 and 20‰ but they also report values as low as -10‰ for 18O fractionation between H₂O and Sulfate in the case of sulfide oxidation (maybe not quite enough to explain the low values observed here but could be

relevant in an anoxic environment). They also report very negative fractionation if oxygen comes from O₂. Even with a δ¹⁸O of O₂ similar to today this could generate sulfate with negative δ¹⁸O (though not as negative as -30‰, not without assuming that O₂ had a different δ¹⁸O compared to today).

We were familiar with some, but not all, of the references for oxygen isotope fractionation in sulfate that were reviewed in Gomes and Johnston, 2017² but we were pleased to look at the issue of ¹⁸O fractionation between sulfate and water in more detail because it was very informative.

Despite being aware that cases exist where sulfate δ¹⁸O can be even lower than the δ¹⁸O of its water oxidant, and realizing that it was problematic to try to confidently assign a source water δ¹⁸O composition based on Kazput barite δ¹⁸O values, we presently rely on a qualitative comparison to make our argument - Kazput barite δ¹⁸O are compared with their host carbonate δ¹⁸O versus what is typical for coeval carbonate and sulfate δ¹⁸O through time (observed in Figure 1).

That said, the studies referenced in Gomez and Johnston that show the δ¹⁸O of sulfate being lower than the δ¹⁸O of their water-oxygen source make it clear that this scenario has limited application to natural samples, and in fact may represent a bias from experimental results due to the use of "heavy" water that is highly enriched in ¹⁸O.

Roughly, the pattern that appears is that the difference in δ¹⁸O of sulfate vs. water should be positive when the starting water is <15‰, typical for cases with seawater and meteoric waters, but negative Δδ¹⁸O sulfate-water when the starting water is >15‰. Many lab experiments use ¹⁸O-labelled water with heavy (highly positive) δ¹⁸O values, likely contributing to the current lack of clarity about the fractionation between sulfate and water and how it should be interpreted with respect to natural systems. The study by van Everdingen and Krouse, 1987⁵ (cited by Gomes and Johnston), despite being an older paper, reviewed the situation well, as shown in two figures from that study below...

Figure 1.-- $\delta^{18}O_s$ values of SO_4^{2-} versus $\delta^{18}O_w$ values of H_2O for oxidation experiments.

Figure 2.-- $\delta^{18}O_s$ values of SO_4^{2-} versus $\delta^{18}O_w$ values of H_2O for field samples.

In both figures the lines labeled "100%" are where $d_{18O}\text{-sulfate} = d_{18O}\text{-water}$.

As observed in the first figure of results from oxidation experiments, when water d_{18O} is $<15\text{‰}$ the sulfate data generally falls above the 100% line with some data falling on that line. For $d_{18O}\text{-water} >15\text{‰}$, the 100% line crosses over and becomes the uppermost line, and thus sulfate d_{18O} is either roughly equal to, or below the 100% line - where below the 100% line sulfate-water d_{18O} is negative.

In contrast to the experimental results, the second figure of only field sample data have water sources with $d_{18O} <15\text{‰}$, and the corresponding difference in d_{18O} between sulfate and water is positive.

Further illustrating the situation where sulfate d_{18O} can be lower than corresponding water are the extreme cases of the Schwarz and Cortecchi, 1974⁶ experimental data (noted in the first of the two figures above), where the $d_{18O}\text{-water}$ is $+127\text{‰}$ and the $d_{18O}\text{-sulfate}$ is $+71\text{‰}$, making the difference in d_{18O} between sulfate and water extremely negative at -56‰ (!)

In another more recent experimental sulfide oxidation study that was referenced in Gomes and Johnston, Poser et al., 2014⁷, the use of labeled waters with d_{18O} ranging extremely heavy, from $+95.4\text{‰}$ to $+115.6\text{‰}$ resulted in product sulfates with sulfate-

water d18O differences of -57.2‰ to -84.2‰... again, sulfates that are extremely depleted with respect to their source water oxidants!

In the pyrite oxidation study of Kohl and Bao, 2011⁸ (not cited in Gomes and Johnston), their use of labeled waters ranging +14.7‰ to 18.7‰, very close to the 15‰ crossover value in the figures from Van Everdingen and Krouse, 1987 mentioned previously above, resulted in pyrite-derived sulfate with mostly muted negative sulfate-water d18O differences ranging -5.3‰ up to (only one positive value) of +1.2‰.

We know of only one case of natural samples with sulfate being apparently 18O-depleted compared to their ambient water, in Hendry et al., 1989⁹. While those sulfates were interpreted as deriving from sulfide oxidation, they are sulfates in groundwaters where it is uncertain if the coexisting groundwaters were contemporaneous with the time of sulfide oxidation. Therefore, in this case the sulfate being lower in d18O values than the coexisting water shows an apparent negative fractionation that may not be original. The data can be seen in the figure below, plotting below the 1:1 line, while it is noteworthy that most data do fall above that line, whereas the sulfates are mostly enriched than their water oxygen sources.

Fig. 3. Plot of $\delta^{18}\text{O}$ of dissolved sulfate versus $\delta^{18}\text{O}$ of associated groundwater. The arrow represents a mixing trend (see text).

As far as the negative fractionation between sulfate and O₂ during sulfide oxidation, indeed that appears to be the case, with the only clearly defined experiment done to determine this being very old. Lloyd, 1968¹⁰ report that the oxygen involved in the oxidation of sodium sulfide (taken to represent sulfides, generally) was fractionated by -8.7‰ d18O. Whereas the 18O-enrichment in O₂ versus seawater (the Dole Effect) is mostly due to respiration that enriches O₂ by ~20‰ compared to seawater, regardless of whether it is respiration on land or in the ocean (reviewed in Luz et al., 2014¹¹), this

scenario would still result in sulfide-derived sulfate that is relatively heavy (d18O near 11‰, if O2 is at ~20‰) compared to the Kazput barites.

Although it is outside the scope of our present study to do this topic the justice it deserves, it is clear there are some unexamined assumptions about the relationship between sulfide-derived sulfate and its water and O2 oxidants that need testing.

In light of the above considerations, we assert that the assumption of sulfate being 18O-enriched versus its water oxidant errs on the conservative side for the case of the Kazput barites, as they are natural samples with low d18O values that are far below the aforementioned range of water d18O >15‰ that appear to result in sulfates that can be much lower in d18O than their water oxidants.

In the main text we now add two new sentences (the latter two below), and a reference to the van Everdingen and Krouse, 1987⁵ study discussed above, to address this important issue of sulfate-water oxygen isotope fractionation:

L232-242: As reviewed by Gomes and Johnston², the oxygen isotope fractionation between pyrite-derived SO_4^{2-} and water spans a range of $\delta^{18}O$ values falling between 0‰ and +20‰ for aerobic and anaerobic conditions, typical of natural samples. However, oxidation experiments using waters >15‰ have produced sulphide-derived sulphates that can display very negative ^{18}O -fractionations versus their source waters, as exemplified by an extreme case of -65‰ fractionation between pyrite-derived sulphate (+71‰) oxidized in isotopically labeled water (+127‰)⁵. Although the topic of sulphate oxygen isotope fractionation demands clarification, we contend that Kazput barites have $\delta^{18}O$ values well within the range, below 15‰, of natural sulphates that are ^{18}O -enriched versus their water oxidant sources.

I have a minor question about this new version. The authors state that a negative correlation between d18O and D33S is indicative of the sulfate origin. They could explain this point better, but this is not my main concern. This negative correlation stands only if they consider the points between 112 and 135 m (points between 100 and 130 m used to be red in the previous version: why the change?). I could not find an explanation as of why they did not consider the entire Kazput formation, in which case they would have simply stated that there is no correlation. If there were a reason to select only those points, what would it imply for the points that do not fall on the correlation? As a result, the authors should probably either justify this subselection and provide additional explanation or remove any reference to the correlation and find a different way to make their argument.

The previous version of Figure 2 is pasted much further above, as is the new version of Figure 2. As seen, yes, the subset of data highlighted for its d18O- Δ 33S correlation has been changed from 100-130 m previously, to 112-135 m in the current version. This change is because it was noted that the correlation held for d18O and Δ 33S for the previous selection, but not as well for the d34S. The current selection highlights the region of the core where d18O-d34S as well as d18O- Δ 33S show correlations.

It is indeed the case that if the whole dataset is considered, such correlations are not apparent. Even though it is not through the whole dataset, we note that this is perhaps the first time such a correlation between oxygen isotopes and $\Delta^{33}\text{S}$ has been observed in sulfate. In fact, it is rare for correlation between significant (or, anomalous) $\Delta^{33}\text{S}$ and any geochemical parameters at all. That there is a general, though imperfect, correlation between high $\delta^{34}\text{S}$, high $\Delta^{33}\text{S}$, and low $\delta^{18}\text{O}$ within the center of the Kazput carbonate seems to indicate a particular end member composition within a mixing scenario. Our experience with riverine sulphate informs the interpretation here, whereas on a continental scale, correlation between S- and O-isotopes can exist within a spread of data, but if the whole riverine sulfate dataset is considered then it may be said that no S-O isotope correlation exists.

We suggest that the change from considering a S-O isotope correlation between 100 and 130 m (previously) to 112 and 135 m (now) improves the assessment of the end member composition of most interest - the end member that has the highest $\Delta^{33}\text{S}$ coincident with the highest $\delta^{34}\text{S}$.

New sentences added within the following sections to help make the significance of the S-O correlations and their relationship to the sulphate end member more clear:

L260-264: An MSR influence can be indicated by positive $\delta^{18}\text{O}$ - $\delta^{34}\text{S}$ correlation in sulphate¹², however, the negative correlations between Kazput barite $\delta^{18}\text{O}$ and its sulphur isotope parameters (Fig. 3c & 3d) imply that source mixing, with sulphide weathering in meteoric waters, was much more important than MSR for setting these S and O isotope signatures.

L303-319: Taken together, the S- and O-isotope systematics suggest a singular scenario for the origin of the Kazput barite. The barite and pyrite $\Delta^{33}\text{S}$ values reach a maximum in the studied drill core as compared to the two other Turee Creek Group drill cores in older underlying sediments¹. The strongest expression of this sulphur source appears with maxima of $\Delta^{33}\text{S}$ and $\delta^{34}\text{S}$ coincident with the $\delta^{18}\text{O}$ minimum in barite, where negative correlations between $\delta^{18}\text{O}$ and $\Delta^{33}\text{S}$ and $\delta^{34}\text{S}$ data exist between 112-135 meters depth within the center of the carbonate unit (Fig. 3c and 3d). Such S-O isotope correlation may indicate the mixing of sulphide-derived SO_4^{2-} sources with distinct S-isotope compositions but overlapping $\delta^{18}\text{O}$ ranges due to their oxidation in meteoric waters under variable humidity, as is observed in major river systems today^{13,14}. However, the preservation of such a strongly riverine sulphate signal in marine sedimentary rocks, such as occurred in Kazput barites, is unlikely today due to buffering by present-day high seawater sulphate concentrations. A previously identified source of sulphur in the Turee Creek Basin with a monotonous $\Delta^{33}\text{S} \approx 0.9\text{‰}$ is well expressed in the Meteorite Bore Member diamictite¹. Mixing between this monotonous sulphur source and a different, continental-weathering, source with a higher $\Delta^{33}\text{S} \approx 1.6\text{‰}$ is observed in S-O isotope space (Fig. 3c and 3d).

I feel however that my and reviewer 3's main comments have been answered and that the manuscript is much closer to publication.

Reviewer #2

NOTE that the reviewer's comments were originally provided in comments in the tracked changes on a .doc file. Comments and previous version manuscript text (preceded by line numbers Lxx-xx:) are pasted here with our responses. Editorial changes to the manuscript text on the tracked changes doc from Reviewer #2 have been adopted throughout the revised manuscript.

L25-26: However, in the transition, signals of anoxia and oxygenation are mixed

What do mean by this? Prior to permanent oxygenation? The former, to me, implies an interim state...

Clarified in new version by designating a specific time interval, as below:

L23-27: After permanent atmospheric oxygenation, anomalous sulphur isotope compositions were lost from sedimentary rocks, demonstrating that atmospheric chemistry ceded its control of Earth's surficial sulphur cycle to weathering. However, ca. 2.5 to 2.3 billion years (Ga) ago, the sulphur cycle's signals of anoxia and oxygenation are mixed...

L32: Detailed evidence suggests

What detailed evidence. You mean "These data"?

The "detailed evidence" includes the data AND its context that is examined in the text, whereas of course in the abstract we are too limited in space (150 words) to elaborate, but we specify more closely now, as below:

L31-31: Geochemical and sedimentary evidence suggests

A stronger last sentence would be nice

Yes, attempted to make it stronger, as below in the new version:

L33-35: Our findings indicate that incipient oxidative continental weathering, ca. 2.8-2.5 Ga or earlier, may be diagnosed with such a combination of low $\delta^{18}\text{O}$ and high $\Delta^{33}\text{S}$ in sulphates.

L40-42: Rivers supply the ocean with substantial loads of dissolved sulphate (SO_4^{2-}) derived from the oxidation of pyrite and the dissolution of sulphate minerals.

Perhaps nice to include an estimate on this?

Added Burke et al 2018 reference for this. Not sure if what is desired here is a total flux estimate or an estimate of the proportion of riverine sulphate flux - the latter is chosen because it is more directly relevant, and useful, here than a total flux, whereas discussing the total river SO₄ flux versus current ocean SO₄ reservoir would seem to require an elaboration that might be too distracting:

L39-41: Rivers supply the ocean with substantial loads of dissolved sulphate (SO₄²⁻) derived from the oxidation of pyrite and the dissolution of sulphate minerals in roughly equal proportions¹⁵.

L42-43: Once in the ocean, the sulphur and oxygen isotope composition of SO₄²⁻ is modified during microbial sulphate reduction (MSR) processes.

Also along flow-path. MSR doesn't care about being in the oceans.

Certainly, but from studying the Mississippi River sulfate budget (Killingsworth and Bao, 2015; Killingsworth et al., 2018), and from the recent global River sulfate assessment by the Burke et al., 2018 study, it appears that MSR is not an important player in riverine systems when considered on a continent-, and global-, scale, although in some rivers MSR could be significant.

You introduce oxygen here but don't say anything about it in the paragraph. Perhaps simply to sulphur initially?

Agree. Removed the mention of oxygen from this first introduction paragraph.

L46-48: The mass-dependent sulphur fractionations induced by MSR lead to marine sulphide having a large, but relatively ³⁴S-depleted, range in δ³⁴S (see Methods for isotope notations) and a Δ³⁶S/Δ³³S slope of ~-9¹⁶.

This is a complicated sentence consider simplifying.

Agreed, now simplified as:

L45-47: Because of mass-dependent sulphur fractionations from MSR, marine sulphide has a large, but relatively ³⁴S-depleted, range in δ³⁴S (see Methods for isotope notations) and a Δ³⁶S/Δ³³S slope of approximately -9 (ref. ¹⁶).

(on the slope above of -9 above).. I thought this was shallower. Around 7 according to Ono, right?

In Ono et al, 2006, the theoretical equilibrium prediction is -7 - that is the reference you mean, is that correct? - but the Johnston et al, 2006 (cited here) shows an empirical, and

different, slope of -9 for sulfate reduction as compared to equilibrium, so the latter slope value is used here.

(on the end of the intro paragraph pasted above).. Again a lot of information in a small space for a general reader

Understood, however we are limited on space and need to introduce the context.

L62-64: The permanent loss of anomalous $\Delta^{33}\text{S}$ signals from the sedimentary record is a crucial constraint on the rise of oxygen, but its timing remains unclear. This transition is constrained in continuous stratigraphic section only in S. Africa.

This isn't strictly fair. Three cores in south Africa.

Sure, now changed to:

L65-67: This transition is constrained in continuous stratigraphic section only in S. Africa, within three different age-equivalent cores from the Transvaal Basin.

L71-74: Since all of these studies interpret the record of S-MIF directly, independent tests are needed to delineate the atmospheric and weathering controls on the early Paleoproterozoic surface sulphur cycle ca. 2.5 to 2.3 Ga.

I would leave out the timing of this as you cant test it directly with 2.31 year old rocks.

Good point. Changed accordingly:

L75-78: Since all of these studies interpret the record of S-MIF directly, independent tests are needed to delineate the importance of atmospheric and weathering controls on the early surficial sulphur cycle.

L82-93: Furthermore, as CAS concentrations should be proportional to ambient seawater SO_4^{2-} concentrations, trivial early Earth seawater sulphate can challenge the recovery of measurable CAS records. In the late Archaean, before 2.5 Ga, marine SO_4^{2-} may have been less than 2.5 μM , as inferred from modern analog environments¹⁷. Between 2.25 to 2.1 Ga, estimated seawater SO_4^{2-} concentrations were 5-20 mM, as based on the $\delta^{34}\text{S}$ variability in extant CAS records¹⁸. While at 2 Ga, the SO_4^{2-} concentration was at least 10 mM, as geochemically constrained with the first basin-scale bedded evaporites¹⁹. During the GOE interval, between 2.5 to 2.3 Ga, however, it can be difficult to obtain enough sulphate from sediments for sulphur isotope measurements, where up to 1500 grams of carbonate may be necessary for sufficient CAS²⁰. Such low sulphate sample yields may prohibit the additional measurement of oxygen isotopes.

I'm not sure any of this is needed. It doesn't add anything and detracts from the story. Delete.

This referenced section on early Earth seawater sulphate concentration estimates was added in response to Reviewer #1, whereas they noted that seawater sulphate concentrations could have been high in the Paleoproterozoic. As such, we imagine that it could still be important to include the state of the art on early Earth seawater sulphate concentrations, which appear to be quite dynamic, so this section was left in.

(refers to sentence about barite detection that was moved to the results section)

This was originally in a strange place: It just seemed quite random. I have moved it here because it doesn't interfere with the flow of the prose. I'm not saying that it must stay here, there are probably other places where it could be satisfactorily relocated, just giving an option.

Suggestion appreciated and used - this sentence on barite detection was moved from to the results section, as below:

L129-131: *Although barites were observed in acid-digested residues, the grains were too small (<3 μm) to be identified via standard transmitted light microscopy, and thus only imaged via SEM (Supplementary Figs. 3 & 4).*

L133-135: The barite of the Kazput Formation was studied within an overall-shallowing succession of the Turee Creek Group that shows evidence for increasing atmospheric oxygenation in its Mn-enriched units and decreasing iron content²¹.

I haven't read that cited paper thoroughly but does it need to be atmospheric? Can it be locally derived? This also seems disjointed. Do you mean oxidant availability? Please smooth the wording here. I have tried but I am not as familiar as you are.

As an aside, is the increase in Mn and decrease in Fe controlled by lithology or is it seen in the same lithological units. We all expect carbonates to have different elemental characteristics than siliclastics, proper.

Indeed, the point holds that it doesn't need to be only atmospheric oxygenation. However, the cited paper interprets it as such, so the wording is changed ("interpreted in favor of") to make that clear. Also, it's certainly true that the evidence for oxygenation can be nuanced due to redox-chemical changes that are dependent on lithology and in turn need to be detangled in terms of proxies that register oxygen in the water column versus proxies that more convincingly show atmospheric oxygenation. However, the overall sweep of things - loss of Fe and concentration of Mn, seems pretty significant.

In further support of this, a new N-isotope paper with samples from Kazput carbonate finds evidence of oxygenation in $\delta^{15}\text{N}$ and iron speciation results. We update the text to also introduce that study and its implications here, where it is further discussed later in the text:

L137-145: *The analysed barites are from the Kazput Formation of the Turee Creek Group, an overall-shallowing succession with Mn-enriched units and decreasing iron contents that have been interpreted in favor of increasing atmospheric oxygen²¹. Recently*

reported evidence from the Kazput Formation further indicates free oxygen availability during its deposition²². An age of 2.25 Ga was assigned, as based on detrital zircon U-Pb dating and estimated sedimentation rates²³. While this age may be appropriate, in order to be conservative with respect to what may be a remarkably young record of isotopically anomalous sulphur, here we simply assume that the Kazput Formation is younger than 2.31 Ga, the age constraint from the lower Meteorite Bore Member diamictite¹.

(referring to moved text shown below) This is repetition from elsewhere. I've moved it here but I'd delete it as it detracts from the text. Simply integrate the cross references within the prose, which I think you've done.

Text moved, as below:

L145-148: Geologic maps locating the TCDP cores (Supplementary Fig. 1), core photographs (Supplementary Fig. 2), and transmitted light photomicrographs with complementary XRF analyses (Supplementary Figs. 5 & 6) are provided in the Supplementary Information.

L137-140: Barlow and authors²⁴ previously conducted detailed lithostratigraphic analysis of the ~350 m Kazput Formation. They described the diversity and prevalence of stromatolites, designating 5 facies associations according to lithologic and microbial morphologic variations.

Do you mean this?...

Barlow and authors³² conducted a detailed lithostratigraphic analysis of the Kazput Formation, describing the prevalence and diversity of stromatolites.

Changed to "frequency" instead of "prevalence" to simplify.

L150-152: Barlow and authors²⁴ conducted detailed lithostratigraphic analysis of the ~350 m Kazput Formation, describing the frequency and diversity of its stromatolites.

L140-142: The lowermost Kazput carbonate from the TCDP3 core used in our study is consistent with their facies association A.

What is this? Please insert a few words of clarification

Facies association A now clarified, as below:

L152-155: Of their 5 facies associations, the carbonate examined here is consistent with their facies association A that is described as mm- to cm-scale laminated fine-grained carbonate with locally occurring dome-shaped stromatolites.

L146-148: Although barite was observed in rock digestion residues (Supplementary Figs. 3 & 4), it is not easily found in place in the rock, with grains of less than 3 µm in size only detected by SEM scan of a full thin section.

This is a weird place for this sentence. Move to somewhere better?

Sure, now moved, as previously shown further above, to L129-131

L156-157: Moreover, the barite S isotope compositions strictly overlap with those of pyrites in the same TCDP3 core (Fig. 2).

I think you can add a few words here to expand on why this similarity is important. For example, add “implying X/Y/Z”

Important suggestion, thanks. General implications added, as below:

L164-171: Importantly, the barite S isotope compositions closely overlap with those of pyrites in the same TCDP3 core (Fig. 2), which implies that the sulphur precursor was mostly unmodified during its conversion to reduced (sulphide) or oxidized (sulphate) mineral phases, or that one phase was derived from the other without isotopic fractionation. Because, under various conditions, it is possible for low temperature redox reactions to proceed without S-isotope fractionation between sulphate and reduced sulphur phases², by itself, the S-isotope match between Kazput barite and pyrite cannot be used to diagnose a specific process.

L157-164: The generation of, and preservation of, S-MIF in Kazput barite could appear to show the persistence or reappearance of atmospheric anoxia younger than 2.31 Ga. Alternatively, it could indicate local control on the capture of isotopically anomalous sulphur that was being weathered under an oxygenated atmosphere. Considered all together, we suggest that a picture emerges of weathering and environmental controls on a regional scale; however, taken individually, the results and contextual observations may be attributable to multiple different processes.

I think it's a little premature to lay your cards on the table here. A general reader may well be lost. Also, I think you're being too specific. As I see it you have two options to explain the data: atmosphere or not. As you discuss, Not = post-depositional processes and inheritance from weathering. I think you are better trying to argue against the former to leave the latter as your preferred explanation. I have tried to reorganise a bit here to streamline your prose and keep the general readers.

Excellent suggestion. New text added and suggested changes adopted, as below...

L172-179: Broadly, however, the S-MIF in Kazput barite requires an atmospheric sulphur source that originated under an atmosphere with <0.001% of present atmospheric oxygen, occurring within one of two scenarios: either the capture of such atmospheric sulphur under an essentially oxygen-free atmosphere existing at <2.31 Ga,

or the reworking/recycling of an archival atmospheric sulphur source that was generated any time before 2.31 Ga. Understanding how and when these barites inherited their atmospherically derived sulphur is of fundamental importance to understanding Earth's oxygenation and its imprints on the minor sulphur isotope record.

L166-172: A possible origin of the nearly identical $\delta^{34}\text{S}$ (and $\Delta^{33}\text{S}$) compositions of Kazput barite and pyrite is low temperature oxidation of pyrite during sample preparation in the laboratory. The sulphur isotope composition of pyrite-derived SO_4^{2-} can closely match its pyrite precursor under abiotic or biologically mediated oxidation^{2,25}. While pyrite oxidation produces dissolved SO_4^{2-} as its immediate product, barite formation further requires the capture of SO_4^{2-} with barium, which makes it more difficult to accidentally produce in the lab during sample handling.

I am not sure these sentences ran together smoothly. I have joined them and reversed their order. Is that better?

This works well, yes. Sentence order now changed and merged here:

L181-187: Given that pyrite-derived SO_4^{2-} can closely match the sulphur isotope composition of its pyrite precursor under abiotic or biologically mediated oxidation^{2,25}, the nearly identical $\delta^{34}\text{S}$ (and $\Delta^{33}\text{S}$) compositions of Kazput barite and pyrite could be attributed to low temperature pyrite oxidation during sample preparation in the laboratory. While pyrite oxidation produces dissolved SO_4^{2-} as its immediate product, barite formation further requires the capture of SO_4^{2-} with barium, which makes it more difficult to accidentally produce in the lab during sample handling.

L178-179: A high-temperature metamorphic origin for the barite must be considered, as it could explain the occurrence of their significant S-MIF, irrespective of age.

I think you should specifically say how high metamorphism can cause the observations. My, perhaps simplistic view on it, is that metamorphism can smear things out via remobilisation and maybe even shallow the slopes by TSR but is temperature important?

Mention of "high temperature" removed here. Sentence added to give an example of how metamorphism can change S-MIF: Cui et al reference. The reason why high-T would be important is that the S isotope fractionation between sulfate and sulfide goes small - a possible explanation for the barite and pyrite S isotope match. In any case, that is already precluded so an elaboration on that might just bog down text, so that is left out.

L193-197: A metamorphic origin for the barite must also be considered, as it could explain the occurrence of their S-MIF signals, irrespective of the host rock's age. For example, post-depositional metamorphic processes have been implicated in the remobilization and dilution of S-MIF signals in sulphur from N. American Palaeoproterozoic Huronian sections²⁶.

L185-186: Considering the apparent genetic link between the Kazput barite and pyrite

What do you mean by this?

Removed phrase "genetic link" - understand that it is perhaps too vague. In place, now using "S isotope match":

L203: Considering the S isotope match between the Kazput barite and pyrite

L202: such as for sulphate from terrestrial settings versus marine carbonates (Fig. 1b).

This seems quite a random statement. Is it needed?

The point is that SO₄ from a terrestrial setting will have water-O from terrestrial sources, in contrast to a marine carbonate that gets its O from seawater. Now highlights the specific cases of interest in Fig. 1 for clarity:

L244-246: These low sulphate values are readily apparent in concurrent non-marine sulphates versus marine carbonates occurring at ca. 0.0, 0.6, and 1.4 Ga (Fig. 1b).

L217-218: Therefore, as compared to the diagenetic pyrite, the different mode of occurrence but mutual sulphur source for the barite precludes its detrital origin.

Perhaps useful to remind the readers that you've now precluded later effects.

Note that the order of paragraphs in the discussion has changed so the logic flow is different. But it is a useful suggestion, now added here:

L268-270: We suggest, given the previous evidence that has ruled out post-depositional processes and a detrital origin while pointing to a meteoric water oxygen source, that the O isotope signals preserved in Kazput barite are best explained by pyrite oxidation on land.

L223-226: Kazput barite $\delta^{18}\text{O}$ data show negative correlation with $\Delta^{33}\text{S}$ and $\delta^{34}\text{S}$ data (Fig. 3c and 3d), which, given the evidence, is most parsimoniously explained by a shared provenance of oxygen and sulphur in low temperature oxidation of pyrite to form SO_4^{2-} .

what evidence? Be specific.

This line has been removed in the new version.

L226-229: We suggest that the negative isotopic correlation, where low $\delta^{18}\text{O}$ is associated with high $\Delta^{33}\text{S}$, fingerprint the source of stronger S-MIF signals being delivered to the Turee Creek Basin as coming from the weathering of sedimentary rock exposures on land.

Not sure this makes sense. Can you simplify

Now simplified to:

L299-301: *The association between low $\delta^{18}\text{O}$ and high $\Delta^{33}\text{S}$ in the barites identifies the vector of the strongest S-MIF signals as being from the weathering of sedimentary sulphides on land.*

L233: Turee Creek Basin

Doesn't the basin have a specific name?

Is "specific" meant in terms of a sub-basin? Not that we are aware of. Besides, it is useful and clear to keep it as the familiar Turee Creek Basin, as the paper that is referenced on sulphur microorganisms in this section, by Schopf et al, only refers to the "Turee Creek Basin" as well.

L246: source effect

What do you mean.

Changed to "source mixing", in context below:

L261-264: *...the negative correlations between Kazput barite $\delta^{18}\text{O}$ and its sulphur isotope parameters (Fig. 3c & 3d) imply that source mixing, with sulphide weathering in meteoric waters, was much more important than MSR for setting these S and O isotope signatures.*

L249-250: We suggest that the context of Kazput barite and its isotopic signatures are explained by pyrite oxidation on land

Maybe explicitly say why you want terrestrial?

As the discussion surrounding the low $\delta^{18}\text{O}$ of Kazput barite has changed significantly in light of the new Galili et al., 2019 paper on seawater $\delta^{18}\text{O}$ (now cited), which is, alongside the S-isotope correlations with $\delta^{18}\text{O}$, key for our interpretation of a terrestrial (as opposed to marine) origin for the sulphate, the whole paragraph of discussion is pasted below:

L268-301: *We suggest, given the previous evidence that has ruled out post-depositional processes and a detrital origin while pointing to a meteoric water oxygen source, that the O isotope signals preserved in Kazput barite are best explained by pyrite oxidation on land. Kazput barite $\delta^{18}\text{O}$, extending to -19.5‰ , are among the lowest values in the sedimentary record, including other sulphate occurrences going down to -20.3‰ from a Neoproterozoic snowball Earth glacial diamictite deposited at 635 Ma^{50} , and in more recent terrestrial deposits from within the Arctic Circle that reach down to -19.7‰^{25}*

(Fig. 1b). Similarly, the range of oxygen isotope compositions of Kazput barite may require a glacial meltwater oxidant source. However, in contrast to those other examples of very low $\delta^{18}\text{O}$ in sulphate, Kazput barite $\delta^{18}\text{O}$ may not necessitate glacial or ice meltwater oxygen sources, and instead could reflect a hydrosphere anchored to seawater with a lower $\delta^{18}\text{O}$ than today. Meteoric waters evolve lower $\delta^{18}\text{O}$ compositions than seawater due to evaporation from seawater followed by Rayleigh distillation in cooling air masses²⁶. The most evolved meteoric waters (and snow and ice melt) that can be found at high latitudes, high elevations, and away from coastlines are marked by the lowest $\delta^{18}\text{O}$ compositions relative to seawater. This relative relationship between seawater and more ^{18}O -depleted meteoric waters is significant here, as it has been suggested that the $\delta^{18}\text{O}$ of seawater (0‰ at present²⁶) may have changed through time, perhaps being as low as -10‰ in the Archaean²⁸. Newly reported iron oxide $\delta^{18}\text{O}$ records, which are insensitive to temperature, convincingly support that seawater was lower in the past, having a $\delta^{18}\text{O}$ at near -8‰ by around 2.0 Ga⁵¹. Palaeoproterozoic seawater appears to be faithfully recorded in Kazput carbonate that is, as mentioned previously, on the order of 10‰ lower than modern carbonates but comparable to carbonates from around 2.3 Ga. Therefore, the Kazput barites recording the lowest $\delta^{18}\text{O}$ indicate their precursor sulphate was oxidized in the most evolved meteoric waters, placing this water-oxygen source for this sulphate firmly on land, as compared to a seawater $\delta^{18}\text{O}$ that was likely around -10‰. Although sulphate-water oxygen isotope fractionation during sulphide oxidation requires further study, we take a median sulphide-derived sulphate-water $\delta^{18}\text{O}$ fractionation value of +10‰³⁷, as compared to the barite, to roughly estimate that the water sources for Kazput sulphate may have ranged from -30‰ to -8‰. This is mostly in the meteoric water range but overlaps the possible seawater value of -10‰, while extending from meteoric to glacial water if seawater was closer to 0‰ like today. The association between low $\delta^{18}\text{O}$ and high $\Delta^{33}\text{S}$ in the barites identifies the vector of the strongest S-MIF signals as being from the weathering of sedimentary sulphides on land.

L251-253: The first-order controls on these signals are relative sea-level and the exposure of sedimentary rocks to weathering

How about ground level weathering. Can we preclude that?

Assuming this refers to differentiating between the cases of weathering where the oxygen that is being produced is consumed in situ, or that weathering occurs due to free O₂ in the atmosphere... the short answer is that it remains to be determined, perhaps in the future, if these two scenarios can be told apart. However, in this case, new d¹⁵N evidence appears consistent with atmospheric O₂ being available at the time of the Kazput Formation, a study that is now referenced in this section of the discussion, as below..

L346-357: Furthermore, recent iron speciation and nitrogen isotope evidence reported by Cheng and authors²² implies free oxygen was available during the deposition of the Kazput carbonate. In samples from core TCDP3 from that study, iron speciation and nitrogen isotope data was found to be consistent with suboxic to oxic water column

conditions and free atmospheric oxygen concentrations well above 0.001% of present atmospheric levels. The availability of significant free oxygen concurrent with sulphur isotope anomalies strongly supports the latter signals being a "memory effect" carried in sulphate of oxidative weathering origin³². It follows that the attendant $\Delta^{36}\text{S}/\Delta^{33}\text{S}$ variations in the barites may reflect control from oxidative weathering of pyrite in continental source rocks of various ages, whereas the S-MIF signal is simply tracking the integrated sulphur isotope composition of the rocks being weathered.

L253-254: that should be expressed as a strong source effect on the S-isotope compositions of SO_4^{2-} .

It's not clear what you mean here. Please clarify.

Mixing of sources. This is clarified throughout the text - no more "source effect", "source mixing" or "mixing of sources" used instead.

L254-255: The Fe/Al ratios in core TCDP3 are 0.33 on average, excluding enriched intervals >1

How reliable do you think these are in carbonates? I would expect carbonates to have different trace element characteristics when compared to average SHALE

Since the Fe/Al ratios are examined, along with iron speciation, in the above-cited Cheng et al study, we do not report original Fe/Al data here, and therefore please see their study for discussion about the comparison with average shale. However, we note that going from 5-10% to ~50% carbonate does not appreciably change the Fe/Al ratios (Fig. 2).

L259-260: while the few measured pyrite concentrations are 0.32 weight % on average (Supplementary Table 2).

This seems random. Perhaps delete.

There were no pyrite concentrations reported for this core previously, so, in this context, the pyrite concentrations support the idea that, overall, the water column + sediments may not have been anoxic enough to where pyrite would be highly concentrated throughout, save for some isolated intervals. Therefore, we leave this in.

L262-263: Anoxic sediment porewaters favoring MSR would also provide a flux of barium in sufficient concentration to precipitate SO_4^{2-} as barite at the sediment-water interface³³.

Do we have to have barite precipitation in the basin rather than the catchment? Surely that implies that the sulphate all had negative d18O is that possible? If so could the basin have been restricted? This is tagged on lower but I think its important.

The discussion of the barite d18O and why it requires that the sulphate formed in meteoric water on the continent (a separate issue as compared to the precipitation of the sulphate as barite) is pasted above, L268-299.

The discussion of the barite precipitation in the basin is discussed here:

L319-328: The first-order controls on the mix of sulphur sources are relative sea level and the exposure of sedimentary rocks to weathering. Minimal overprinting resulted in a largely faithful transfer of S- and O-isotope signatures as sulphate was captured as barite or reduced to form pyrite. Anoxic sediment porewaters favoring MSR would provide a flux of barium in sufficient concentration to precipitate SO_4^{2-} as barite at the sediment-water interface³³. Remaining porewater SO_4^{2-} would be quantitatively reduced to sulphide, via closed system MSR, with capture by iron to form iron sulphides (pyrites) during early diagenesis in the sediment¹. This is consistent with the lack of CAS-sulphate from the Kazput carbonate that supports near-quantitative reduction of SO_4^{2-} in porewaters.

And yes, the Turee Creek Basin could have been relatively restricted (or relatively more restricted than S. African basins, as they are all intracratonic basins), though its carbonate d18O and d13C imply global seawater signatures. Line added about basin restriction as well, at the tail-end of the paragraph of discussion comparing with S. Africa:

L388-390: Finally, the Turee Creek Basin may have also been relatively more isolated from the global ocean as compared to contemporaneous S. African basins.

L271-273: Such mixing, between sulphide sources with distinct S-isotope compositions but similar overlapping ranges of $\delta^{18}O$ values of their resulting SO_4^{2-} , has been observed in major river systems today^{13,14}.

Do you mean this? If so, this need to be explained a little better. For example, sulphide has no oxygen....

Thanks for catching this. Clarified now as "sulphide-derived" sulphate:

L309-312: Such S-O isotope correlation may indicate the mixing of sulphide-derived SO_4^{2-} sources with distinct S-isotope compositions but overlapping $\delta^{18}O$ ranges due to their oxidation in meteoric waters under variable humidity, as is observed in major river systems today^{13,14}.

L283: This is consistent with the lowest $\delta^{18}O$ reported

What is?

This sentence no longer exists.

Thus, via analogy, sulphate $\delta^{18}\text{O}$ values approaching -20% , precipitated in a low-latitude setting

I think this statement is useful here and strengthens the oddity of your data. Can you include it and reference it out?

Referenced text added for low latitude settings..

L365-366: Similarly, both sedimentary successions were deposited in intracratonic basins for S. Africa¹² and W. Australia³⁴, near the equator⁷.

However, the discussion about low d18O has changed significantly, as already pasted further above (in L268-301), whereas a glacial meltwater source is not required to explain the barite data when the contemporaneous seawater d18O during Kazput deposition was likely near -10% (again, as discussed further above).

Is this what you are saying: Too light for equatorial waters? Can you reference this with Palaeomag data.

Please see last response and changes above.

L296-299: It might be possible for snowmelt water from high altitude mountainous regions to contribute to ^{18}O -depleted signatures in SO_4^{2-} , but we believe that this is unlikely due to the significant flux of such sulphate, and its preservation, that is required by the Kazput barite record.

I don't understand what you're saying here. This isn't clear please clarify?

This discussion on d18O has changed, as above.

L299-301: Considering the time window of deposition for the Kazput Formation, its barite $\delta^{18}\text{O}$ might be linked to meltwaters from the final, ca. 2.25 Ga, Paleoproterozoic glaciation

Providing that they are not reworked of course?

The above line has been deleted.

L290-296: Three Paleoproterozoic glacial episodes are recognized in W. Australia, with minimum detrital zircon U-Pb ages of $<2.460 \pm 0.009$ Ga for the Boolgeeda glacial diamictite and $<2.340 \pm 0.022$ Ga for the lower Meteorite Bore Member diamictite²³, with the latter further constrained by a Re-Os age of 2312.7 ± 5.6 Ma from pyrites within core TCDP2¹. The youngest Paleoproterozoic glaciation, only recognized in S. Africa thus far, is constrained between $2250-2240$ Ma³⁵ to 2256 ± 6 Ma³⁶.

I am not sure that this detail is warranted? The references are good and what you're saying is correct but it detracts from your story. I have moved this around a lot. I think this is what you wanted to say but it was lost. Do you agree?

Agree, especially now as a glacial meltwater source is not necessitated by the barite d18O. Therefore, the details above are now removed.

L311-316: The S-MIF signals of the Kazput barite show relatively small $\Delta^{36}\text{S}$ and $\Delta^{33}\text{S}$ ranges in comparison to the larger ranges observed in Neoarchaeon sulphur, where this limited range of S-MIF signals in the Turee Creek Group has been interpreted in favor of homogenization during the weathering of older, more isotopically variable sulphur which was previously formed in an anoxic atmosphere¹.

The higher slopes are rarer and speak to specific time intervals. Do you think the trends in the CAP-CAP data reflect weathering? If so why are the silts steeper? Do the detrital zircons help the story? For example is there evidence of a provenance change.

While the detrital zircon ages in Caquineau et al (cited in this paper) are intriguing for their relation to weathering in Turee Creek Basin, linking the detrital zircon record to that of sulphur here seems problematic due to the differential mobility of detrital zircon vs. sulphide and sulphate. Most importantly, the stratigraphic resolution of the zircon record is not high enough to compare it with the $\Delta^{36}\text{S}/\Delta^{33}\text{S}$ slope change within this one core.

L318-319: Sulphur isotope distributions compared between S. Africa and W. Australia (Fig. 4) reveal preservational controls on S-MIF during the GOE interval, ca. 2.45 to 2.2 Ga.

What about SIMS data?

A detailed SIMS S isotope comparison between W. Australia and S. Africa would be great, but here, because it is bulk data from the barite, we restrict the comparison between bulk data instead of SIMS. Also, there is no SIMS sulphate data from S. Africa during this interval, either, since it is only from CAS records thus far...

L351-353: An alternative possibility is that the expansion of $\Delta^{33}\text{S}$ before 2.5 Ga is due to changing oxidation state of volcanic gases³⁷.

Any gases I guess. Biogenic gases might be important too. I'd deemphise the statement and cite to people that have previously speculated on such matters.

The cited work stated volcanic gases, but point noted. Leaving it more open now. Changed to "gas influxes" instead:

L397-399: *Another possibility is that the expansion of $\Delta^{33}\text{S}$ before 2.5 Ga is due to changing oxidation state of gas influxes⁵⁹.*

L433-435: Barite in TCDP3 was only observed by scanning electron microscopy detection of barites <10 µm in size in sample residue and in a full thin section scan, but not in petrographic observations of thin sections.

This sentence didn't make sense. I have tried to reword it for you. IS THIS CORRECT. IF not please amend but its must be changed as I have no clue what you were saying.

Yes thanks! Indeed, that was a garbled sentence that was overlooked. Oops! Fixed:

L487-488: Barite in TCDP3 was only observed by scanning electron microscopy, where barites less than 10 µm were observed in both sample residues and in thin sections.

Figure 3 caption

Can you write out the significance of the open and closed data points?

Yes, now added to caption of Figure 2, where the filled symbols are first shown.

L792-794: The filled symbols are from the core depth interval 112-135 m with specific sulphur-oxygen isotope correlations that are discussed in the text and shown in figure 3.

Figure 4

Why are the data from Luo et al not included here? Are they and not referenced in the figure caption? This is an important oversight that should be corrected.

Does it change if you include the SIMS Data? Perhaps you could include an insert to the Australian data with the SIMS from reference 15, 37 and Swanner et al (2013).

Luo et al data now included in the histogram in Figure 4, and this figure was pasted much earlier here, in the response to Reviewer #1.

As for the SIMS data, as mentioned before, we avoid discussion and emphasis on SIMS data simply because the new barite data from this study are bulk data and we want to do a direct comparison with other bulk data. Also, again, there is thus far a lack of sulphate SIMS data from Paleoproterozoic rocks in general.

Figure 1

Can you include the data from Izon et al. (2015; 2017). I don't know why people seem to ignore these datasets but there is a lot of high magnitude data in this dataset, which will help frame your argument.

Absolutely. Now included the data from Izon et al 2015 and 2017. It was only not included before due to relying on two large compilations from other workers - so it was an oversight. It is now in the new version of Figure 1, in the upper panel here:

Supplementary Fig. 1 caption

L60-61: 84.3m: return to laminated carbonated siltstone / low carbonate content.

Is this not contradictory?

Fixed:

L60-61: 84.3m: return to laminated carbonated siltstone.

L61: 77.1m: laminated carbonated siltstone near top of core.

Is this right or do you mean carbonate cemented.

No, just carbonated siltstone. Left as is.

Supplementary Fig. 3

These (note: the spectra) could be labelled A-C to make the caption more explicit.

Suggestion appreciated, but left as is because it seems clear already.

Supplementary Fig. 4

Why two of the same sample? Is this correct?

Yes that is correct. Well.. they are subsamples of one sample, isolated sections of the sample powder residue from one sample, to be more accurate.

REFERENCES

- 1 Philippot, P. *et al.* Globally asynchronous sulphur isotope signals require re-
definition of the Great Oxidation Event. *Nature Communications* **9**, 2245,
doi:10.1038/s41467-018-04621-x (2018).
- 2 Gomes, M. L. & Johnston, D. T. Oxygen and sulfur isotopes in sulfate in modern
euxinic systems with implications for evaluating the extent of euxinia in ancient

- oceans. *Geochim. Cosmochim. Acta* **205**, 331-359, doi:<https://doi.org/10.1016/j.gca.2017.02.020> (2017).
- 3 Luo, G. *et al.* Rapid oxygenation of Earth's atmosphere 2.33 billion years ago. *Science Advances* **2**, e1600134 (2016).
- 4 Shen, Y., Farquhar, J., Masterson, A., Kaufman, A. & Buick, R. Evaluating the role of microbial sulfate reduction in the early Archean using quadruple isotope systematics. *Earth Planet. Sci. Lett.* **279**, 383-391 (2009).
- 5 van Everdingen, R. O. & Krouse, H. R. Interpretation of isotopic compositions of dissolved sulfates in acid mine drainage. (1988).
- 6 Schwarcz, H. & Cortecchi, G. Isotopic analyses of spring and stream water sulfate from the Italian Alps and Apennines. *Chem. Geol.* **13**, 285-294 (1974).
- 7 Poser, A. *et al.* Stable sulfur and oxygen isotope fractionation of anoxic sulfide oxidation by two different enzymatic pathways. *Environmental science & technology* **48**, 9094-9102 (2014).
- 8 Kohl, I. & Bao, H. M. Triple-oxygen-isotope determination of molecular oxygen incorporation in sulfate produced during abiotic pyrite oxidation (pH=2-11). *Geochim. Cosmochim. Acta* **75**, 1785-1798, doi:10.1016/j.gca.2011.01.003 (2011).
- 9 Hendry, M. J., Krouse, H. R. & Shakur, M. A. Interpretation of oxygen and sulfur isotopes from dissolved sulfates in tills of southern Alberta, Canada. *Water Resour. Res.* **25**, 567-572, doi:10.1029/WR025i003p00567 (1989).
- 10 Lloyd, R. M. Oxygen isotope behavior in the Sulfate-Water System. *J. Geophys. Res.* **73**, 6099-6110, doi:10.1029/JB073i018p06099 (1968).
- 11 Luz, B., Barkan, E. & Severinghaus, J. P. in *Treatise on Geochemistry (Second Edition)* (eds Heinrich D. Holland & Karl K. Turekian) 363-383 (Elsevier, 2014).
- 12 Brunner, B. *et al.* The reversibility of dissimilatory sulphate reduction and the cell-internal multi-step reduction of sulphite to sulphide: insights from the oxygen isotope composition of sulphate. *Isot. Environ. Health Stud.* **48**, 33-54 (2012).
- 13 Calmels, D., Gaillardet, J., Brenot, A. & France-Lanord, C. Sustained sulfide oxidation by physical erosion processes in the Mackenzie River basin: Climatic perspectives. *Geology* **35**, 1003-1006 (2007).
- 14 Killingsworth, B. A., Bao, H. & Kohl, I. E. Assessing Pyrite-Derived Sulfate in the Mississippi River with Four Years of Sulfur and Triple-Oxygen Isotope Data. *Environmental Science & Technology* **52**, 6126-6136, doi:10.1021/acs.est.7b05792 (2018).
- 15 Burke, A. *et al.* Sulfur isotopes in rivers: Insights into global weathering budgets, pyrite oxidation, and the modern sulfur cycle. *Earth Planet. Sci. Lett.* **496**, 168-177, doi:<https://doi.org/10.1016/j.epsl.2018.05.022> (2018).
- 16 Johnston, D. T., Farquhar, J. & Canfield, D. E. Sulfur isotope insights into microbial sulfate reduction: When microbes meet models. *Geochim. Cosmochim. Acta* **71**, 3929-3947, doi:<http://dx.doi.org/10.1016/j.gca.2007.05.008> (2007).
- 17 Crowe, S. A. *et al.* Sulfate was a trace constituent of Archean seawater. *Science* **346**, 735-739, doi:10.1126/science.1258966 (2014).
- 18 Planavsky, N. J., Bekker, A., Hofmann, A., Owens, J. D. & Lyons, T. W. Sulfur record of rising and falling marine oxygen and sulfate levels during the

- Lomagundi event. *Proc Natl Acad Sci U S A* **109**, 18300-18305, doi:10.1073/pnas.1120387109 (2012).
- 19 Blättler, C. L. *et al.* Two-billion-year-old evaporites capture Earth's great oxidation. *Science*, doi:10.1126/science.aar2687 (2018).
- 20 Guo, Q. *et al.* Reconstructing Earth's surface oxidation across the Archean-Proterozoic transition. *Geology* **37**, 399-402 (2009).
- 21 Van Kranendonk, M. J., Mazumder, R., Yamaguchi, K. E., Yamada, K. & Ikehara, M. Sedimentology of the Paleoproterozoic Kungarra Formation, Turee Creek Group, Western Australia: A conformable record of the transition from early to modern Earth. *Precambrian Res.* **256**, 314-343 (2015).
- 22 Cheng, C. *et al.* Nitrogen isotope evidence for stepwise oxygenation of the ocean during the Great Oxidation Event. *Geochim. Cosmochim. Acta*, doi:<https://doi.org/10.1016/j.gca.2019.07.011> (2019).
- 23 Caquineau, T., Paquette, J.-L. & Philippot, P. U-Pb detrital zircon geochronology of the Turee Creek Group, Hamersley Basin, Western Australia: timing and correlation of the Paleoproterozoic glaciations. *Precambrian Res.*, doi:<https://doi.org/10.1016/j.precamres.2018.01.003> (2018).
- 24 Barlow, E., Van Kranendonk, M., Yamaguchi, K., Ikehara, M. & Lepland, A. Lithostratigraphic analysis of a new stromatolite–thrombolite reef from across the rise of atmospheric oxygen in the Paleoproterozoic Turee Creek Group, Western Australia. *Geobiology* **14**, 317-343 (2016).
- 25 Balci, N., Shanks, W. C., Mayer, B. & Mandernack, K. W. Oxygen and sulfur isotope systematics of sulfate produced by bacterial and abiotic oxidation of pyrite. *Geochim. Cosmochim. Acta* **71**, 3796-3811, doi:10.1016/j.gca.2007.04.017 (2007).
- 26 Cui, H. *et al.* Searching for the Great Oxidation Event in North America: A Reappraisal of the Huronian Supergroup by SIMS Sulfur Four-Isotope Analysis. *Astrobiology* **18**, 519-538 (2018).
- 27 Peng, Y., Bao, H., Zhou, C., Yuan, X. & Luo, T. Oxygen isotope composition of meltwater from a Neoproterozoic glaciation in South China. *Geology* **41**, 367-370, doi:10.1130/g33830.1 (2013).
- 28 Crockford, P. W. *et al.* Claypool continued: Extending the isotopic record of sedimentary sulfate. *Chem. Geol.*, doi:<https://doi.org/10.1016/j.chemgeo.2019.02.030> (2019).
- 29 Luz, B. & Barkan, E. Variations of $^{17}\text{O}/^{16}\text{O}$ and $^{18}\text{O}/^{16}\text{O}$ in meteoric waters. *Geochim. Cosmochim. Acta* **74**, 6276-6286, doi:10.1016/j.gca.2010.08.016 (2010).
- 30 Kasting, J. F. *et al.* Paleoclimates, ocean depth, and the oxygen isotopic composition of seawater. *Earth Planet. Sci. Lett.* **252**, 82-93 (2006).
- 31 Galili, N. *et al.* The geologic history of seawater oxygen isotopes from marine iron oxides. *Science* **365**, 469-473, doi:10.1126/science.aaw9247 (2019).
- 32 Reinhard, C. T., Planavsky, N. J. & Lyons, T. W. Long-term sedimentary recycling of rare sulphur isotope anomalies. *Nature* **497**, 100-104 (2013).
- 33 Magnall, J. *et al.* Open system sulphate reduction in a diagenetic environment—Isotopic analysis of barite ($\delta^{34}\text{S}$ and $\delta^{18}\text{O}$) and pyrite ($\delta^{34}\text{S}$) from the Tom and

- Jason Late Devonian Zn–Pb–Ba deposits, Selwyn Basin, Canada. *Geochim. Cosmochim. Acta* **180**, 146-163 (2016).
- 34 Gumsley, A. P. *et al.* Timing and tempo of the Great Oxidation Event. *Proceedings of the National Academy of Sciences*, doi:10.1073/pnas.1608824114 (2017).
- 35 Schröder, S., Beukes, N. J. & Armstrong, R. A. Detrital zircon constraints on the tectonostratigraphy of the Paleoproterozoic Pretoria Group, South Africa. *Precambrian Res.* **278**, 362-393, doi:<https://doi.org/10.1016/j.precamres.2016.03.016> (2016).
- 36 Rasmussen, B., Bekker, A. & Fletcher, I. R. Correlation of Paleoproterozoic glaciations based on U–Pb zircon ages for tuff beds in the Transvaal and Huronian Supergroups. *Earth Planet. Sci. Lett.* **382**, 173-180, doi:10.1016/j.epsl.2013.08.037 (2013).
- 37 Halevy, I., Johnston, D. T. & Schrag, D. P. Explaining the structure of the Archean mass-independent sulfur isotope record. *Science* **329**, 204-207, doi:10.1126/science.1190298 (2010).

REVIEWERS' COMMENTS:

Reviewer #1 (Remarks to the Author):

Dear Authors

Sorry for the delay in submitting my review. As this is now my third review of the paper and being away from the office, I focused on the manuscript itself. Similarly to what happened between the previous rounds, it has evolved, for the better.

The paper has become stronger as it now contains discussions of the main points (ages, geological/sedimentary context, more thorough discussion of the origin of the d_{18O} signatures, locus of sulfate formation, relationship between pyrite and barite isotopic signatures).

I feel that I cannot longer contribute with new comments or suggestions and I thank the authors for their improvement of the text.

One small detail : line 319, I'm not sure that a ref to fig. 3d makes sense here, as the sentence is also about D33S.

Sincerely,

Guillaume Paris

Reviewer #2 (Remarks to the Author):

The manuscript by Killingsworth et al. has undergone a lengthy review process, which I believe has improved the quality, impact and longevity of the manuscript. Having reviewed the manuscript, and the accompanying reviewers' comments, I believe that the authors have made every effort to address the outstanding concerns and I support the publication of the manuscript.

Given the length of the review process, I believe that the manuscript has increased in length and, on occasion, is a little difficult to follow. In the appended word document I have suggested changes that hopefully improve the accessibility of the manuscript and some minor comments. I don't think that the authors need to adopt all these changes piecemeal but they may want to consider them more generally to improve the accessibility of their paper.

Finally, I apologise for the delay in delivering this review. I hope that the appended comments prove useful and I see the manuscript in press soon.

G

Reviewer #1:

One small detail : line 319, I'm not sure that a ref to fig. 3d makes sense here, as the sentence is also about D33S.

Good catch. Removed ref to Fig. 3d, now only 3c, as follows:

L316-318: Mixing between this monotonous sulphur source and a different, continental-weathering, source with a higher $\Delta^{33}\text{S} \approx 1.6\text{‰}$ is observed in S-O isotope space (Fig. 3c).

Reviewer #2 (comments pasted here from tracked changes doc):

L25-27: However, ca. 2.5 to 2.3 billion years (Ga) ago, the sulphur cycle's signals of anoxia and oxygenation are mixed, requiring independent tests, such as with oxygen isotopes in sulphate.

This is quite a generic sentence. Perhaps it would be better rewritten?

Now rewritten for (hopefully) better emphasis as:

L27-30: However, the existence of mixed signals of anoxia and oxygenation in the sulphur isotope record between 2.5 to 2.3 billion years (Ga) ago requires independent clarification, for example via oxygen isotopes in sulphate.

Changes to original L41-42: Within the marine realm, microbial sulphate reduction (MSR) exerts the most important control on the isotopic composition of SO_4^{2-} .

Is this what you meant? The existing sentence was quite strange

Response: the better sentence structure is appreciated, and adopted, but fractionation needs to be here, as controls on SO₄ composition would also include mixing of sources, etc. Changed to:

L44-46: Within marine and terrestrial settings, microbial sulphate reduction (MSR) processes exert the most important controls on sulphur isotopic fractionation of SO₄²⁻.

L87-91: In the late Archaean, before 2.5 Ga, marine SO₄²⁻ may have been less than 2.5 μM, as inferred from modern analog environments²¹. Between 2.25 to 2.1 Ga, estimated seawater SO₄²⁻ concentrations were 5-20 mM, as based on the δ³⁴S variability in extant CAS records²², while at 2 Ga, the SO₄²⁻ concentration was at least 10 mM, as geochemically constrained with the first basin-scale bedded evaporites²³.

This doesn't seem relevant and detracts from the flow.

Response: Understood that this wasn't the right place. Text is now moved and reworked to fit within the later discussion section, where it is much more relevant:

L364-372: Geochemical constraints from the first basin-scale bedded evaporites at 2 Ga indicate a significant marine SO₄²⁻ concentration of at least 10 mM⁵⁴, however, between 2.25 to 2.1 Ga, the δ³⁴S variability in extant CAS records imply a range of seawater SO₄²⁻ concentrations of 5-20 mM⁵⁵. Meanwhile, modern analog environments suggest late Archaean marine SO₄²⁻ concentrations of less than 2.5 μM⁵⁶. As a result of such sulphate concentrations that could span four orders of magnitude between 2.5 to 2.1 Ga, sulphur isotope variations would be very sensitive to local environmental controls. We suggest that such local control extends to the record of sulphur isotope anomalies.

L112-114: The 2.45 Ga^{30,31} to 2.2 Ga³² Turee Creek Group sedimentary succession from W. Australia is ideal for seeking barite records to characterize the surface sulphur cycle around the time of the GOE.

Seems weird to say GOE if you argue that its not true...

Response: Point taken. Now changed to a more open phrasing:

L112: ...around the time of atmospheric oxygenation.

L139-141: Recently reported evidence from the Kazput Formation further indicates free oxygen availability during its deposition³⁵.

Probably useful to explicitly say what this evidence is. Nitrogen and Fe-Speciation?

Response: Indeed, it was more specific before. Now changed to:

L133-135: Recently reported **nitrogen isotope and iron speciation** evidence from the Kazput Formation further indicates free oxygen availability during its deposition³¹.

L152-156: Of their 5 facies associations, the carbonate examined here is consistent with their facies association A that is described as mm- to cm-scale laminated fine-grained carbonate with locally occurring dome-shaped stromatolites. Above and below it, the carbonate horizon grades into, and out of, finely cross-laminated carbonate-rich siltstones (Fig. 2).

This seemed to be overly detailed for the targeted journal. I have removed some of the detail and brought the text more in line with what you've done. This is nit-picking and could be ignored...

Response: Not ignored, and in fact, appreciated. Suggestions taken into account in new version here:

L148-151: Our examined carbonates are fine-grained with mm- to cm-scale laminations, resembling facies association A of Barlow and co-authors³³, who further described local occurrences of dome-shaped stromatolites. The carbonate horizon gradates into, and out of, finely cross-laminated carbonate-rich siltstones (Fig. 2).

Changes to original L232-247: As reviewed by Gomes and Johnston³⁷, the oxygen isotope fractionation between pyrite-derived SO_4^{2-} and water spans a range of $\delta^{18}\text{O}$ values between 0‰ and +20‰ for naturally relevant conditions. Oxidation experiments using waters with $\delta^{18}\text{O}$ values higher than 15‰ have produced sulphide-derived sulphates with very negative $\delta^{18}\text{O}$ values versus their source waters, as exemplified by an extreme case of a -65‰ fractionation between pyrite-derived sulphate (+71‰) oxidised in isotopically labelled water (+127‰)⁴⁶. Although the topic of sulphate oxygen isotope fractionation demands clarification, we contend that the Kazput barites have $\delta^{18}\text{O}$ values well within the range, below 15‰, of natural sulphates that are ^{18}O -enriched versus their water oxidant sources. Regardless of the assumptions concerning sulphate-water O-isotope fractionation, distinct water-oxygen sources are required to produce sulphate $\delta^{18}\text{O}$ values that are appreciably lower than those of coeval carbonates require. These low sulphate values are clear in concurrent non-marine sulphates versus marine carbonates occurring at ca. 0.0, 0.6, and 1.4 Ga (Fig. 1b). Similarly, the low $\delta^{18}\text{O}$ values of our barites, relative to the coeval Kazput carbonates, require that the barites had a meteoric-water-oxygen source.

I'm not overly familiar with this. It seems complicated. Check that I've not altered this. It may benefit from simplifying for the general reader

Response: It is complicated but simplification is good where possible. Suggestions incorporated into the text below. To be clear, it is not "very negative $\delta^{18}\text{O}$ values versus their source waters", it is negative fractionation - in other words, relative and not necessarily negative or positive in $\delta^{18}\text{O}$ values. New text as below:

L229-243: As reviewed by Gomes and Johnston³⁴, the oxygen isotope fractionation between pyrite-derived SO_4^{2-} and water spans a range of $\delta^{18}\text{O}$ values between 0‰ and +20‰ for naturally relevant conditions. Oxidation experiments using waters with $\delta^{18}\text{O}$ compositions >15‰ have produced sulphide-derived sulphates that have very low $\delta^{18}\text{O}$ compositions versus their source waters, as exemplified by an extreme case of -65‰ fractionation between pyrite-derived sulphate (+71‰) oxidized in isotopically labeled water (+127‰)⁴³. Although the topic of sulphate oxygen isotope fractionation demands clarification, we contend that Kazput barites are well within the range of $\delta^{18}\text{O}$ values, below 15‰, of natural sulphates that are ^{18}O -enriched versus their water oxidant sources. Regardless of assumptions concerning sulphate-water oxygen isotope fractionations, distinct water-oxygen sources are required to produce sulphate $\delta^{18}\text{O}$ compositions that are appreciably lower than those of coeval carbonates. These low sulphate compositions are apparent in non-marine sulphates as compared to concurrent marine carbonates at ca. 0.0, 0.6, and 1.4 Ga (Fig. 1b). Similarly, relative to coeval Kazput carbonates, the low $\delta^{18}\text{O}$ values of our barites require a meteoric-water-oxygen source.

Changes to original L257-260: We can imagine that sulphur-oxidising bacteria may have had a limited contribution to signals preserved by the Kazput barites, perhaps by introducing oxygen isotope signatures of Turee Creek Basinal water to SO_4^{2-} produced during re-oxidation of the MSR-produced sulphide, or by enhancing rates of sulphide oxidation on land.

Check this. IT reads weird and I'm not sure what you are trying to say.

Response: Simplified the sentence as follows:

L253-256: Sulphur-oxidizing bacteria may have contributed to the Kazput barites' oxygen isotope signals, perhaps by introducing Turee Creek Basinal water-oxygen to the SO_4^{2-} produced during re-oxidation of MSR-produced sulphide, or by enhancing rates of sulphide oxidation on land.

L336-337: The few available pyrite concentration measurements are 0.32 weight % on average (Supplementary Table 2).

Seems a random sentence.

Response: Merged with its preceding sentence so it hopefully appears as less of an orphan, in context as follows:

L333-337: The enrichments in Fe/Al and total sulphur observed at 164 and 182 meters depth in two muddy intervals below the carbonate are also likely pyrite (FeS_2) enrichments. **These horizons of pyrite enrichments were not quantified directly, while a few pyrite concentration results show 0.32 weight % on average (Supplementary Table 2).**

L341-342: This slope, however, steepens to near -1.5 in the siltstones above and below the carbonate interval (Fig. 2).

Is this true? Looks like they're in carbonate per the figure.

Response: You are correct. It is going towards the siltstones, but still within the carbonate. Now fixed:

L341-342: This slope, however, steepens to near -1.5 within the upper and lower portion of the carbonate (Fig. 2).

Changes to original L346-351: Furthermore, recent iron-speciation and nitrogen isotope evidence imply that suboxic–oxic water column conditions prevailed beneath an atmosphere containing appreciable oxygen³⁵.

The two juxtaposed sentences were largely repetitive. I've tried to run them into one another to prevent this and ensure that the text follows better.

Response: Yes, much better. Change adopted:

L346-348: Furthermore, recent iron-speciation and nitrogen isotope evidence imply that suboxic-to-oxic water column conditions prevailed beneath an atmosphere containing appreciable oxygen³¹.

Changes to original L351-354: The apparent availability of significant free oxygen ($>0.001\%$ PAL), concurrent with mass-independent sulphur isotope anomalies, strongly supports the latter being a "memory effect" carried by sulphate derived from oxidative weathering of MIF-bearing rocks¹⁵.

The Neoproterozoic stuff I did shows that the water column was periodically oxic by iron speciation....

I think this tone is needed as. I don't think that Fe-speciation speaks to the atmosphere. The Neoproterozoic wiggle had low HR/T Fe ratios, also.

Response: Here it is not Fe-speciation alone that speaks to the atmospheric state. Regardless, this tone is fair, and is adopted as below:

L348-352: The apparent availability of significant free O₂ ($>0.001\%$ PAL), concurrent with mass independent sulphur isotope anomalies, strongly supports the latter being due to a memory effect carried by sulphate derived from oxidative weathering of S-MIF-bearing rocks¹⁵.

Changes to original L369-373: Frequency distributions of $\delta^{34}\text{S}$ data support this assertion, with $\delta^{34}\text{S}$ data from W. Australia data displaying a broadly unimodal distribution, centred around $\sim 0\text{‰}$ (Fig. 4a); while data from S. Africa show a bimodal $\delta^{34}\text{S}$ distribution (Fig.

4b), reflecting the preferential incorporation of ^{32}S into sulphide at the expense of sulphate.

Changes within original L378-381: Closed system MSR is also suggested by the $\delta^{34}\text{S}$ - $\Delta^{33}\text{S}$ systematics.

This section was quite hard to read at a quick pass. I have tried to make it clearer.

Response: Agreed, changes incorporated as below:

L374-381: Frequency distributions of $\delta^{34}\text{S}$ data support this assertion, with W. Australian sulphur $\delta^{34}\text{S}$ data displaying a broadly unimodal distribution, centered near 5‰ (Fig. 4a); while S. African data show a more bimodal $\delta^{34}\text{S}$ distribution (Fig. 4b), reflecting the preferential incorporation of ^{32}S into sulphide at the expense of sulphate. These different behaviors are also manifested in their respective $\Delta^{33}\text{S}$ distributions. Despite similar ranges of $\Delta^{33}\text{S}$ values, an important source of mass-dependent sulphur ($\Delta^{33}\text{S} \approx 0\text{‰}$, Fig. 4d) is indicated for S. Africa, while two sources of S-MIF-bearing sulphur ($\Delta^{33}\text{S} = +0.9\text{‰}$ and $+1.6\text{‰}$, Fig. 4c) are observed from W. Australia.

L382-390: The $\delta^{18}\text{O}$ compositions of CAS from S. Africa are yet to be measured. As suggested by the S-isotope evidence for open system MSR, it is likely that the sulphate from S. Africa would be relatively enriched in ^{18}O as compared to the Kazput barite. Further, we suggest that globally low marine sulphate concentrations and an intracratonic basin setting made the capture of weathering input fluxes of SO_4^{2-} , with their S- and O-isotope compositions relatively intact, possible in the Turee Creek Basin. Finally, the Turee Creek Basin may have also been relatively more isolated from the global ocean as compared to contemporaneous S. African basins.

We've gone from Sulphur to Oxygen. This seems strange is something missing here?

Some of these points are common to the south African records

Response: Tried to do a better job of bridging from sulphur to oxygen. The point is a prediction for S. African data with which to help test the different mechanisms in each basin that preserved their respective S isotope signals. Also removed mention of Turee Creek as intracratonic basin - yes, so are S. African basins, which was previously mentioned in the same paragraph. As below:

L386-391: Finally, to help elucidate the possibility that the Turee Creek Basin may have been relatively more isolated from the global ocean as compared to contemporaneous S. African basins, the $\delta^{18}\text{O}$ compositions of CAS from S. Africa may be useful but are yet to be measured. As suggested by the S-isotope evidence for open system MSR, it is likely that sulphate from S. Africa is relatively enriched in ^{18}O as compared to the Kazput barite.

Changes to original L394-397: For example, the large magnitude $\Delta^{33}\text{S}$ values seen before 2.5 Ga have been explained by changing the locus of MSR⁵⁸ or, via the evolution of the atmosphere's oxidation state⁵⁹.

Check I've not changed the meaning.

Response: The clarity and condensation is appreciated, but there is some important nuance lost (what is the locus of MSR; and that it is not atmospheric state, but influx oxidation state that changes) that would be good to preserve. Changed as follows:

L397-400: For example, the enhanced preservation of large magnitude $\Delta^{33}\text{S}$ compositions in the lead up to 2.5 Ga have been explained by a shift in the locus of MSR from euxinic water columns to sediment porewaters due to oxygenation of shallow surface oceans⁵⁸, or by the changing oxidation state of gas influxes⁵⁹.

L447-448: Through their oxygen isotope records, it may be possible to constrain this early flux of SO_4^{2-} in the Neoarchaeon.

Strange last sentence.

Response: Yes, perhaps it leaves things too vague. Changed to:

L448-452: By using oxygen isotope compositions to identify sulphide-derived sulphates of continental weathering origin from the rock record, it may be possible to directly constrain the timing of early weathering fluxes of SO_4^{2-} and their relative significance for the late Archaean surficial sulphur and oxygen cycles.

Text moved to top of Methods: Drill core samples were manually crushed in a tungsten-carbide piston chamber before being powdered in an agate ring and puck mill.

Seems more sensible to have this upfront?

Response: Agreed, sentence moved from barite methods section to here, just not at the top of the paragraph. Now:

L457-459: Drill core samples were first manually crushed in a tungsten-carbide piston chamber before powdering in an agate ring and puck mill.

Deleted L453-454: and despite CAS yields too low for analyses of oxygen and sulphur isotope compositions, this is an important negative result.

I'm not sure you need this. It's repeated below.

Response: Agreed, deleted.

L487-488: Barite in TCDP3 was only observed by scanning electron microscopy, where barites less than 10 μm were observed in both sample residues and in thin sections.

Useful to shout to the supplement here?

Response: Agreed, now:

L489-491: Barite in TCDP3 was only observed by scanning electron microscopy, where barites less than 10 μm were observed in both sample residues (Supplementary Figs. 3 and 4) and in thin sections.

Figures look good. Some suggestions on how to clarify the figure captions.

(refers to text edits)

Response: Suggestions appreciated and adopted.